# Demo2Code: From Summarizing Demonstrations to Synthesizing Code via Extended Chain-of-Thought

**Huaxiaoyue Wang**
Cornell University
yukiwang@cs.cornell.edu

**Gonzalo Gonzalez-Pumariega**
Cornell University
gg387@cornell.edu

**Yash Sharma**
Cornell University
ys749@cornell.edu

**Sanjiban Choudhury**
Cornell University
sanjibanc@cornell.edu

## Abstract

Language instructions and demonstrations are two natural ways for users to teach robots personalized tasks. Recent progress in Large Language Models (LLMs) has shown impressive performance in translating language instructions into code for robotic tasks. However, translating demonstrations into task code continues to be a challenge due to the length and complexity of both demonstrations and code, making learning a direct mapping intractable. This paper presents `Demo2Code`, a novel framework that generates robot task code from demonstrations via an *extended chain-of-thought* and defines a common latent specification to connect the two. Our framework employs a robust two-stage process: (1) a recursive summarization technique that condenses demonstrations into concise specifications, and (2) a code synthesis approach that expands each function recursively from the generated specifications. We conduct extensive evaluation on various robot task benchmarks, including a novel game benchmark `Robotouille`, designed to simulate diverse cooking tasks in a kitchen environment. The project's website is at https://portal-cornell.github.io/demo2code/

## 1 Introduction

How do we program home robots to perform a wide variety of *personalized* everyday tasks? Robots must learn such tasks online, through natural interactions with the end user. A user typically communicates a task through a combination of language instructions and demonstrations. This paper addresses the problem of learning robot task code from those two inputs. For instance, in Fig. 1, the user teaches the robot how they prefer to make a burger through both language instructions, such as "make a burger", and demonstrations, which shows the order in which the ingredients are used.

Recent works [24, 23, 33, 80, 61, 35] have shown that Large Language Models (LLMs) are highly effective in using language instructions as prompts to plan robot tasks. However, extending LLMs to take demonstrations as input presents two fundamental challenges. The first challenge comes from demonstrations for long-horizon tasks. Naively concatenating and including all demonstrations in the LLM's prompt would easily exhaust the model's context length. The second challenge is that code for long-horizon robot tasks can be complex and require control flow. It also needs to check for physics constraints that a robot may have and be able to call custom perception and action libraries. Directly generating such code in a single step is error-prone.

***Our key insight is that while demonstrations are long and code is complex, they both share a latent task specification that the user had in mind.*** This task specification is a detailed language

37th Conference on Neural Information Processing Systems (NeurIPS 2023).

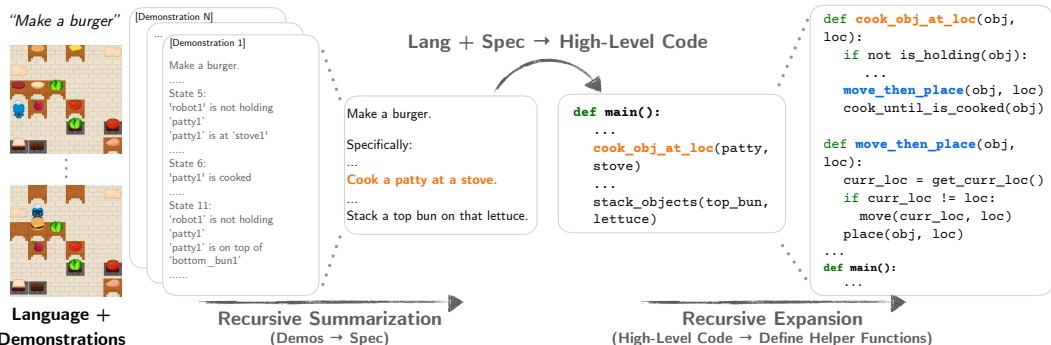

Figure 1: Overview of `Demo2Code` that converts language instruction and demonstrations to task code that the robot can execute. The framework recursively summarizes both down to a specification, then recursively expands the specification to an executable task code with all the helper functions defined.

description of how the task should be completed. It is latent because the end user might not provide all the details about the desired task via natural language. We build an extended chain-of-thought [73] that recursively summarizes demonstrations to a compact specification, maps it to high-level code, and recursively expands the code by defining all the helper functions. Each step in the chain is small and easy for the LLM to process.

We propose a novel framework, `Demo2Code`, that generates robot task code from language instructions and demonstrations through a two-stage process (Fig. 1). *(1) Summarizing demonstrations to task specifications:* Recursive summarization first works on each demonstration individually. Once all demonstrations are compactly represented, they are then jointly summarized in the final step as the task specification. This approach helps prevent each step from exceeding the LLM's maximum context length. *(2) Synthesizing code from the task specification:* Given a task specification, the LLM first generates high-level task code that can call undefined functions. It then recursively expands each undefined function until eventually terminating with only calls to the existing APIs imported from the robot's low-level action and perception libraries. These existing libraries also encourage the LLM to write reusable, composable code.

Our key contributions are:

1. A method that first recursively summarizes demonstrations to a specification and then recursively expands specification to robot code via an extended chain-of-thought prompt.
2. A novel game simulator, `Robotouille`, designed to generate cooking tasks that are complex, long-horizon, and involve diverse food items, for benchmarking task code generation.
3. Comparisons against a range of baselines, including prior state of the art [33], on a manipulation benchmark, `Robotouille`, as well as a real-world human activity dataset.

## 2 Related Work

Controlling robots from natural language has a rich history [74, 66, 37], primarily because it provides a natural means for humans to interact with robots [5, 30]. Recent work on this topic can be categorized as semantic parsing [39, 30, 69, 67, 55, 40, 68], planning [60, 22, 23, 24, 61, 35, 34, 28], task specification [64, 19, 58, 12], reward learning [46, 56, 7], learning low-level policies [46, 2, 57, 56, 7], imitation learning [25, 38, 58, 64] and reinforcement learning [26, 18, 10, 45, 1]. However, these approaches fall in one of two categories: generating open-loop action sequences, or learning closed-loop, but short-horizon, policies. In contrast, we look to generate *task code*, which is promising in solving long-horizon tasks with control flows. The generated code also presents an interpretable way to control robots while maintaining the ability to generalize by composing existing functions.

Synthesizing code from language too has a rich history. Machine learning approaches offer powerful techniques for program synthesis [49, 4, 14]. More recently, these tasks are extended to general-purpose programming languages [79, 78, 8], and program specifications are fully described in natural English text [21, 3, 51]. Pretrained language models have shown great promise in code generation by exploiting the contextual representations learned from massive data of codes and texts [16, 11, 72, 71, 9, 47]. These models can be trained on non-MLE objectives [20], such as RL [32] to pass unit tests. Alternatively, models can also be improved through prompting methods

such as Least-to-Most [82], Think-Step-by-Step [29] or Chain-of-Thought [73], which we leverage in our approach. Closest to our approach is CodeAsPolicies [33], that translates language to robot code. We build on it to address the more challenging problem of going from few demonstrations to code.

We broadly view our approach as inverting the output of code. This is closely related to *inverse graphics*, where the goal is to generate code that has produced a given image or 3D model [76, 36, 15, 70, 17]. Similar to our approach [65] trains an LSTM model that takes as input multiple demonstrations, compresses it to a latent vector and decodes it to domain specific code. Instead of training custom models to generate custom code, we leverage pre-trained LLMs that can generalize much more broadly, and generate more complex Python code, even create new functions. Closest to our approach [77] uses pre-trained LLMs to summarize demonstrations as rules in *one step* before generating code that creates a sequence of pick-then-place and pick-then-toss actions. However, they show results on short-horizon tasks with small number of primitive actions. We look at more complex, long-horizon robot tasks, where demonstrations cannot be summarized in one step. We draw inspiration from [75, 50, 43] to recursively summarize demonstrations until they are compact.

## 3  Problem Formulation

We look at the concrete setting where a robot must perform a set of everyday tasks in a home, like cooking recipes or washing dishes, although our approach can be easily extended to other settings. We formalize such tasks as a Markov Decision Process (MDP), $< \mathcal{S}, \mathcal{A}, \mathcal{T}, \mathcal{R} >$, defined below:

- **State** ($s \in \mathcal{S}$) is the set of all objects in the scene and their propositions, e.g. `open(obj)` ("obj is open"), `on-top(obj1, obj2)` ("obj1 is on top of obj2").
- **Action** ($a \in \mathcal{A}$) is a primitive action, e.g. `pick(obj)` ("pick up obj"), `place(obj, loc)` ("place obj on loc"), `move(loc1, loc2)` ("move from loc1 to loc2").
- **Transition function** ($\mathcal{T}(.|s, a)$) specifies how objects states and agent changes upon executing an action. The transition is stochastic due to hidden states, e.g. `cut('lettuce')` must be called a variable number of times till the state changes to `is_cut('lettuce')`.
- **Reward function** ($r(s, a)$) defines the task, i.e. the subgoals that the robot must visit and constraints that must not be violated.

We assume access to state-based demonstrations because most robotics system have perception modules that can parse raw sensor data into predicate states [42, 27]. We also assume that a system engineer provides a perception library and an action library. The perception library uses sensor observations to maintain a set of state predicates and provides helper functions that use these predicates (e.g. `get_obj_location(obj)`, `is_cooked(obj)`). Meanwhile, the action library defines a set of actions that correspond to low-level policies, similar to [33, 61, 77, 80].

The goal is to learn a policy $\pi_\theta$ that maximizes cumulative reward $J(\pi_\theta) = \mathbb{E}_{\pi_\theta} \left[ \sum_{t=1}^{T} [r(s_t, a_t)] \right]$, $\theta$ being the parameters of the policy. We choose to represent the policy as code $\theta$ for a number of reasons: code is interpretable, composable, and verifiable.

In this setting, the reward function $r(s, a)$ is not explicit, but implicit in the task specification that the user has in mind. Unlike typical Reinforcement Learning (RL), where the reward function is hand designed, it is impractical to expect everyday users to program such reward functions for every new task that they want to teach their robots. Instead, they are likely to communicate tasks through natural means of interaction such as language instructions $l$ (e.g. "Make a burger"). We can either use a model to generate reward $r(s, a)$ from $l$ [31] or directly generate the optimal code $\theta$ [33].

However, language instructions $l$ from everyday users can be challenging to map to precise robot instructions [63, 44, 81]: they may be difficult to ground, may lack specificity, and may not capture users' intrinsic preferences or hidden constraints of the world. For example, the user may forget to specify how they wanted their burger done, what toppings they preferred, etc. Providing such level of detail through language every time is taxing. A more scalable solution is to pair the language instruction $l$ with demonstrations $\mathcal{D} = \{s_1, s_2, \ldots, s_T\}$ of the user doing the task. The state at time-step $t$ only contains the propositions that have changed from $t - 1$ to $t$. Embedded in the states are specific details of how the user wants a task done.

Our goal is to infer the most likely code given both the language and the demonstrations: $\arg\max_\theta P(\theta|l, \mathcal{D})$. For a long-horizon task like cooking, each demonstration can become long. Naively concatenating all demonstrations together to query the LLM can either exhaust the model's

**Algorithm 1** `Demo2Code`: Generating task code from language instructions and demonstrations

---

**Input:** Language instructions `lang`, Demonstrations `demos`
**Output:** Final task code `final_code` that can be executed

```python
def summarize(demos):
    if is_summarized(demos):
        all_demos = "".join(demos)
        return llm(summary_prompt, all_demos)
    else:
        summarized_demos = []
        for demo in demos:
            summarized_demos.append(llm(summary_prompt, demo))
        return summarize(summarized_demos)

def expand_code(code):
    if is_expanded():
        return code
    else:
        expanded_code = code
        for fun in get_undefined_functions(code):
            fun_code = llm(code_prompt, fun)
            expanded_code += expand_code(fun_code)
        return expanded_code

def main():
    spec = summarize(demos)
    high_level_code = llm(code_prompt, lang + spec)
    final_code = expand_code(high_level_code)
```

---

context length or make directly generating the code challenging. We propose an approach that overcomes these challenges.

## 4   Approach

We present a framework, `Demo2Code`, that takes both language instructions and a set of demonstrations from a user as input to generate robot code. The key insight is that while both input and output can be quite long, they share a *latent, compact specification* of the task that the user had in mind. Specifically, the task specification is a detailed language description of how the task should be completed. Since our goal is to generate code, its structure is similar to a pseudocode that specifies the desired code behavior. The specification is latent because we assume that users do not explicitly define the task specification and do not provide detailed language instructions on how to complete the task.

Our approach constructs an extended chain-of-thought that connects the users' demonstrations to a latent task specification, and then connects the generated specification to the code. Each step is small and easy for the LLM to process. Algorithm 1 describes our overall approach, which contains two main stages. Stage 1 recursively summarizes demonstrations down to a specification. The specification and language instruction is then converted to a high-level code with new, undefined functions. Stage 2 recursively expands this code, defining more functions along the way.

### 4.1   Stage 1: Recursively Summarize Demonstrations to Specifications

Figure 2: Recursive summarization of input demonstrations to a compact specification. (Stage 1)

The goal of this stage is to summarize the set of demonstrations provided by the user into a compact specification (refer to `summarize(demos)` in Algorithm 1). Each demonstration is first independently summarized until the LLM determines that the demonstration can no longer be compressed, then the summaries are concatenated and summarized together. Fig. 2 shows example interim outputs during this stage. First, states in each demonstration get summarized into low-level actions (e.g. "patty6 is cooked" is summarized as "robot1 cooked patty6.") Then, low-level actions across time-steps are summarized into high-level subtasks, such as stacking, cutting, (e.g. "At state 3-8, the high level subtask is cook..."). The LLM determines to stop recursively summarizing after the entire demonstration gets converted to high-level subtasks, but this can have a different stopping condition (e.g. setting a maximum step) for task settings different than Fig. 2's. Next, these demonstrations' summaries are concatenated together for the LLM to generate the task specification. The LLM is prompted to first perform some intermediate reasoning to extract details on personal preferences, possible control loops, etc. For instance, the LLM aggregates high-level subtasks into an ordered list, which empirically helps the model to identify repeated subsets in that list and reason about control loops. An example final specification is shown in Fig. 2, which restates the language instruction first, then states "Specifically: .." followed by a more detailed instruction of the task.

## 4.2 Stage 2: Recursively Expand Specification to Task Code

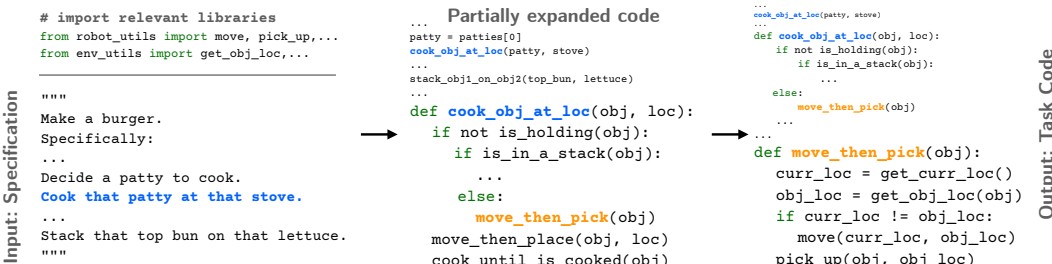

Figure 3: Recursive expansion of the high-level code generated from the specification, where new functions are defined by the LLM along the way. (Stage 2)

The goal of this stage is to use the generated specification from stage 1 to define all the code required for the task (see `expand_code(code)` in Algorithm 1). The LLM is prompted to first generate high-level code that calls functions that may be undefined. Subsequently, each of these undefined functions in the code is recursively expanded. Fig. 3 shows an example process of the code generation pipeline. The input is the specification formatted as a docstring. We import custom robot perception and control libraries for the LLM and also show examples of how to use such libraries in the prompt. The LLM first generates a high-level code, *that can contain new functions*, e.g. `cook_obj_at_loc`, that it has not seen in the prompt or import statements before. It expands this code by calling additional functions (e.g. `move_then_pick`), which it defines in the next recursive step. The LLM eventually reaches the base case when it only uses imported APIs to define a function (e.g. `move_then_pick`).

## 5 Experiments

### 5.1 Experimental Setup

**Baselines and Metrics** We compare our approach **Demo2Code** against prior work, CodeAsPolicies [33], which we call **Lang2Code**. This generates code only from language instruction. We also compare against **DemoNoLang2Code** that generates code from demonstrations without a language instruction, which is achieved by modifying the LLM prompts to redact the language. Finally, we also compare to an oracle **Spec2Code**, which generates task code from detailed specifications on how to complete a task. We use `gpt-3.5-turbo-16k` for all experiments with temperature 0.

We evaluate the different methods across three metrics. **Execution Success Rate** is the average 0/1 success of whether the generated code can run without throwing an error. It is independent from whether the goal was actually accomplished. **Unit Test Pass Rate** is based on checking whether all subgoals are achieved and all constraints are satisfied. The unit test module checks by examining the state transitions created from executing the generated code. **Code BLEU score** is the BLEU

Table 1: Results for Tabletop Manipulation simulator. The tasks are categories into 3 clusters: Specificity ("Specific"), Hidden World Constraint ("Hidden"), and Personal Preference ("Pref").

| | Task | Lang2Code[33] | | | DemoNoLang2Code | | | Demo2Code(*ours*) | | |
|---|---|---|---|---|---|---|---|---|---|---|
| | | Exec. | Pass. | BLEU. | Exec. | Pass. | BLEU. | Exec. | Pass. | BLEU. |
| Specific | Place A next to B | 1.00 | 0.33 | 0.73 | 0.90 | 0.80 | 0.82 | 1.00 | 1.00 | 0.98 |
| | Place A at a corner of the table | 1.00 | 0.30 | 0.08 | 1.00 | 1.00 | 0.85 | 1.00 | 1.00 | 1.00 |
| | Place A at an edge of the table | 1.00 | 0.20 | 0.59 | 1.00 | 0.95 | 0.84 | 1.00 | 1.00 | 0.84 |
| Hidden | Place A on top of B | 1.00 | 0.03 | 0.23 | 0.60 | 0.70 | 0.56 | 0.90 | 0.40 | 0.40 |
| | Stack all blocks | 1.00 | 0.20 | 0.87 | 1.00 | 0.70 | 0.50 | 1.00 | 0.70 | 0.50 |
| | Stack all cylinders | 1.00 | 0.37 | 0.89 | 1.00 | 0.83 | 0.49 | 1.00 | 1.00 | 1.00 |
| Prefs | Stack all blocks into one stack | 1.00 | 0.13 | 0.07 | 1.00 | 0.67 | 0.52 | 1.00 | 0.87 | 0.71 |
| | Stack all cylinders into one stack | 1.00 | 0.13 | 0.00 | 0.90 | 0.77 | 0.19 | 1.00 | 0.90 | 0.58 |
| | Stack all objects into two stacks | 1.00 | 0.00 | 0.00 | 1.00 | 0.90 | 0.68 | 1.00 | 0.90 | 0.65 |
| | Overall | 1.00 | 0.19 | 0.39 | 0.93 | 0.81 | 0.60 | 0.99 | 0.88 | 0.77 |

score [48] between a method's generated code and the oracle `Spec2Code`'s generated code. We tokenize each code by the spaces, quotations, and new lines.

**Tabletop Manipulation Simulator [80, 23]**   We build upon a physics simulator from [80, 23], which simulates a robot arm manipulating blocks and cylinders in different configurations. The task objectives are to place objects at specific locations or stack objects on top of each other. The LLM has access to action primitives (e.g. pick and place) and perception modules (e.g. to get all the objects in the scene). We create a range of tasks that vary in complexity and specificity, use the oracle `Spec2Code` to generate reference code, and execute that code to get demonstrations for other methods. For each task, we test the generated code for 10 random initial conditions of objects.

Table 2: Results for the `Robotouille` simulator. The training tasks in the prompt are at the top of the table and highlighted in gray. All tasks are ordered by the horizon length (the number of states). Below the table shows four `Robotouille` tasks where the environments gradually become more complex.

| Task | Lang2Code[33] | | | DemoNoLang2Code | | | Demo2Code(*ours*) | | | Horizon Length |
|---|---|---|---|---|---|---|---|---|---|---|
| | Exec. | Pass. | BLEU. | Exec. | Pass. | BLEU. | Exec. | Pass. | BLEU. | |
| Cook a patty | 1.00 | 1.00 | 0.90 | 1.00 | 1.00 | 0.90 | 1.00 | 1.00 | 0.90 | 8.0 |
| Cook two patties | 0.80 | 0.80 | 0.92 | 0.80 | 0.80 | 0.92 | 0.80 | 0.80 | 0.92 | 16.0 |
| Stack a top bun on top of a cut lettuce on top of a bottom bun | 1.00 | 1.00 | 0.70 | 0.00 | 0.00 | 0.75 | 1.00 | 1.00 | 0.60 | 14.0 |
| Cut a lettuce | 1.00 | 1.00 | 0.87 | 0.00 | 0.00 | 0.76 | 1.00 | 1.00 | 0.87 | 7.0 |
| Cut two lettuces | 0.80 | 0.80 | 0.92 | 0.00 | 0.00 | 0.72 | 0.80 | 0.80 | 0.92 | 14.0 |
| Cook first then cut | 1.00 | 1.00 | 0.88 | 1.00 | 1.00 | 0.88 | 1.00 | 1.00 | 0.88 | 14.0 |
| Cut first then cook | 1.00 | 1.00 | 0.88 | 0.00 | 0.00 | 0.82 | 1.00 | 1.00 | 0.88 | 15.0 |
| Assemble two burgers one by one | 0.00 | 0.00 | 0.34 | 1.00 | 1.00 | 0.77 | 1.00 | 1.00 | 0.76 | 15.0 |
| Assemble two burgers in parallel | 0.00 | 0.00 | 0.25 | 1.00 | 1.00 | 0.51 | 0.00 | 0.00 | 0.71 | 15.0 |
| Make a cheese burger | 1.00 | 0.00 | 0.24 | 1.00 | 1.00 | 0.69 | 1.00 | 1.00 | 0.69 | 18.0 |
| Make a chicken burger | 0.00 | 0.00 | 0.57 | 0.00 | 0.00 | 0.64 | 0.90 | 0.90 | 0.69 | 25.0 |
| Make a burger stacking lettuce atop patty immediately | 1.00 | 0.00 | 0.74 | 0.20 | 0.00 | 0.71 | 0.00 | 0.00 | 0.71 | 24.5 |
| Make a burger stacking patty atop lettuce immediately | 0.00 | 0.00 | 0.74 | 0.20 | 0.00 | 0.71 | 1.00 | 1.00 | 0.74 | 25.0 |
| Make a burger stacking lettuce atop patty after preparation | 1.00 | 0.00 | 0.67 | 0.10 | 0.00 | 0.65 | 0.00 | 0.00 | 0.66 | 26.5 |
| Make a burger stacking patty atop lettuce after preparation | 1.00 | 0.00 | 0.67 | 0.00 | 0.00 | 0.53 | 1.00 | 0.00 | 0.69 | 27.0 |
| Make a lettuce tomato burger | 0.00 | 0.00 | 0.13 | 1.00 | 1.00 | 0.85 | 1.00 | 0.00 | 0.66 | 34.0 |
| Make two cheese burgers | 0.00 | 0.00 | 0.63 | 1.00 | 1.00 | 0.68 | 1.00 | 1.00 | 0.68 | 38.0 |
| Make two chicken burgers | 0.00 | 0.00 | 0.52 | 0.00 | 0.00 | 0.68 | 1.00 | 0.00 | 0.56 | 50.0 |
| Make two burgers stacking lettuce atop patty immediately | 0.80 | 0.00 | 0.66 | 0.80 | 1.00 | 0.69 | 0.00 | 0.00 | 0.66 | 50.0 |
| Make two burgers stacking patty atop lettuce immediately | 0.80 | 0.00 | 0.67 | 1.00 | 0.00 | 0.48 | 1.00 | 1.00 | 0.73 | 50.0 |
| Make two burgers stacking lettuce atop patty after preparation | 0.80 | 0.00 | 0.66 | 0.60 | 0.00 | 0.66 | 0.80 | 0.00 | 0.67 | 54.0 |
| Make two burgers stacking patty atop lettuce after preparation | 0.80 | 0.00 | 0.67 | 0.50 | 0.00 | 0.71 | 0.80 | 0.00 | 0.68 | 54.0 |
| Make two lettuce tomato burgers | 1.00 | 0.00 | 0.55 | 0.00 | 0.00 | 0.70 | 1.00 | 1.00 | 0.84 | 70.0 |
| Overall | 0.64 | 0.29 | 0.64 | 0.49 | 0.38 | 0.71 | 0.79 | 0.59 | 0.74 | 28.8 |

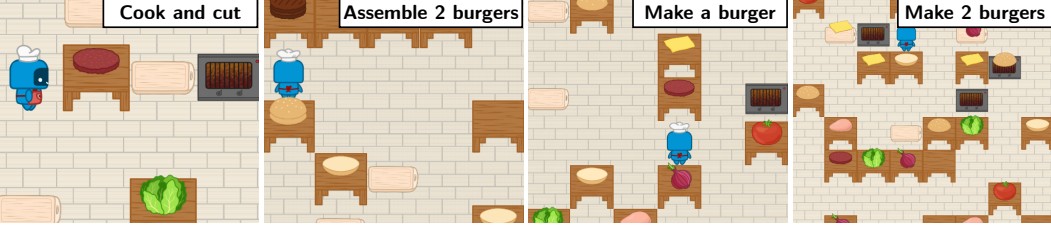

| Cook and cut | Assemble 2 burgers | Make a burger | Make 2 burgers |

Table 3: Results for EPIC-Kitchens dataset on 7 different user demonstrations of dish-washing (length of demonstration in parentheses). The unit test pass rate is evaluated by a human annotator, and BLEU score is calculated between each method's code and the human annotator's reference code.

| | P4-101 (7) | | P7-04 (17) | | P7-10 (6) | | P22-05 (28) | | P22-07 (30) | | P30-07 (11) | | P30-08 (16) | |
|---|---|---|---|---|---|---|---|---|---|---|---|---|---|---|
| | Pass. | BLEU. | Pass. | BLEU. | Pass. | BLEU. | Pass. | BLEU. | Pass. | BLEU. | Pass. | BLEU. | Pass. | BLEU. |
| Lang2Code[33] | 1.00 | 0.58 | 0.00 | 0.12 | 0.00 | 0.84 | 0.00 | 0.48 | 0.00 | 0.37 | 1.00 | 0.84 | 0.00 | 0.66 |
| DemoNoLang2Code | 0.00 | 0.00 | 0.00 | 0.00 | 1.00 | 0.00 | 0.00 | 0.37 | 0.00 | 0.51 | 1.00 | 0.57 | 0.00 | 0.00 |
| Demo2Code | 1.00 | 0.33 | 0.00 | 0.19 | 1.00 | 0.63 | 1.00 | 0.43 | 1.00 | 0.66 | 1.00 | 0.58 | 0.00 | 0.24 |

**Cooking Task Simulator: Robotouille** [1] We introduce a novel, open-source simulator to simulate complex, long-horizon cooking tasks for a robot, e.g. making a burger by cutting lettuces and cooking patties. Unlike existing simulators that focus on simulating physics or sensors, `Robotouille` focuses on high level task planning and abstracts away other details. We build on a standard backend, PDDLGym [59], with a user-friendly game as the front end to easily collect demonstrations. For the experiment, we create a set of tasks, where each is associated with a set of preferences (e.g. what a user wants in the burger, how the user wants the burger cooked). For each task and each associated preference, we procedurally generate 10 scenarios.

**EPIC-Kitchens Dataset [13]** EPIC-Kitchens is a real-world, egocentric video dataset of users doing tasks in their kitchen. We use this to test if `Demo2Code` can infer users' preferences from real videos, with the hopes of eventually applying our approach to teach a real robot personalized tasks. We focus on dish washing as we found preferences in it easy to qualify. While each video has annotations of low-level actions, these labels are insufficient for describing the tasks. Hence, we choose 7 videos of 4 humans washing dishes and annotate each demonstration with dense state information. We compare the code generated by `Lang2Code`, `DemoNoLang2Code` and `Demo2Code` on whether it satisfies the annotated preference and how well it matches against the reference code.

## 5.2 Results and Analysis

Overall, `Demo2Code` has the closest performance to the oracle (`Spec2Code`). Specifically, our approach has the highest unit test pass rates in all three benchmarks, as well as the highest execution success in `Robotouille` (table 2) and EPIC-Kitchens (table 3). Meanwhile, `Lang2Code` [33] has a higher overall execution success than `Demo2Code` for the Tabletop simulator (table 1). However, `Lang2Code` has the lowest unit test pass rate among all baselines because it cannot fully extract users' specifications without demonstrations. `DemoNoLang2Code` has a relatively higher pass rate, but it sacrifices execution success because it is difficult to output plausible code without context from language. We provide prompts, detailed results and ablations in the Appendix.[2] We now ask a series of questions of the results to characterize the performance difference between the approaches.

**How well does `Demo2Code` generalize to unseen objects and tasks?** `Demo2Code` exhibits its generalization ability in three axes. First, `Demo2Code` generalizes and solves unseen tasks with longer horizons and more predicates compared to examples in the prompt at train time. For `Robotouille`, table 2 shows the average horizon length for each training task (highlighted in gray) and testing task. Overall, the training tasks have an average of 12.7 states compared the testing tasks (31.3 states). Compared to the baselines, `Demo2Code` performs the best for long burger-making tasks (an average of 32 states) even though the prompt does not show this type of task. Second, `Demo2Code` uses control flow, defines hierarchical code, and composes multiple subtasks together to solve these long-horizon tasks. The appendix details the average number of loops, conditionals, and helper functions that `Demo2Code` generates for tabletop simulator (in section 8.3) and `Robotouille` (in section 9.3). Notably, `Demo2Code` generates code that uses a for-loop for the longest task (making two lettuce tomato burgers with 70 states), which requires generalizing to unseen subtasks (e.g. cutting tomatoes) and composing 7 distinct subtasks. Third, `Demo2Code` solves tasks that contain unseen objects or a different number of objects compared to the training tasks in the prompt. For `Robotouille`, the prompt only contains examples of preparing burger patties and lettuce, but `Demo2Code` still has the highest unit test pass rate for making burgers with unseen ingredients: cheese, chicken, and

---

[1]Codebase and usage guide for `Robotouille` is available here: https://github.com/portal-cornell/robotouille
[2]Codebase is available here: https://github.com/portal-cornell/demo2code

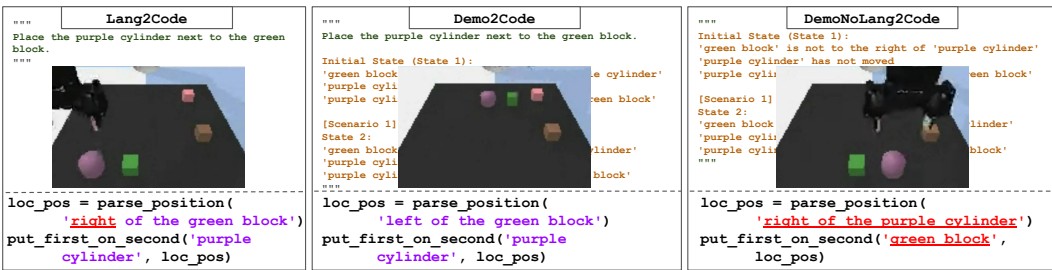

Figure 4: `Demo2Code` successfully extracts specificity in tabletop tasks. `Lang2Code` lacks demonstrations and randomly chooses a spatial location while DemoNoLang2Code lacks context in what the demonstrations are for.

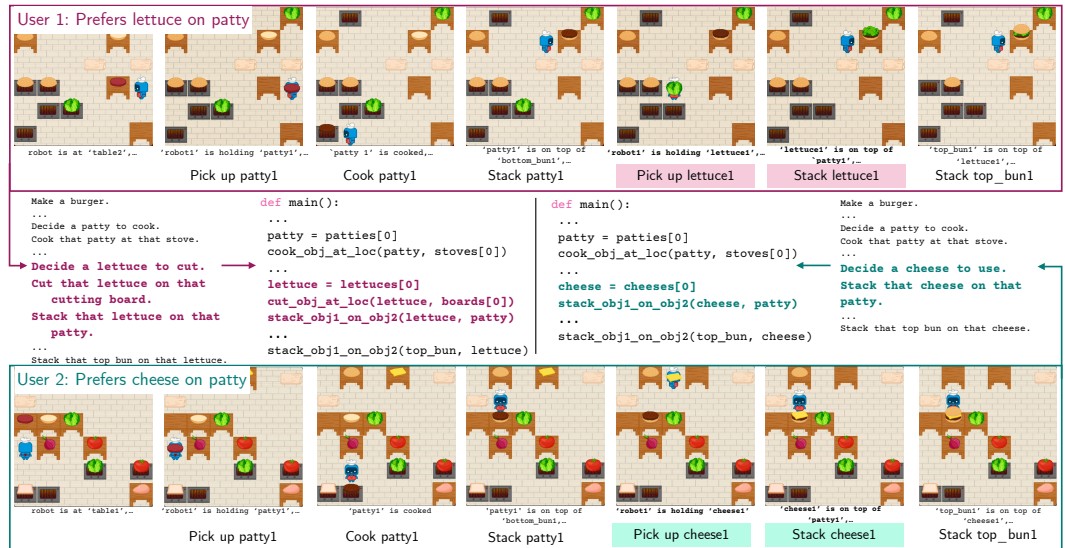

Figure 5: `Demo2Code` summarizes demonstrations and identify different users' preferences on how to make a burger (e.g. whether to include lettuce or cheese) in `Robotouille` simulator. Then, it generates personalized burger cooking code to use the user's preferred ingredients.

tomatoes. Similarly, for tabletop (table 1), although the prompt only contains block-stacking tasks, our approach maintains high performance for cylinder-stacking tasks.

**Is `Demo2Code` able to ground its tasks using demonstrations?**  Language instructions sometimes cannot ground the tasks with specific execution details. Since demonstrations provide richer information about the task and the world, we evaluate whether `Demo2Code` can utilize them to extract details. Tasks under the "Specific" cluster in Table 1 show cases when the LLM needs to use demonstrations to ground the desired goal. Fig. 4 illustrates that although the language instruction ("Place the purple cylinder next to the green block") does not ground the desired spatial relationship between the two objects, our approach is able to infer the desired specification ("to the left"). In contrast, `Lang2Code` can only randomly guess a spatial relationship, while `DemoNoLang2Code` can determine the relative position, but it moved the green block because it does not have language instruction to ground the overall task. Similarly, tasks under the "Hidden" cluster in Table 1 show how `Demo2Code` outperforms others in inferring hidden constraints (e.g the maximum height of a stack) to ground its tasks.

**Is `Demo2Code` able to capture individual user preference?**  As a pipeline for users to teach robots personalized tasks, `Demo2Code` is evaluated on its ability to extract a user's preference. Table 3 shows that our approach performs better than `Lang2Code` in generating code that matches each EPIC-Kitchens user's dish washing preference, without overfitting to the demonstration like in `DemoNoLang2Code`. Because we do not have a simulator that completely matches the dataset, human annotators have to manually inspect the code. The code passes the inspection if it has correct syntax, does not violate any physical constraints (e.g. does not rinse a dish without turning on the tap), and matches the user's dish-washing preference. Qualitatively, Fig. 6 shows that our approach is able to extract the specification and generate the correct code respectively for user 22, who prefers to soap

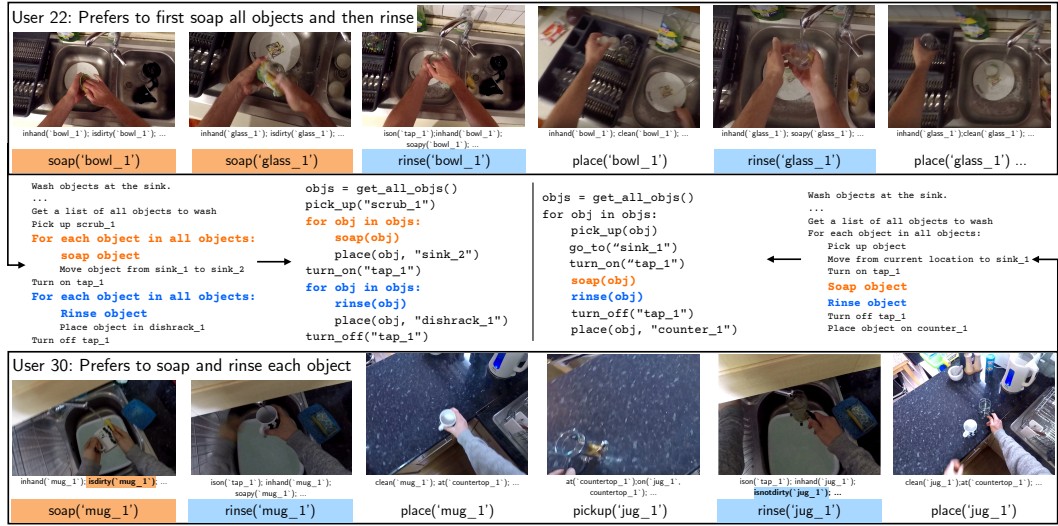

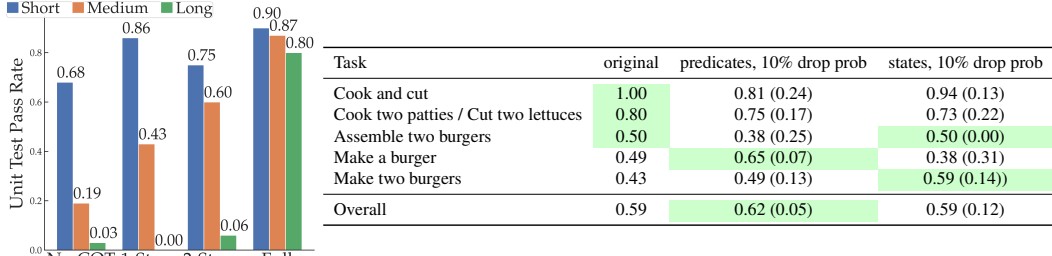

Figure 6: `Demo2Code` summarizes different styles of users washing dishes from demonstration (how to soap and rinse objects) in EPIC-Kitchens, and generates personalized dish washing code.

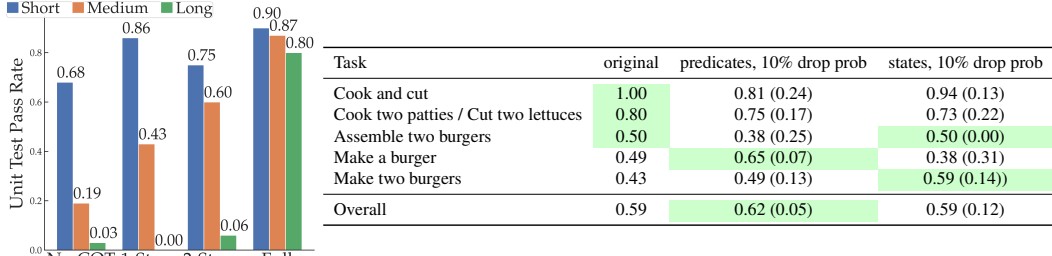

| Task | original | predicates, 10% drop prob | states, 10% drop prob |
|---|---|---|---|
| Cook and cut | 1.00 | 0.81 (0.24) | 0.94 (0.13) |
| Cook two patties / Cut two lettuces | 0.80 | 0.75 (0.17) | 0.73 (0.22) |
| Assemble two burgers | 0.50 | 0.38 (0.25) | 0.50 (0.00) |
| Make a burger | 0.49 | 0.65 (0.07) | 0.38 (0.31) |
| Make two burgers | 0.43 | 0.49 (0.13) | 0.59 (0.14)) |
| Overall | 0.59 | 0.62 (0.05) | 0.59 (0.12) |

Figure 7: (Left) Unit test result for ablating different degrees of chain-of-thought across tasks with short, medium, long horizon. (Right) `Demo2Code`'s unit test result for `Robotouille` demonstrations with different level of noises: (1) each predicate has 10% chance of being dropped, and (2) each state has 10% chance of being completely dropped. We ran the experiment 4 times and report the average and variance.

all objects before rinsing them, and user 30, who prefers to soap then rinse each object individually. Similarly, Fig. 5 provides an example of how `Demo2Code` is able to identify a user's preference of using cheese vs lettuce even when the language instruction is just "make a burger." Quantitatively, Table 2 shows more examples of our approach identifying a user's preference in cooking order, ingredient choice, etc, while Table 1 also shows our approach performing well in tabletop tasks.

**How does chain-of-thought compare to directly generating code from demonstrations?** To evaluate the importance of our extended chain-of-thought pipeline, we conduct ablation by varying the length of the chain on three clusters of tasks: short-horizon (around 2 states), medium-horizon (5-10 states), and long-horizon ($\geq 15$ states). We compare the unit test pass rate on four different chain lengths, ranging from **No chain-of-thought** (the shortest), which directly generates code from demonstrations, to **Full** (the longest), which represents our approach `Demo2Code`. The left bar plot in Fig. 7 shows that directly generating code from demonstrations is not effective, and the LLM performs better as the length of the chain increases. The chain length also has a larger effect on tasks with longer horizons. For short-horizon tasks, the LLM can easily process the short demonstrations and achieve high performances by just using **1-step**. Meanwhile, the stark difference between **2-steps** and **Full**'s results on long-horizon tasks emphasizes the importance of taking as many small steps as the LLM needs in summarizing long demonstrations so that it will not lose key information.

**How do noisy demonstrations affect `Demo2Code`'s performance?** We study how `Demo2Code` performs (1) when each predicate has a 10% chance to be removed from the demonstrations, and (2) when each state has a 10% chance to be completely removed. Fig. 7's table shows that `Demo2Code`'s overall performance does not degrade even though demonstrations are missing information. While

removing predicates or states worsen `Demo2Code`'s performance for shorter tasks (e.g. cook and cut), they surprisingly increase the performance for longer tasks. Removing any predicates can omit essential information in shorter tasks' demonstrations. Meanwhile, for longer tasks, the removed predicates are less likely to be key information, while reducing the length of demonstrations. Similarly, for the longest tasks to make two burgers, one burger's missing predicates or states can be explained by the other burger's demonstration. In section 11, we show a specific example of this phenomenon. We also study the effect of adding additional predicates to demonstrations, which has degraded `Demo2Code`'s performance from satisfying 5 users' preferences to 2 users' in EPIC-Kitchens.

## 6 Discussion

In this paper, we look at the problem of generating robot task code from a combination of language instructions and demonstrations. We propose `Demo2Code` that first recursively summarizes demonstrations into a latent, compact specification then recursively expands code generated from that specification to a fully defined robot task code. We evaluate our approach against prior state-of-the-art [33] that generates code only from language instructions, across 3 distinct benchmarks: a tabletop manipulation benchmark, a novel cooking game `Robotouille`, and annotated data from EPIC-Kitchens, a real-world human activity dataset. We analyze various capabilities of `Demo2Code`, such as grounding language instructions, generalizing across tasks, and capturing user preferences.

`Demo2Code` **can generalize across complex, long-horizon tasks.** Even though `Demo2Code` was shown only short-horizon tasks, it's able to generalize to complex, long demonstrations. Recursive summarization compresses long chains of demonstrations and recursive expansion generates complex, multi-layered code.

`Demo2Code` **leverages demonstrations to ground ambiguous language instructions and infer hidden preferences and constraints.** The latent specification explicitly searches for missing details in the demonstrations, ensuring they do not get explained away and are captured explicitly in the specification.

`Demo2Code` **strongly leverages chain-of-thought.** Given the complex mapping between demonstrations and code, chain-of-thought plays a critical role in breaking down computation into small manageable steps during summarization, specification generation and code expansion.

In future directions, we are looking to close the loop on code generation to learn from failures, integrate with a real home robot system and run user studies with `Robotouille`.

## 7 Limitations

`Demo2Code` is limited by the capability of LLMs. Recursive summarization assumes that once all the demonstrations are sufficiently summarized, they can be concatenated to generate a specification. However, in extremely long horizon tasks (e.g. making burgers for an entire day), it is possible that the combination of all the sufficiently summarized demonstrations can still exceed the maximum context length. A future work direction is to prompt the LLM with chunks of the concatenated demonstrations and incrementally improve the specifications based on each new chunk. In recursive expansion, our approach assumes that all low-level action primitives are provided. `Demo2Code` currently cannot automatically update its prompt to include any new action. Another direction is to automatically build the low-level skill libraries by learning low-level policy via imitation learning and iteratively improve the code-generation prompt over time. Finally, since LLMs are not completely reliable and can hallucinate facts, it is important to close the loop by providing feedback to the LLM when they fail. One solution [62, 52] is to incorporate feedback in the query and reprompt the language model. Doing this in a self-supervised manner with a verification system remains an open challenge.

In addition, the evaluation approach for `Demo2Code` or other planners that generate code [33, 61, 77] is different from the one for classical planners [53, 54]. Planners that generate code measure a task's complexity by the horizon length, the number of control flows, whether that task is in the training dataset, etc. Meanwhile, many classical planners use domain specific languages such as Linear Temporal Logic (LTL) to specify tasks [41], which leads to categorizing tasks and measuring the task complexity based on LTL. Future work needs to resolve this mismatch in evaluation standards.

## Acknowledgements

We sincerely thank Nicole Thean for creating our art assets for `Robotouille`. This work was supported in part by the National Science Foundation FRR (#2327973).

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

# Appendix

## Table of Contents

# 8 Tabletop Manipulation Simulator Pipeline

## 8.1 Pipeline Overview

The tabletop manipulation simulator contains simple tasks. Consequently, the demonstrations do not have too many states ($\leq 8$ states) and the code is not complex. Thus, `Demo2Code`'s prompt for this domain does not need a long extended chain-of-thought. In stage 1 recursive summarization, the LLM just needs to summarize each states into a sentences that describes the low-level action (e.g. move, pick, place, etc.) In stage 2 recursive expansion, because the code is simple, the LLM can directly use all the low-level actions that are provided to output the task code given a specification.

The prompt demonstrating this pipeline is listed at the end of the appendix in section 18.1.

## 8.2 Experiment Setup

In the paper, we categorize the tabletop tasks into three clusters. For each cluster, we list all the tasks and their variants of possible requirements below. The tasks that are used in the prompt are bolded.

- Specificity
  - Place A next to B
    * **No hidden specificity: A can be placed in any relative position next to B**
    * **A must be to the left of B**
    * A must be to the right of B
    * A must be behind B
    * A must be in front of B
  - Place A at a corner of the table
    * **No hidden specificity: A can be placed at any corner.**
    * A must be at the top left corner
    * A must be at the top right corner
    * A must be at the bottom left corner
    * A must be at the bottom right corner
  - Place A at an edge of the table
    * No hidden specificity: A can be placed at any corner.
    * A must be at the top edge
    * A must be at the bottom edge
    * A must be at the left edge
    * A must be at the right edge
- Hidden Constraint
  - Place A on top of B
    * **No hidden constraint: A can be directly placed on top of B in one step**
    * There is 1 additional object on top of A, so that needs to be removed before placing A on top of B.
    * There are 2 additional objects on top of A.
    * **There are 3 additional objects on top of A.**
  - Stack all blocks
    * **No hidden constraint: All blocks can be stacked into one stack**
    * Each stack can be at most 2 blocks high

* **Each stack can be at most 3 blocks high**
        * Each stack can be at most 4 blocks high
    – Stack all cylinders (Same set of hidden constraints as "stack all blocks." None of the examples appears in the prompt.)
  • Personal Preference
    – Stack all blocks into one stack
        * 2 blocks must be stacked in a certain order, and the rest can be unordered
        * **3 blocks must be stacked in a certain order**
        * All blocks must be stacked in a certain order
    – Stack all cylinders into one stack (Same set of hidden constraints as "stack all blocks into one stack" None of the examples appears in the prompt.)
    – Stack all objects
        * **No hidden preference: The objects do not need to be stacked in to different stacks based on their type**
        * All the blocks should be stacked in one stack, and all the cylinders should be stacked in another stack

### 8.2.1 Provided Low-Level APIs

We have provided the following APIs for the perception library and low-level skill library:

  • Perception Library
    – `get_obj_names()`: return a list of objects in the environment
    – `get_all_obj_names_that_match_type(type_name, objects_list)`: return a list of objects in the environment that match the `type_name`.
    – `determine_final_stacking_order(objects_to_enforce_order, objects_without_order)`: return a sorted list of objects to stack.
  • Low-level Skill Library
    – `put_first_on_second(arg1, arg2)`: pick up an object (`arg1`) and put it at `arg2`. If `arg2` is an object, `arg1` will be on top of `arg2`. If `arg2` is 'table', `arg1` will be somewhere random on the table. If `arg2` is a list, `arg1` will be placed at location [x, y].
    – `stack_without_height_limit(objects_to_stack)`: stack the list of `objects_to_stack` into one stack without considering height limit.
    – `stack_with_height_limit(objects_to_stack, height_limit)`: stack the list of `objects_to_stack` into potentially multiple stacks, and each stack has a maximum height based on `height_limit`.

### 8.3 Characterize Tabletop Tasks' Complexity

In table 4, we characterize the complexity of the tasks in terms of the demonstrations' length, the code's length, and the expected code's complexity (i.e. how many loops/conditionals/functions are needed to solve this task).

Table 4: For tabletop tasks, we group them by cluster and report: 1. number of states in demonstrations (range and average) 2. number of predicates in demonstrations (range and average) 3. number of lines in the oracle `Spec2Code`'s generated code (range and average) 4. average number of loops 5. average number of conditionals 6. average number of functions

| Task | Input Demo Length | | Code Length | # of loops | # of conditionals | # of functions |
| | # of states | # of predicates | | | | |
| --- | --- | --- | --- | --- | --- | --- |
| Place A next to B | 1-1 (1.00) | 2-5 (3.53) | 3-7 (3.38) | 0.00 | 0.02 | 1.00 |
| Place A at corner/edge | 1-1 (1.00) | 1-5 (2.09) | 2-4 (3.03) | 0.00 | 0.00 | 1.00 |
| Place A on top of B | 1.0-4.0 (2.50) | 3-19 (9.40) | 2-6 (3.65) | 0.10 | 0.00 | 1.00 |
| Stack all blocks/cylinders | 2-7 (4.43) | 4-33 (14.09) | 3-15 (4.44) | 0.24 | 0.06 | 1.00 |
| Stack all blocks/cylinders into one stack | 3.5-4 (3.98) | 12-23 (14.77) | 12-12 (12) | 1.00 | 1.00 | 1.00 |
| Stack all objects into two stacks | 6-8 (6.95) | 16-42 (23.90) | 7-25 (8.1) | 0.05 | 0.20 | 1.00 |

# 9 Robotouille Simulator Pipeline

## 9.1 Overview

### 9.1.1 Simulator Description

In `Robotouille`, a robot chef performs cooking tasks in a kitchen environment. The state of the kitchen environment consists of items such as buns, lettuce, and patties located on stations which could be tables, grills, and cutting boards. The actions of the robot consist of moving around from one station to another, picking items from and placing items on stations, stacking items atop and unstacking items from another item, cooking patties on stoves, and cutting lettuce on cutting boards. The state and actions are described through the Planning Domain Description Language (PDDL).

These PDDL files consist of a domain and a problem. The domain file defines an environment; it contains the high-level predicates that describe the state of the world as well as the actions of the world including their preconditions and effects on the world's predicate state. The problem file describes a configuration of an environment; it contains the domain name for the environment, the initial objects and true predicates, and the goal state. These files are used with PDDLGym [59] as a backend to create an OpenAI Gym [6] environment which given a state and action can be stepped through to produce the next state.

There are 4 problem files for different example scenarios including cooking a patty and cutting lettuce, preparing ingredients to make a burger, preparing ingredients to make two burgers, and assembling a burger with pre-prepared ingredients. In a scenario, various different tasks can be carried out, such as varying the order and ingredients for making a burger. These problem files contain the minimum number of objects necessary to complete the scenario for any specified task.

One issue with having pre-defined problem files for each scenario is that the code produced in code generation could be hardcoded for a scenario. This is avoided by procedurally generating the problem files. There are two types of procedural generation: noisy randomization and full randomization. Noisy randomization, which is used for every `Robotouille` experiment in this paper, ensures that the minimum required objects in a problem file appear in an environment in the same grouped arrangement (so an environment with a robot that starts at a table with a patty on it and a cutting board with lettuce on it will maintain those arrangements) but the locations are all randomized and extra stations and items are added (noise). The location of stations and items determines the ID suffix which prevents code generation from always succeeding using hardcoded code.

Full randomization does everything except enforcing that the minimum required objects in a problem file appear in the same grouped arrangement. This would require code that handles edge cases as simple as utilizing ingredients that are already cooked or cut in the environment rather than preparing new ones to more extreme cases such as the kitchen being cluttered with stacked items requiring solving a puzzle to effectively use the kitchen. The simpler case is more appropriate in a real setting and we leave it to future work to remove initial arrangement conditions.

### 9.1.2 Pipeline Overview

In stage 1 recursive summarization, the LLM first recursively summarizes the provided demonstrations, which are represented as state changes since the previous state, until it determines that the trajectories are sufficiently summarized. For this domain, the LLM in general terminates after it summarizes the trajectory into a series of high-level subtasks. Then, `Demo2Code` concatenates all trajectories together before prompting the LLM to reason about invariant in subtask's order before generating the task specification.

In stage 2 recursive expansion, there are 3 steps that occur for `Demo2Code`. First, (1) the task specification is converted directly to code which uses provided helper functions and may use undefined higher-level functions. Second, (2) the undefined higher-level functions are defined potentially including undefined lower-level functions. Finally, (3) the undefined lower-level functions are unambiguously defined.

The prompt demonstrating this pipeline is listed at the end of the appendix in section 18.2.

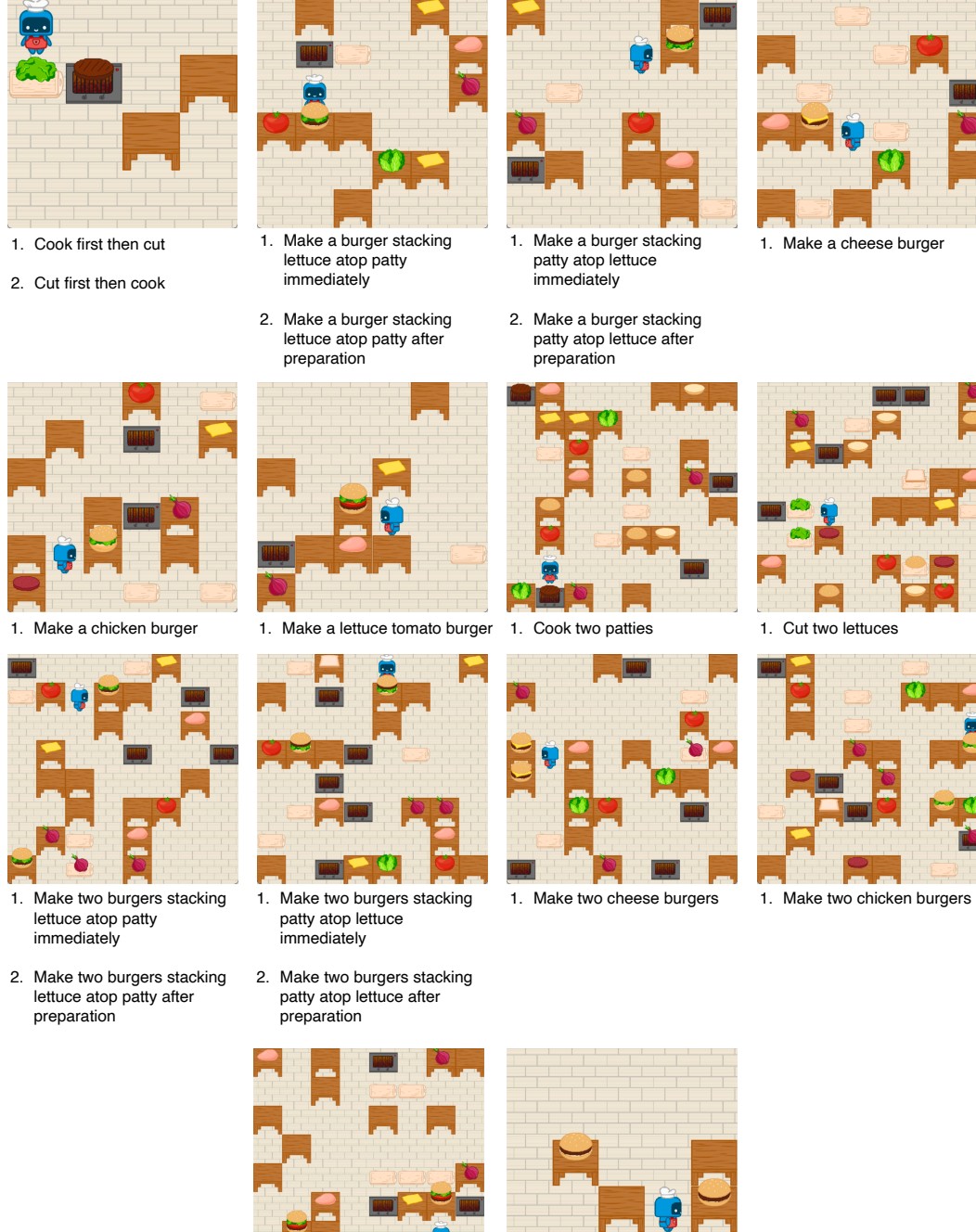

Figure 8: Examples of goal states with the respective tasks underneath.

## 9.2 Experiment Setup

In the paper, we categorized the `Robotouille` simulator into 4 example scenarios. Below are all the scenarios as well as possible tasks, visualized in Fig. 8.

- Cook a patty and cut lettuce
  - Cook a patty
  - Cut a lettuce
  - Cook first then cut
  - Cut first then cook
- Assemble two burgers from prepared ingredients
  - Assemble two burgers one by one
  - Assemble two burgers in parallel
- Make a burger

  - Stack a top bun on top of a cut lettuce on top of a bottom bun
  - Make a burger stacking lettuce atop patty immediately
  - Make a burger stacking patty atop lettuce immediately
  - Make a burger stacking lettuce atop patty after preparation
  - Make a burger stacking patty atop lettuce after preparation
  - Make a cheese burger
  - Make a chicken burger
  - Make a lettuce tomato burger
- Make two burgers
  - Cook two patties
  - Cut two lettuces
  - Make two burgers stacking lettuce atop patty immediately
  - Make two burgers stacking patty atop lettuce immediately
  - Make two burgers stacking lettuce atop patty after preparation
  - Make two burgers stacking patty atop lettuce after preparation
  - Make two cheese burgers
  - Make two chicken burgers
  - Make two lettuce tomato burgers

### 9.2.1 Provided Low-Level APIs

We have provided the following APIs for the perception library and low-level skill library:

- Perception Library
  - `get_all_obj_names_that_match_type(obj_type)`: return a list of string of objects that match the `obj_type`.
  - `get_all_location_names_that_match_type(location_type)`: return a list of string of locations that match the `location_type`.
  - `is_cut(obj)`: return true if `obj` is cut.
  - `is_cooked(obj)`: return true if `obj` is cooked.
  - `is_holding(obj)`: return true if the robot is currently holding `obj`.
  - `is_in_a_stack(obj)`: return true if the `obj` is in a stack.
  - `get_obj_that_is_underneath(obj_at_top)`: return the name of the object that is underneath `obj_at_top`.
  - `get_obj_location(obj)`: return the location that `obj` is currently at.
  - `get_curr_location()`: return the location that the robot is currently at.
- Low-level Skill Library
  - `move(curr_loc, target_loc)`: move from the `curr_loc` to the `target_loc`.

- `pick_up(obj, loc)`: pick up the `obj` from the `loc`.
- `place(obj, loc)`: place the `obj` on the `loc`.
- `cut(obj)`: make progress on cutting the `obj`. Need to call this function multiple times to finish cutting the `obj`.
- `start_cooking(obj)`: start cooking the `obj`. Only need to call this once. The `obj` will take an unknown amount before it is cooked.
- `noop()`: do nothing.
- `stack(obj_to_stack, obj_at_bottom))`: stack `obj_to_stack` on top of `obj_at_bottom`.
- `unstack(obj_to_unstack, obj_at_bottom)`: unstack `obj_to_unstack` from `obj_at_bottom`.

## 9.3 Characterize Robotouille's Tasks' Complexity

In table 5, we characterize the complexity of the tasks in terms of the demonstrations' length, the code's length, and the expected code's complexity (i.e. how many loops/conditionals/functions are needed to solve this task).

Table 5: For Robotouille's tasks, we group them by cluster and report the following: 1. number of states in demonstrations (range and average) 2. number of predicates in demonstrations (range and average) 3. number of lines in the oracle Spec2Code's generated code (range and average) 4. average number of loops 5. average number of conditionals 6. average number of functions

| Task | Input Demo Length | | Code Length | # of loops | # of conditionals | # of functions |
|---|---|---|---|---|---|---|
| | # of states | # of predicates | | | | |
| Cook and cut | 7-15 (10.75) | 8-19 (13.5) | 98-98 (98.0) | 2.00 | 12.0 | 8.00 |
| Cook two patties / cut two lettuces | 14-16 (24.3) | 19-19 (19.0) | 50-54 (52.0) | 1.50 | 6.00 | 6.00 |
| Assemble two burgers | 15-15 (15.0) | 36-36 (36.0) | 58-62 (60.0) | 1.5 | 6.00 | 5.00 |
| Make a burger | 32-55 (42.6) | 26-55 (40.5) | 109-160 (146.3) | 1.86 | 17.1 | 9.86 |
| Make two burgers | 38-70 (52.3) | 68-114 (86.85) | 112-161 (149) | 2.86 | 17.1 | 9.86 |

In addition, to bridge the different evaluation standards between planners that generate code and classical planners, we also characterize the Robotouille's tasks based on [41]'s taxonomy in table 6

Table 6: For each Robotouille task, we check if it contains the specification pattern defined in [41].

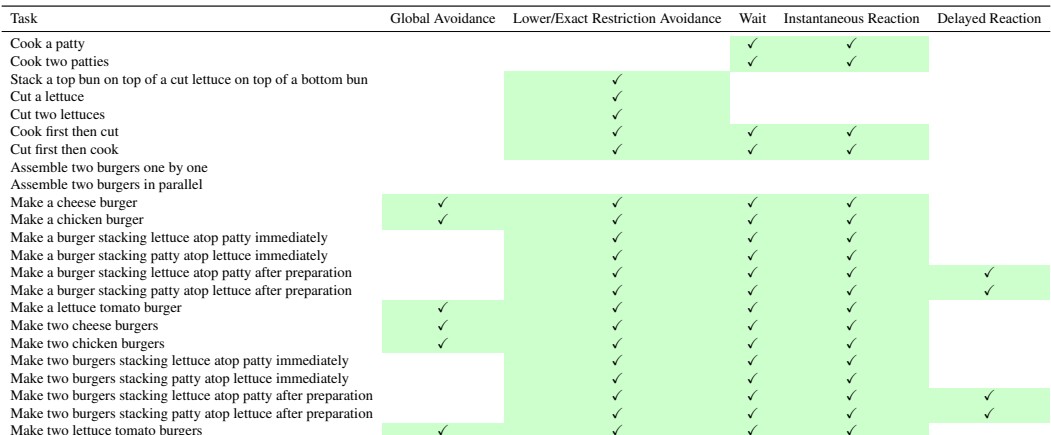

| Task | Global Avoidance | Lower/Exact Restriction Avoidance | Wait | Instantaneous Reaction | Delayed Reaction |
|---|---|---|---|---|---|
| Cook a patty | | | ✓ | ✓ | |
| Cook two patties | | | ✓ | ✓ | |
| Stack a top bun on top of a cut lettuce on top of a bottom bun | | ✓ | | | |
| Cut a lettuce | | ✓ | | | |
| Cut two lettuces | | ✓ | | | |
| Cook first then cut | | ✓ | ✓ | ✓ | |
| Cut first then cook | | ✓ | ✓ | ✓ | |
| Assemble two burgers one by one | | | | | |
| Assemble two burgers in parallel | | | | | |
| Make a cheese burger | ✓ | ✓ | ✓ | ✓ | |
| Make a chicken burger | ✓ | ✓ | ✓ | ✓ | |
| Make a burger stacking lettuce atop patty immediately | | ✓ | ✓ | ✓ | |
| Make a burger stacking patty atop lettuce immediately | | ✓ | ✓ | ✓ | |
| Make a burger stacking lettuce atop patty after preparation | | ✓ | ✓ | ✓ | ✓ |
| Make a burger stacking patty atop lettuce after preparation | | ✓ | ✓ | ✓ | ✓ |
| Make a lettuce tomato burger | ✓ | ✓ | ✓ | ✓ | |
| Make two cheese burgers | ✓ | ✓ | ✓ | ✓ | |
| Make two chicken burgers | ✓ | ✓ | ✓ | ✓ | |
| Make two burgers stacking lettuce atop patty immediately | | ✓ | ✓ | ✓ | |
| Make two burgers stacking patty atop lettuce immediately | | ✓ | ✓ | ✓ | |
| Make two burgers stacking lettuce atop patty after preparation | | ✓ | ✓ | ✓ | ✓ |
| Make two burgers stacking patty atop lettuce after preparation | | ✓ | ✓ | ✓ | ✓ |
| Make two lettuce tomato burgers | ✓ | ✓ | ✓ | ✓ | |

# 10 EPIC-Kitchens Pipeline

## 10.1 Annotations

We take 9 demonstrations of dishwashing by users 4, 7, 22 and 30, and use 2 of these as *in-context examples* for the LLM, by writing down each intermediate step's expected output.

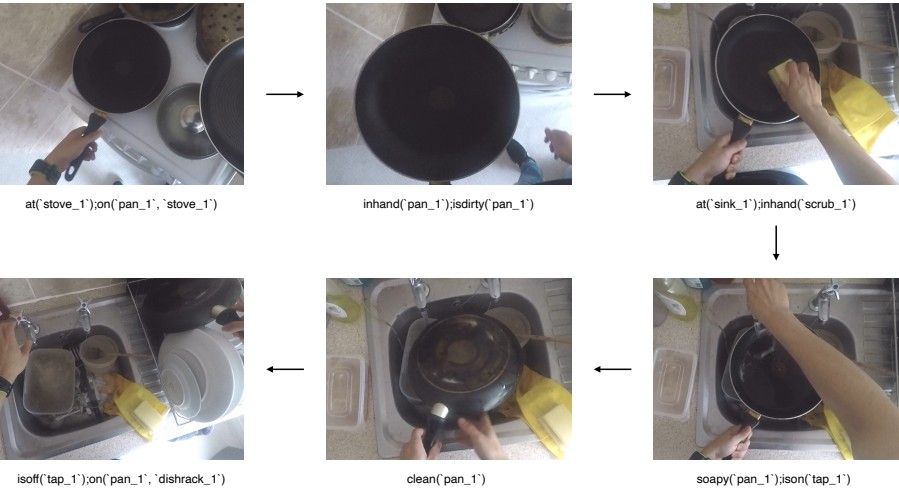

Figure 9: Example of annotations for video ID P07_10 in 6 frames

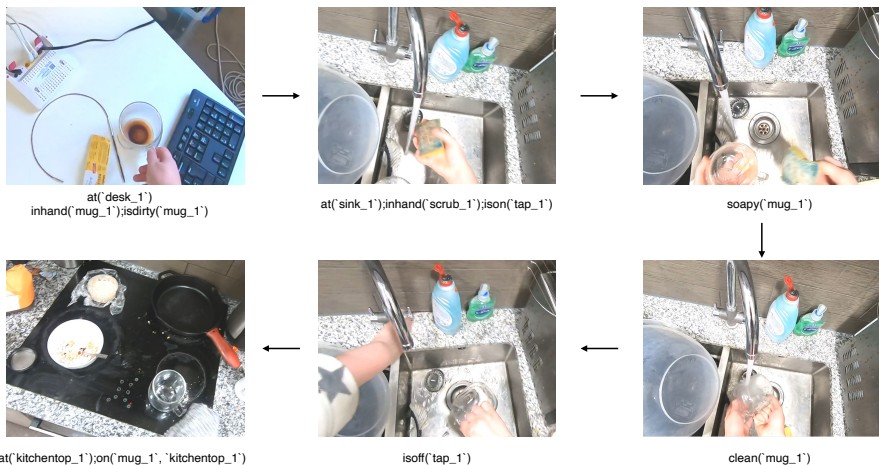

Figure 10: Example of annotations for video ID P04_101 in 6 frames

Predicates are of the form - `foo('<obj>_<id>', ...)` where `foo` is a predicate function like adjective (`is\_dirty`, `is\_soapy` etc) or preposition (`at`, `is\_in\_hand`, `on` etc). Each argument is a combination of object name and unique id, the latter added to distinguish multiple objects of the same kind. Note that these annotations or object ids are not available in the EPIC-Kitchens dataset.

Not all predicates are enumerated exhaustively, because this can be difficult for a human annotator, as well as useless and distracting for the LLM. The state predicate annotations in the demonstrations are limited to incremental changes to the observable environment. For example, `is\_in\_hand('plate_1')` comes after `in('plate_1', 'sink_1')`

Examples of these incremental state predicate annotations are described in figures 9 and 10. We avoid annotating unnecessary human-like actions like picking up something then immediately placing it back, or turning on a tap momentarily.

## 10.2 Pipeline Overview

In stage 1 recursive summarization, the LLM first recursively summarizes the provided demonstrations, which are represented as state changes since the previous state, until it determines that the

trajectories are sufficiently summarized. For this domain, the LLM in general terminates after it summarizes the trajectory into a series of high-level subtasks, which each consist of multiple states and low-level actions. For example, low-level actions "Pick up spoon_1", "Pick up fork_1", and "Go from countertop_1 to sink_1" get combined as the subtask "bring spoon_1 and fork_1 from countertop_1 to the sink_1." Then, `Demo2Code` concatenates all trajectories together before prompting the LLM to reason about the control flow (e.g. whether a for-loop is needed) before generating the task specification.

In stage 2 recursive expansion, because the dishwashing does not use that many unique actions, the LLM is asked to directly use all the low-level actions that are provided as APIs to output the task code given a specification.

The prompt demonstrating this pipeline is listed at the end of the appendix in section 18.3.

### 10.2.1   Provided Low-Level APIs

We have provided the following APIs for the perception library and low-level skill library:

- Perception Library
  - `get_all_objects()`: return a list of objects in the environment.
- Low-level Skill Library
  - `bring_objects_to_loc(obj, loc)`: bring all the objects to `loc`.
  - `turn_off(tap_name)`: turn off tap.
  - `turn_on(tap_name)`: turn on tap.
  - `soap(obj)`: soap the object.
  - `rinse(obj)`: rinse the object.
  - `pick_up(obj)`: pick up the object.
  - `place(obj, loc)`: pick up the object at `loc`.
  - `clean_with(obj, tool)`: clean the object with the `tool`, which could be a sponge or a towel.

## 11   Noisy Demonstration Ablation Experiment

As seen in our own annotations for EPIC-Kitchens demonstrations, human annotations or annotations generated by automatic scene summarizers and object detectors may not be noise-free. They may omit some predicates or completely missed predicates in an entire timestep. They may contain objects that the users did not interact with during the demonstration, so predicates about these objects are of little importance to the robot task plan. Thus, we conducted two noisy demonstration ablations:

1. Randomly removing predicates/states from the demonstrations (tested in Robotouille)
2. Randomly adding predicates about irrelevant objects to the demonstrations (tested in EPIC-Kitchens).

We found that:

- Randomly removing predicates/states
  - Removing predicates reduces `Demo2Code`'s performance for tasks with short horizons.
  - Surprisingly, it does not significantly worsen the performance for tasks with long horizons.
- Randomly adding irrelevant predicates
  - Additional irrelevant predicates worsen `Demo2Code`'s performance for correctly generating code for 5 users to 2 users.

### 11.1   Randomly removing predicates/states

### 11.1.1   Experimental Details

For each task in `Robotouille`, we modified the demonstrations in two ways:

1. for each predicate in the demonstration, there is a 10% probability that the predicate would be removed from the demonstration.

2. for each state (which could consist of multiple predicates), there is a 10% probability that the entire state would be removed from the demonstration.

We ran the experiment on 4 seeds to report the average and the variance.

### 11.1.2 Qualitative Result

We analyze a qualitative example (making a burger where the patty needs to be stacked on top of the lettuce immediately after it is cooked) where removing predicates did not affect `Demo2Code`'s performance.

When each predicate has 10% probability of being removed, the demonstration is missing 6 predicates, Half of them omits information such as picking up the lettuce, moved from one location to another location, etc. However, the other half does not omit any information. For example, one of the predicate that gets removed is "'robot1' is not holding 'top_bun3'".

```
State 26:
'top_bun3' is at 'table4'
'top_bun3' is on top of 'patty3'
>>>'robot1' is not holding 'top_bun3'<<<
```

Removing this predicate does not lose key information because "'top_bun3' is on top of 'patty3'" still indicates that 'top_bun3' has been placed on top of 'patty3'. Consequently, the LLM is still able to summarize for that state:

```
* At state 26, the robot placed 'top_bun3' on top of 'patty3' at
    location 'table4'.
```

Thus, `Demo2Code` is able to generate identical predicates

Using the same seed, when each state has 10% probability of being completely removed, the demonstration is missing 5 states (9 predicates). Because all the predicate in a selected state gets removed, the LLM misses more context. For example, because the following two states are randomly removed, the LLM does not know that the demonstration has moved and placed 'lettuce1' at 'cutting_boarding1'.

```
State 3:
'lettuce1' is not at 'table2'
'robot1' is holding 'lettuce1'

>>>State 4:<<<
>>>'robot1' is at 'cutting_board1'<<<
>>>'robot1' is not at 'table2'<<<

>>>State 5:<<<
>>>'lettuce1' is at 'cutting_board1'<<<
>>>'robot1' is not holding 'lettuce1'<<<
```

Consequently, it causes the LLM to incorrectly summarizes the states and misses the subtask of cutting the lettuce.

```
* In [Scenario 1], at state 2, the robot moved from 'table1' to '
    table2'.
* At state 3-4, the subtask is "pick up lettuce". This subtask
    contains: 1. picking up 'lettuce1' (state 3)
```

### 11.2 Randomly removing predicates/states

### 11.2.1 Experimental Details

For each EPIC-Kitchens task, we add additional predicates (i.e. showing the position of additional objects in the scene) in at least 2 separate states in the demonstrations. We also do the same

Table 7: Results for `Demo2Code`'s performance on the original EPIC-Kitchens demonstrations v.s. on the demonstrations with additional irrelevant predicates. The unit test pass rate is evaluated by a human annotator, and BLEU score is calculated between each method's code and the human annotator's reference code.

| | P4-101 (7) | | P7-04 (17) | | P7-10 (6) | | P22-05 (28) | | P22-07 (30) | | P30-07 (11) | | P30-08 (16) | |
|---|---|---|---|---|---|---|---|---|---|---|---|---|---|---|
| | Pass. | BLEU. | Pass. | BLEU. | Pass. | BLEU. | Pass. | BLEU. | Pass. | BLEU. | Pass. | BLEU. | Pass. | BLEU. |
| Demo2Code | 1.00 | 0.33 | 0.00 | 0.19 | 1.00 | 0.63 | 1.00 | 0.43 | 1.00 | 0.66 | 1.00 | 0.58 | 0.00 | 0.24 |
| Demo2Code + additional states | 0.00 | 0.21 | 0.00 | 0.15 | 1.00 | 0.27 | 0.00 | 0.22 | 0.00 | 0.49 | 1.00 | 0.67 | 0.00 | 0.22 |

modification for the training examples, while keeping the rest of the prompt identical. We expect the LLM to weed out these additional states during recursive summarization.

For example, for annotations of video ID $P22\_07$ as seen in appendix 17.3.1, we add distractions in 2 states -

```
State 3:
at(`sink_1`)
is_in_hand(`sponge_1`)
++++++is_in(`brush_1`, `sink_1`)++++++
++++++is_in(`bowl_1`, `sink_2`)++++++

State 4:
is_on(`tap_1`)
++++++on(`jar_1`, `microwave_1`)++++++
```

### 11.3 Quantitative Analysis

We see that in table 7 that `Demo2Code` suffers from degradation on most demonstrations when distracting states are added, resulting in only being able to generate correct code for 2 out of 7 demonstrations instead of 5 out of 7 demonstrations.

### 11.4 Qualitative Analysis

When adding distracting states, the LLM mostly ignores the distracting states and only shows variation in output as a result of change in input. However, the distracting states can interfere with the final code, as can be seen in the output for demonstration $P44\_101$.

Code using clean demonstrations:

```
1  objects = get_all_objects()
2  for object in objects:
3    pick_up(object)
4    go_to("sink_1")
5    pick_up("sponge_1")
6    turn_on("tap_1")
7    soap(object)
8    rinse(object)
9    turn_off("tap_1")
10   go_to("kitchentop_1")
11   place(object, "kitchentop_1")
```

Code generated with demonstration that has additional irrelevant predicates:

```
1  objects = get_all_objects()
2  for object in objects:
3    bring_objects_to_loc([object], "sink_1")
4    pick_up("sponge_1")
5    turn_on("tap_1")
6    place("brush_1", "sink_1")
7    place("bowl_1", "sink_2")
8    soap(object)
9    rinse(object)
10   turn_off("tap_1")
11   go_to("kitchentop_1")
```

```
12     place(object, "kitchentop_1")
```

When compared to the generated output with clean annotations, we see that while the `on(`jar_1`, `microwave_1`)` was ignored, `in(`brush_1`, `sink_1`)` and `in(`bowl_1`, `sink_2`)` result in the LLM generating additional code that does not align with the demonstrations. Specifically, even though `brush_1` and `bowl_1` were objects that the users were not interested in interacting with, the LLM has generated a `place()` code (lines 6-7) for these two objects. This type of mistake could be avoided by adding reasoning during recursive summarization. The LLM can be guided to ignore irrelevant objects and avoid hallucinating actions relating to these objects - for example, ground `place` action only when both `is_in_hand(...)` and `on(..., loc)` are seen one after the other.

## 12 Chain-of-thought Ablation Experiment

This experiment studies the effect of the chain-of-thought's length (in stage 1 recursive summarization) on the LLM's performance. We found:

- It is helpful to guide the LLM to take small recursive steps when summarizing demonstrations (especially for tasks with long demonstrations).
- The LLM performs the worst if it is asked to directly generate code from demonstrations.

### 12.1 Experiment Detail

We defined 3 ablation models listed below from the shortest chain-of-thought length to the longest chain length. In addition, because the tabletop's `Demo2Code` pipeline is different from `Robotouille`'s pipeline, we also describe how these pipelines are adapted to each ablation model:

- **No-Cot**: Tabletop and `Robotouille` has exactly the same process of prompting the LLM ONCE to generate code given the language model and the demonstrations.
- **1-Step**
  - Tabletop: First, the LLM receives all the demonstrations concatenated together as input to generate the specification without any intermediate reasoning. Next, the LLM generates the code given the specification.
  - `Robotouille`: First, the LLM receives all the demonstrations concatenated together as input to generate the specification. It can have intermediate reasoning because the tasks are much more complex. Next, the LLM generates the high-level code given the specification and recursively expands the code by defining all helper functions.
- **2-Steps**
  - Tabletop: First, the LLM classifies the task into either placing task or stacking task. Second, the LLM receives all the demonstrations concatenated together as input to generate the specification without any intermediate reasoning. Finally, the LLM generates the code given the specification.
  - `Robotouille`: First, for each demonstration, the LLM gets its state trajectories as input to identify a list of the low-level action that happened at each state. Second, all the low-level actions from each scenario are concatenated together and used by the LLM to generate the specification. The LLM can have intermediate reasoning at this step because the tasks are much more complex. Finally, the LLM generates the high-level code given the specification and recursively expands the code by defining all helper functions.

We identified 3 clusters of tasks based on the number of states they have, and for each cluster, we selected two tasks to test. For each task and for each of that task's specific requirements, we tested the approach 10 times and took an average of the unit test pass rate.

- Short-horizon tasks (around 2 states): "Place A next to B" and "Place A at a corner"
- Medium-horizon tasks (around 5-10 states): "Place A on top of B" and "Stack all blocks/cylinders (where there might be a maximum stack height)"
- Long-horizon tasks (more than 15 states): "Make a burger" and "Make two burgers"

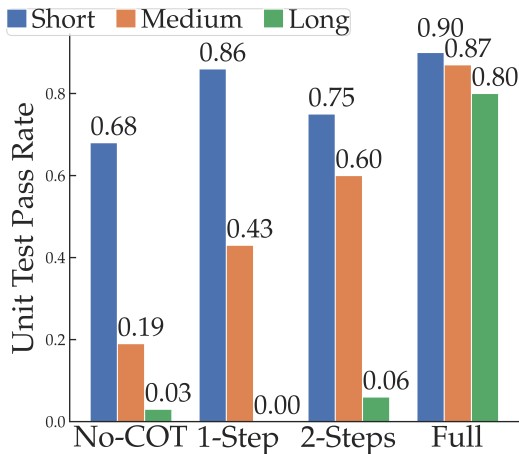

Figure 11: Ablation of different degrees of chain-of-thought (recursive summarization) across tasks with short, medium, long horizon.

## 12.2 Quantitative Result

We include the quantitative result again here as Fig. 11. Overall, `Demo2Code`/**Full** performs the best, and there's the general trend that as the length of the chain of length increases, the LLM also generates code that has a higher unit test pass rate. For short-horizon tasks, the difference in the chain-of-thought's length has a small effect on the pass rate because these tasks have short demonstrations that are easy to process without intermediate chain-of-thought steps. In contrast, both medium-horizon tasks and long-horizon tasks see great improvement when the LLM is prompted to take smaller recursive summarization steps. In the next section, we will examine one example from each cluster of tasks to analyze how the length of the chain-of-thought prompts affects the LLM's output.

## 12.3 Qualitative example for a short-horizon task

We examine all four models' outputs on the task: placing A next to B. There are 2 states per demonstration and 1 demonstration in total. The query and the **Full** approach's correctly generated code for a particular instance of the task are shown below:

```
1 objects=['yellow block', 'blue block', 'green block', 'cyan cylinder',
     'gray cylinder', 'red cylinder']
2 """
3 Place the gray cylinder next to the green block.
4
5 Initial State (State 1):
6 'green block' is not behind 'gray cylinder'
7 'gray cylinder' has not moved
8 'gray cylinder' is not in front of 'green block'
9
10 [Scenario 1]
11 State 2:
12 'green block' is behind 'gray cylinder'
13 'gray cylinder' has moved
14 'gray cylinder' is in front of 'green block'
15 """
```

```
1 say("Place the gray cylinder in front of the green block.")
2 location_pos = parse_position('in front of the green block')
3 put_first_on_second('gray cylinder', location_pos)
```

For this particular example, the LLM needs to be able to identify that the robot needs to specifically place the gray cylinder in front of the green block. Only **No-Cot** failed to generate the correct code,

while **1-step** and **2-steps** were able to identify the specification and generate the code exactly the same as **Full**.

**Why No-Cot failed?**

**No-Cot** seemed to overfit to the prompt. For this example where the specific requirement is to place A to the right of B, it outputs "to the right of the green block." We hypothesize that because the specific requirement of *placing A to the left of B* appears in the prompt, the LLM just assumes that any state that differs from the example in the prompt is *placing A to the right of B*. Below is the code outputted by **No-Cot**:

```
1 say("Place the gray cylinder to the right of the green block.")
2 location_pos = parse_position('right of the green block')
3 put_first_on_second('gray cylinder', location_pos)
```

## 12.4 Qualitative example for a medium-horizon task

We examine all four models' output on the task: placing A on top of B. There are 5 states per demonstration and 1 demonstration in total. The query and the **Full** approach's correctly generated code for a particular instance of the task are shown below:

```
1 objects=['orange block', 'brown block', 'red cylinder', 'purple
      cylinder', 'pink cylinder', 'yellow cylinder']
2 """
3 Place the brown block on the purple cylinder.
4
5 Initial State (State 1):
6 'orange block' is not on top of 'table'
7 'orange block' has not moved
8 'orange block' is on top of 'brown block'
9 'orange block' is not to the right of 'yellow cylinder'
10 'brown block' is on top of 'table'
11 'brown block' has not moved
12 'brown block' is not on top of 'purple cylinder'
13 'red cylinder' is not on top of 'table'
14 'red cylinder' has not moved
15 'red cylinder' is on top of 'orange block'
16 'red cylinder' is on top of 'brown block'
17 'red cylinder' is on top of 'pink cylinder'
18 'pink cylinder' is not on top of 'table'
19 'pink cylinder' has not moved
20 'pink cylinder' is on top of 'orange block'
21 'pink cylinder' is on top of 'brown block'
22 'yellow cylinder' is not to the left of 'orange block'
23
24 [Scenario 1]
25 State 2:
26 'red cylinder' is on top of 'table'
27 'red cylinder' has moved
28 'red cylinder' is not on top of 'orange block'
29 'red cylinder' is not on top of 'brown block'
30 'red cylinder' is not on top of 'pink cylinder'
31
32 State 3:
33 'pink cylinder' is on top of 'table'
34 'pink cylinder' has moved
35 'pink cylinder' is not on top of 'orange block'
36 'pink cylinder' is not on top of 'brown block'
37
38 State 4:
39 'orange block' is on top of 'table'
40 'orange block' has moved
41 'orange block' is not on top of 'brown block'
42 'orange block' is to the right of 'yellow cylinder'
```

```
43  'yellow cylinder' is to the left of 'orange block'
44
45  State 5:
46  'brown block' is not on top of 'table'
47  'brown block' has moved
48  'brown block' is on top of 'purple cylinder'
49  """
```

```
1  say("First, place the red cylinder, the pink cylinder, and the orange
       block on the table. Then, place the brown block on the purple
       cylinder.")
2  items_to_place_first_in_order = ['red cylinder', 'pink cylinder', '
       orange block']
3  for item in items_to_place_first_in_order:
4      put_first_on_second(item, "table")
5  put_first_on_second('brown block', 'purple cylinder')
```

For this particular example, the LLM needs to recognize that there are 3 additional objects on top of the brown block. In order to place the brown block on top of the purple cylinder, the robot needs to first place those 3 additional objects on the table. **No-Cot** and **1-step** approaches failed to generate the correct code, while **2-steps** was able to identify the specification and generate the code exactly the same as **Full**.

**Why No-Cot and 1-step failed?**

Both **No-Cot** and **1-step** seem to be over-reliant on the language instruction instead of reasoning through the state transitions. Specifically, examining the specification outputted by **1-step** suggests that the LLM simply copied the language instruction instead of recognizing that more objects are moved than just moving the brown block to be on top of the purple cylinder.

```
1  The detailed requirement is: <Place the brown block on the purple
       cylinder.
```

Below show the identical code generated by **No-Cot** and **1-step**.

```
1  say("Place the brown block on the purple cylinder.")
2  put_first_on_second('brown block', 'purple cylinder')
```

## 12.5   Qualitative example for a long-horizon task

We examine all four models' output on the long-horizon task: making a burger with patty, lettuce, and tomato. This task has 71 states per demonstration and 2 demonstrations in total.

The **Full** pipeline's approach outputs the correct code as shown below. Next, we analyze how other models failed to produce the correct code.

```
1  # Get a list of all the patties in the kitchen.
2  patties = get_all_obj_names_that_match_type('patty')
3  # Get a list of all the lettuces in the kitchen.
4  lettuces = get_all_obj_names_that_match_type('lettuce')
5  # Get a list of all the tomatoes in the kitchen.
6  tomatoes = get_all_obj_names_that_match_type('tomato')
7  # Get a list of all the bottom buns in the kitchen.
8  bottom_buns = get_all_obj_names_that_match_type('bottom bun')
9  # Get a list of all the top buns in the kitchen.
10 top_buns = get_all_obj_names_that_match_type('top bun')
11 # Get a list of all the stoves in the kitchen.
12 stoves = get_all_location_names_that_match_type('stove')
13 # Get a list of all the cutting boards in the kitchen.
14 cutting_boards = get_all_location_names_that_match_type('cutting_board
       ')
15 # Get a list of all the tables in the kitchen.
16 tables = get_all_location_names_that_match_type('table')
17
```

```
18  # Decide a stove to use.
19  stove_to_use = stoves[0]
20  # Decide a cutting board to use.
21  cutting_board_to_use = cutting_boards[0]
22  # Decide a table to use.
23  table_to_use = tables[0]
24
25  # Repeat the following twice:
26  for i in range(2):
27      # Decide a patty to cook.
28      patty_to_cook = patties[i]
29      # Cook that patty at that stove.
30      cook_object_at_location(obj=patty_to_cook, location=stove_to_use)
31      # Decide a bottom bun to use.
32      bottom_bun_to_use = bottom_buns[i]
33      # Stack the patty on top of the bottom bun.
34      stack_obj1_on_obj2(obj1=patty_to_cook, obj2=bottom_bun_to_use)
35
36      # Decide a tomato to cut.
37      tomato_to_cut = tomatoes[i]
38      # Cut that tomato at that cutting board.
39      cut_object_at_location(obj=tomato_to_cut, location=
        cutting_board_to_use)
40      # Stack the tomato on top of the patty.
41      stack_obj1_on_obj2(obj1=tomato_to_cut, obj2=patty_to_cook)
42
43      # Decide a lettuce to cut.
44      lettuce_to_cut = lettuces[i]
45      # Cut that lettuce at that cutting board.
46      cut_object_at_location(obj=lettuce_to_cut, location=
        cutting_board_to_use)
47      # Stack the lettuce on top of the tomato.
48      stack_obj1_on_obj2(obj1=lettuce_to_cut, obj2=tomato_to_cut)
49
50      # Stack the top bun on top of the lettuce, tomato, and patty stack
        .
51      top_bun_to_use = top_buns[i]
52      stack_obj1_on_obj2(obj1=top_bun_to_use, obj2=lettuce_to_cut)
```

**Why No-Cot failed?**

The main issue that **No-Cot** suffers from is the maximum token length issue because the prompt needs to include demonstrations as the example query and solution code as the example response. Thus, the LLM was not able to finish completely the code as seen below (line 37). In addition, although the LLM showed some ability to use the control flow, it was only creating a for loop to cook the patties but it did not use a for-loop to assemble the burger. Specifically, the LLM generated code that index to specific items: (e.g. line 23 bottom_buns[0], line 25 lettuces[0], etc.)

```
1   # Get a list of all the patties in the kitchen.
2   patties = get_all_obj_names_that_match_type('patty')
3   # Get a list of all the stoves in the kitchen.
4   stoves = get_all_location_names_that_match_type('stove')
5   # Decide a stove to go to.
6   stove_to_cook_at = stoves[0]
7
8   # Repeat the following twice
9   for i in range(2):
10      # Decide a patty to cook.
11      patty_to_cook = patties[i]
12      # Cook that patty at that stove.
13      cook_object_at_location(obj=patty_to_cook, location=
        stove_to_cook_at)
14
15  # Get a list of all the bottom buns in the kitchen.
```

```
16 bottom_buns = get_all_obj_names_that_match_type('bottom bun')
17 # Get a list of all the lettuces in the kitchen.
18 lettuces = get_all_obj_names_that_match_type('lettuce')
19 # Get a list of all the tomatoes in the kitchen.
20 tomatoes = get_all_obj_names_that_match_type('tomato')
21
22 # Decide a bottom bun to use.
23 bottom_bun_to_use = bottom_buns[0]
24 # Decide a lettuce to use.
25 lettuce_to_use = lettuces[0]
26 # Decide a tomato to use.
27 tomato_to_use = tomatoes[0]
28
29 # Stack the patty on top of the bottom bun.
30 # obj1 should be the patty, obj2 should be the bottom bun.
31 stack_obj1_on_obj2(obj1=patties[0], obj2=bottom_bun_to_use)
32
33 # Stack the lettuce on top of the patty.
34 # obj1 should be the lettuce, obj2 should be the patty on top of the
       bottom bun.
35 stack_obj1_on_obj2(obj1=lettuce_to_use, obj2=patties[0])
36
37 #
```

**Why 1-step failed?**

In Fig. 11's result, **1-step** actually performed worse than **No-Cot**. The worsened performance is caused by low-quality specifications produced by the LLM. Because stage 2 recursive expansion is highly reliant on the specification that gets passed in, any mistaken or missing specification will be reflected in the code as well. Below shows the intermediate reasonings and specification outputted by the LLM. Notice that it does not capture any information about the lettuce and the tomato needing to be cut before being added to the burger (see lines 22-23 and lines 25-26).

```
1 * The order of high level actions is: ['cook', 'assemble', 'cook', '
      assemble']
2 * Because there are two 'cook' actions and two 'assemble' actions, we
       assume that we must cook and assemble each burger separately.
3 Thus:
4 <Make two burgers.
5
6 Specifically:
7 Get a list of all the patties in the kitchen.
8 Get a list of all the stoves in the kitchen.
9 Get a list of all the buns in the kitchen.
10 Get a list of all the lettuces in the kitchen.
11 Get a list of all the tomatoes in the kitchen.
12
13 Repeat the following twice:
14     Decide a patty to use.
15     Decide a stove to go to.
16     Cook that patty at that stove.
17
18     Decide a bottom bun to use.
19     Place that bottom bun on a plate.
20     Place that patty on top of the bottom bun.
21
22     Decide a lettuce to use.
23     Place that lettuce on top of the patty.
24
25     Decide a tomato to use.
26     Place that tomato on top of the lettuce.
27
28     Decide a top bun to use.
29     Place that top bun on top of the tomato.
```

Consequently, although the code (shown below) closely matched the specification and included the for-loop, because the specification was wrong, the code also failed to solve the problem.

```python
# Get a list of all the patties in the kitchen.
patties = get_all_obj_names_that_match_type('patty')
# Get a list of all the stoves in the kitchen.
stoves = get_all_location_names_that_match_type('stove')
# Get a list of all the buns in the kitchen.
buns = get_all_obj_names_that_match_type('bottom bun') +
    get_all_obj_names_that_match_type('top bun')
# Get a list of all the lettuces in the kitchen.
lettuces = get_all_obj_names_that_match_type('lettuce')
# Get a list of all the tomatoes in the kitchen.
tomatoes = get_all_obj_names_that_match_type('tomato')

# Repeat the following twice:
for i in range(2):
    # Decide a patty to use.
    patty_to_use = patties[i]
    # Decide a stove to go to.
    stove_to_cook_at = stoves[i]
    # Cook that patty at that stove.
    cook_object_at_location(obj=patty_to_use, location=
    stove_to_cook_at)

    # Decide a bottom bun to use.
    bottom_bun_to_use = buns[i*2]
    # Place that bottom bun on a plate.
    place_obj_on_obj(obj1=bottom_bun_to_use, obj2='plate')
    # Place that patty on top of the bottom bun.
    place_obj_on_obj(obj1=patty_to_use, obj2=bottom_bun_to_use)

    # Decide a lettuce to use.
    lettuce_to_use = lettuces[i]
    # Place that lettuce on top of the patty.
    place_obj_on_obj(obj1=lettuce_to_use, obj2=patty_to_use)

    # Decide a tomato to use.
    tomato_to_use = tomatoes[i]
    # Place that tomato on top of the lettuce.
    place_obj_on_obj(obj1=tomato_to_use, obj2=lettuce_to_use)

    # Decide a top bun to use.
    top_bun_to_use = buns[i*2+1]
    # Place that top bun on top of the tomato.
    place_obj_on_obj(obj1=top_bun_to_use, obj2=tomato_to_use)
```

**Why 2-steps failed?**

**2-steps** improved its specification compared to **1-step**'s. Specifically, its specification was able to identify that lettuce and tomato require cutting before they are stacked as a part of the burger. However, it still fails to identify the physical constraint that in order to cut an object, it also needs to specify the location. While it is able to identify the stove used for cooking (line 17), it fails to identify the cutting board (line 22 and line 25).

```
* The order of high level actions is: ['cook', 'assemble', 'cook', '
    assemble']
* Because 'cook' is before 'assemble', we assume that we must cook the
     patties before assembling the burgers.
* Because 'assemble' is repeated twice, we assume that we must
    assemble two burgers.
Thus:
<Make two burgers.

Specifically:
```

```
 8  Get a list of all the patties in the kitchen.
 9  Get a list of all the buns in the kitchen.
10  Get a list of all the lettuces in the kitchen.
11  Get a list of all the tomatoes in the kitchen.
12
13  Repeat the following twice:
14      Decide a patty to cook.
15      Get a list of all the stoves in the kitchen.
16      Decide a stove to use.
17      Cook that patty on that stove.
18      Decide a bun to use as the bottom bun.
19      Place that bun on a plate.
20      Stack that patty on top of that bun.
21      Decide a tomato to use.
22      Cut that tomato.
23      Place that tomato on top of that patty.
24      Decide a lettuce to use.
25      Cut that lettuce.
26      Place that lettuce on top of that tomato.
27      Decide a bun to use as the top bun.
28      Place that bun on top of that lettuce.
```

Consequently, in the code that the LLM produced based on the specification, it created a function `cut_object(obj)`. Then, because the function signature does not specify any location to cut the object at, the downstream function definition also fails (see the definition starting from line 43).

```
 1  # Get a list of all the patties in the kitchen.
 2  patties = get_all_obj_names_that_match_type('patty')
 3  # Get a list of all the buns in the kitchen.
 4  buns = get_all_obj_names_that_match_type('bottom bun') +
          get_all_obj_names_that_match_type('top bun')
 5  # Get a list of all the lettuces in the kitchen.
 6  lettuces = get_all_obj_names_that_match_type('lettuce')
 7  # Get a list of all the tomatoes in the kitchen.
 8  tomatoes = get_all_obj_names_that_match_type('tomato')
 9
10  # Repeat the following twice:
11  for i in range(2):
12      # Decide a patty to cook.
13      patty_to_cook = patties[i]
14      # Get a list of all the stoves in the kitchen.
15      stoves = get_all_location_names_that_match_type('stove')
16      # Decide a stove to use.
17      stove_to_use = stoves[i]
18      # Cook that patty on that stove.
19      cook_object_at_location(obj=patty_to_cook, location=stove_to_use)
20      # Decide a bun to use as the bottom bun.
21      bottom_bun_to_use = buns[i]
22      # Place that bun on a plate.
23      place_obj_on_obj(obj1=bottom_bun_to_use, obj2='plate')
24      # Stack that patty on top of that bun.
25      stack_obj1_on_obj2(obj1=patty_to_cook, obj2=bottom_bun_to_use)
26      # Decide a tomato to use.
27      tomato_to_use = tomatoes[i]
28      # Cut that tomato.
29      cut_object(obj=tomato_to_use)
30      # Place that tomato on top of that patty.
31      place_obj_on_obj(obj1=tomato_to_use, obj2=patty_to_cook)
32      # Decide a lettuce to use.
33      lettuce_to_use = lettuces[i]
34      # Cut that lettuce.
35      cut_object(obj=lettuce_to_use)
36      # Place that lettuce on top of that tomato.
37      place_obj_on_obj(obj1=lettuce_to_use, obj2=tomato_to_use)
38      # Decide a bun to use as the top bun.
```

```
39      top_bun_to_use = buns[i+2]
40      # Place that bun on top of that lettuce.
41      place_obj_on_obj(obj1=top_bun_to_use, obj2=lettuce_to_use)
42
43 def cut_object(obj):
44      if not is_holding(obj):
45          # To cut obj, the robot needs to hold obj first.
46          if is_in_a_stack(obj):
47              # Because obj is in a stack, robot need to move then
    unstack the obj from the obj_at_bottom first
48              obj_at_bottom = get_obj_that_is_underneath(obj_at_top=obj)
49              move_then_unstack(obj_to_unstack=obj, obj_at_bottom=
    obj_at_bottom, unstack_location=get_obj_location(obj_at_bottom))
50          else:
51              # Since obj is not in a stack, robot can just move then
    pick it up
52              move_then_pick(obj=obj)
53      # cut the object
54      cut_until_is_cut(obj=obj)
```

# 13 Intermediate Reasoning Ablation Experiment

This experiment studies how the thoroughness of intermediate reasoning (in stage 3 of recursive summarization) affects the LLM's performance. We found:

- Our approach may be improved by ablating away some part of reasoning (e.g. listing the high-level actions)
- Having intermediate reasoning of any form is helpful for hard cooking tasks that have many sub-tasks
- Future work is necessary to engineer better prompts

## 13.1 Experiment detail

This experiment compares `Demo2Code` (labeled as **Full**) with three additional ablation models each with differing levels of reasoning

- **No reasoning:** The LLM generates the specification directly from step 2 of recursive summarization with no intermediate reasoning.
- **Only List:** The LLM generates the specification after intermediate reasoning which lists the high-level actions in common with the scenarios from step 2 of recursive summarization.
- **Only Analyze:** The LLM generates the specification after intermediate reasoning which describes the repetition and ordering of high-level actions from step 2 of recursive summarization.

These models are tested on all the `Robotouille` tasks. We use 3 clusters of tasks based on the number of high-level actions/sub-tasks they have.

- Easy cooking tasks ($\leq 2$ high-level actions/sub-tasks): "cook and cut", "cook two patties", and "cut two lettuces"
- Normal cooking tasks (between 2-7 high-level actions/sub-tasks): "make a burger" and "assemble two burgers with already cooked patties"
- Hard cooking tasks ($\geq 8$ high-level actions/sub-tasks): "make two burgers"

## 13.2 Quantitative result

For each task and for each of that task's specific requirements, we tested each model on 10 randomly generated environments and took an average of the unit test pass rate. Fig. 12 shows that the **Only Analyze** model outperforms the other models. All methods easily achieve 100% pass rate on Easy

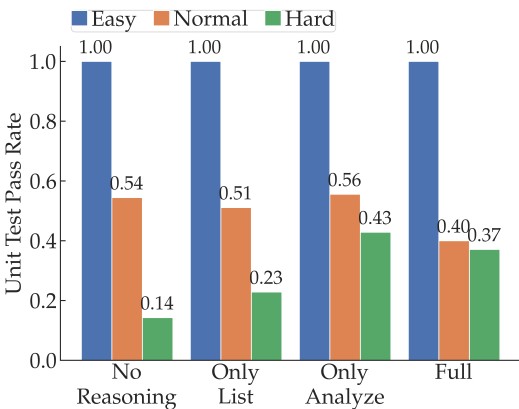

Figure 12: Ablation of different amount and style of intermediate reasoning at step 3 of recursive summarization across `Robotouille` tasks. These tasks are clustered into easy cooking tasks, normal cooking tasks, and hard cooking tasks based on the number of high-level actions/sub-tasks each task has.

tasks. There is a trend that including any kind of reasoning improves performance on hard tasks. **Only List** only slightly suffers in Normal performance while increasing significantly for Hard performance relative to **No Reasoning**. **Full**'s Normal performance suffers significantly yet its Hard performance increases as well, suggesting a tradeoff between Normal performance and Hard performance; however, **Only Analyze** has the highest Normal and Hard performance of all models. Since **Full** combines both **Only List** and **Only Analyze**, this shows that concatenating reasoning methods doesn't necessarily mean better performance.

### 13.3 Qualitative example

In this section, we show a qualitative example of all models' output on the long-horizon task "make two burgers" with the specific requirement "stack lettuce atop patty immediately." We choose this example to show the effects of reasoning on the code as well as to shed light on why **Full** underperforms relative to **Only Analyze**. **Only Analyze** is the only model with 1.0 success rate while the other models all have 0.0 success rates. For each model's output, the helper functions can be found in section 19.3.

#### 13.3.1 Why No Reasoning failed?

**No Reasoning** is the only unsuccessful model that generates code that executes. Due to the lack of any reasoning, it performs the task correctly but with the incorrect requirement; rather than preparing one burger at a time and assembling the burger immediately as ingredients are prepared, it prepares all ingredients before assembling both burgers in parallel.

```python
# Get a list of all the patties in the kitchen.
patties = get_all_obj_names_that_match_type('patty')
# Get a list of all the lettuces in the kitchen.
lettuces = get_all_obj_names_that_match_type('lettuce')
# Get a list of all the bottom buns in the kitchen.
bottom_buns = get_all_obj_names_that_match_type('bottom bun')
# Get a list of all the top buns in the kitchen.
top_buns = get_all_obj_names_that_match_type('top bun')
# Get a list of all the stoves in the kitchen.
stoves = get_all_location_names_that_match_type('stove')
# Get a list of all the cutting boards in the kitchen.
cutting_boards = get_all_location_names_that_match_type('cutting_board
    ')

# Decide two patties to cook.
patties_to_cook = patties[:2]
# Cook those patties at two different stoves.
for i in range(2):
```

```
18      cook_object_at_location(obj=patties_to_cook[i], location=stoves[i
        ])
19
20  # Decide two lettuces to cut.
21  lettuces_to_cut = lettuces[:2]
22  # Cut those lettuces at two different cutting boards.
23  for i in range(2):
24      cut_object_at_location(obj=lettuces_to_cut[i], location=
        cutting_boards[i])
25
26  # Decide two bottom buns to use.
27  bottom_buns_to_use = bottom_buns[:2]
28  # Stack a patty on top of each bottom bun.
29  for i in range(2):
30      patty_to_use = patties_to_cook[i]
31      bottom_bun_to_use = bottom_buns_to_use[i]
32      stack_obj1_on_obj2(obj1=patty_to_use, obj2=bottom_bun_to_use)
33
34  # Decide two lettuces to use.
35  lettuces_to_use = lettuces_to_cut[:2]
36  # Stack a lettuce on top of each patty.
37  for i in range(2):
38      lettuce_to_use = lettuces_to_use[i]
39      patty_to_use = patties_to_cook[i]
40      stack_obj1_on_obj2(obj1=lettuce_to_use, obj2=patty_to_use)
41
42  # Decide two top buns to use.
43  top_buns_to_use = top_buns[:2]
44  # Stack a top bun on top of each lettuce.
45  for i in range(2):
46      top_bun_to_use = top_buns_to_use[i]
47      lettuce_to_use = lettuces_to_use[i]
48      stack_obj1_on_obj2(obj1=top_bun_to_use, obj2=lettuce_to_use)
```

### 13.3.2   Why Only List failed?

**Only List** improves on **No Reasoning** by generating code in the correct order of the specification. Patties are cooked and then stacked on the bottom bun before lettuces are cut and stacked on the patties before finally the top buns. While the order is now correct, just specifying the order wasn't effective enough because the reasoning missed making burgers one by one rather than all in parallel. In addition, this code doesn't execute due to the attempt of cutting lettuces back to back at the same cutting board. This violates the environment's constraints since only one object can be at one station at a time.

```
1   # Get a list of all the patties in the kitchen.
2   patties = get_all_obj_names_that_match_type('patty')
3   # Get a list of all the lettuces in the kitchen.
4   lettuces = get_all_obj_names_that_match_type('lettuce')
5   # Get a list of all the bottom buns in the kitchen.
6   bottom_buns = get_all_obj_names_that_match_type('bottom bun')
7   # Get a list of all the top buns in the kitchen.
8   top_buns = get_all_obj_names_that_match_type('top bun')
9   # Get a list of all the stoves in the kitchen.
10  stoves = get_all_location_names_that_match_type('stove')
11  # Get a list of all the cutting boards in the kitchen.
12  cutting_boards = get_all_location_names_that_match_type('cutting_board
        ')
13
14  # Decide two patties to cook.
15  patties_to_cook = patties[:2]
16  # Cook those patties at two stoves.
17  for i in range(2):
18      cook_object_at_location(obj=patties_to_cook[i], location=stoves[i
        ])
```

```
19
20  # Decide two bottom buns to use.
21  bottom_buns_to_use = bottom_buns[:2]
22  # Stack a patty on top of each bottom bun.
23  for i in range(2):
24      stack_obj1_on_obj2(obj1=patties_to_cook[i], obj2=
        bottom_buns_to_use[i])
25
26  # Decide two lettuces to use.
27  lettuces_to_use = lettuces[:2]
28  # Cut those lettuces at a cutting board.
29  for lettuce in lettuces_to_use:
30      cut_object(obj=lettuce, location=cutting_boards[0])
31  # Stack a lettuce on top of each patty.
32  for i in range(2):
33      stack_obj1_on_obj2(obj1=lettuces_to_use[i], obj2=patties_to_cook[i
        ])
34
35  # Decide two top buns to use.
36  top_buns_to_use = top_buns[:2]
37  # Stack a top bun on top of each lettuce.
38  for i in range(2):
39      stack_obj1_on_obj2(obj1=top_buns_to_use[i], obj2=lettuces_to_use[i
        ])
```

### 13.3.3 Why Full failed?

**Full** gets much closer to correctly solving the task than **Only List**. Unlike **Only List**, **Full** has the correct ordering for the requirement; it stacks the patty and lettuce immediately after preparation and it does so for an entire burger before moving on. The issue is that while with the analysis of ordering and repetition it manages to put all the code in one for loop rather than multiple, it retained some of the poor aspects of **Only List** by attempting to cook two patties at once. Doing this within the for loop is incorrect logic. In addition, the cooking is attempted on the same stove twice which violates the environment's constraints of one item being on a station at a time.

```
1   # Get a list of all the bottom buns in the kitchen.
2   bottom_buns = get_all_obj_names_that_match_type('bottom bun')
3   # Get a list of all the lettuces in the kitchen.
4   lettuces = get_all_obj_names_that_match_type('lettuce')
5   # Get a list of all the patties in the kitchen.
6   patties = get_all_obj_names_that_match_type('patty')
7   # Get a list of all the top buns in the kitchen.
8   top_buns = get_all_obj_names_that_match_type('top bun')
9   # Get a list of all the stoves in the kitchen.
10  stoves = get_all_location_names_that_match_type('stove')
11  # Get a list of all the cutting boards in the kitchen.
12  cutting_boards = get_all_location_names_that_match_type('cutting_board
        ')
13
14  # Repeat the following twice
15  for i in range(2):
16      # Decide a stove to use.
17      stove_to_use = stoves[i]
18      # Cook two patties at that stove.
19      for j in range(2):
20          cook_object_at_location(obj=patties[j], location=stove_to_use)
21
22      # Decide a bottom bun to use.
23      bottom_bun_to_use = bottom_buns[i]
24      # Stack the patty on top of the bottom bun.
25      # obj1 should be the patty, obj2 should be the bottom bun.
26      stack_obj1_on_obj2(obj1=patties[i], obj2=bottom_bun_to_use)
27
28      # Decide a lettuce to use.
```

```
29        lettuce_to_use = lettuces[i]
30        # Cut that lettuce at a cutting board.
31        cut_object_at_location(obj=lettuce_to_use, location=cutting_boards
          [i])
32        # Stack the lettuce on top of the patty.
33        # obj1 should be the lettuce, obj2 should be the patty on top of
          the bottom bun.
34        stack_obj1_on_obj2(obj1=lettuce_to_use, obj2=patties[i])
35
36        # Decide a top bun to use.
37        top_bun_to_use = top_buns[i]
38        # Stack the top bun on top of the lettuce.
39        # obj1 should be the top bun, obj2 should be the lettuce on top of
           the patty on top of the bottom bun.
40        stack_obj1_on_obj2(obj1=top_bun_to_use, obj2=lettuce_to_use)
```

### 13.3.4 Only Analyze Output

`Only Analyze` is the only model to adhere to the environment's constraints. Note that while the same stove is used for cooking twice, the second time a patty is cooked the first patty has already been removed from the stove and stacked into the first burger.

```
1  # Get a list of all the patties in the kitchen.
2  patties = get_all_obj_names_that_match_type('patty')
3  # Get a list of all the lettuces in the kitchen.
4  lettuces = get_all_obj_names_that_match_type('lettuce')
5  # Get a list of all the bottom buns in the kitchen.
6  bottom_buns = get_all_obj_names_that_match_type('bottom bun')
7  # Get a list of all the top buns in the kitchen.
8  top_buns = get_all_obj_names_that_match_type('top bun')
9  # Get a list of all the stoves in the kitchen.
10 stoves = get_all_location_names_that_match_type('stove')
11 # Get a list of all the cutting boards in the kitchen.
12 cutting_boards = get_all_location_names_that_match_type('cutting_board
      ')
13
14 # Decide a stove to use.
15 stove_to_use = stoves[0]
16
17 # Repeat the following twice
18 for i in range(2):
19     # Decide a patty to cook.
20     patty_to_cook = patties[i]
21     # Cook that patty at that stove.
22     cook_object_at_location(obj=patty_to_cook, location=stove_to_use)
23
24     # Decide a bottom bun to use.
25     bottom_bun_to_use = bottom_buns[i]
26     # Stack the patty on top of the bottom bun.
27     # obj1 should be the patty, obj2 should be the bottom bun.
28     stack_obj1_on_obj2(obj1=patty_to_cook, obj2=bottom_bun_to_use)
29
30     # Decide a lettuce to use.
31     lettuce_to_use = lettuces[i]
32     # Cut that lettuce at that cutting board.
33     cut_object_at_location(obj=lettuce_to_use, location=cutting_boards
          [i])
34     # Stack the lettuce on top of the patty.
35     # obj1 should be the lettuce, obj2 should be the patty on top of
          the bottom bun.
36     stack_obj1_on_obj2(obj1=lettuce_to_use, obj2=patty_to_cook)
37
38     # Decide a top bun to use.
39     top_bun_to_use = top_buns[i]
```

```
40      # Stack the top bun on top of the lettuce.
41      # obj1 should be the top bun, obj2 should be the lettuce on top of
        the patty on top of the bottom bun.
42      stack_obj1_on_obj2(obj1=top_bun_to_use, obj2=lettuce_to_use)
```

# 14    Recursive Expansion Ablation Experiment

This experiment studies how the number of recursive code expansion steps (in stage 2 recursive expansion) affects the LLM's performance. We found:

- It is helpful to guide the LLM to slowly expand the code instead of asking it to directly generate all the code at once using only the low-level imported APIs.

- The initial code that LLM uses to expand the rest of the functions should align closer to the given specifications.

## 14.1   Experiment detail

This experiment compares `Demo2Code` (labeled as **Full**) with three additional ablation models each with a different amount of recursive code expansion steps:

- **1-layer:** Given a specification, the LLM directly outputs the code for the task using only low-level action and perception APIs from the import statement. The LLM is not allowed to define any helper function.

- **2-layer (Comp):** Given a specification, the LLM first outputs corresponding code that can call undefined "composite functions." Each composite functions contain at most two low-level actions, e.g. `move_then_pick`, `move_then_stack`. In the next step, the LLM defines these composite functions using only low-level action and perception APIs from the import statement.

- **2-layer (High):** Given a specification, the LLM first outputs corresponding high-level code that can call undefined "high-level functions." This step is the same as `Demo2Code`. Then, the LLM defines these high-level functions using only low-level action and perception APIs from the import statement.

These models are tested on all the `Robotouille` tasks. Because section 13 on Intermediate Reasoning Ablation Experiments also tested only on `Robotouille`, we use the same 3 clusters of tasks based on the number of high-level actions/sub-tasks they have.

- Easy cooking tasks ($\leq 2$ high-level actions/sub-tasks): "cook and cut", "cook two patties", and "cut two lettuces"

- Normal cooking tasks (between 2-7 high-level actions/sub-tasks): "make a burger" and "assemble two burgers with already cooked patties"

- Hard cooking tasks ($\geq 8$ high-level actions/sub-tasks): "make two burgers"

## 14.2   Quantitative result

For each task and for each of that task's specific requirements, we tested each model on 10 randomly generated environments and took an average of the unit test pass rate. Fig. 13 shows how our approach `Demo2Code` (**Full**) outperforms the other models. Overall, as the number of recursive code generation steps increases, the LLM's performance also increases. Interestingly, **2-Layers (Comp)** performs worse than **2-Layer's (High)** despite having the same amount of recursive steps. The two model's performance difference suggests that the granularity of the initial code also affects the performance. While **2-Layers (High)** first asks the LLM to generate high-level code, which is the same approach as `Demo2Code`, **2-Layers (Comp)** first asks the LLM to generate code that calls composite functions. Consequently, for **2-Layers (Comp)**, the LLM needs to first output longer, more complicated code the LLM has to produce in one-step, which is more prone to error.

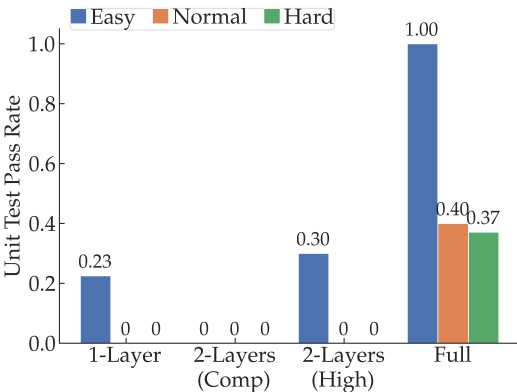

Figure 13: Ablation of different amount of recursive code expansion steps across `Robotouille` tasks. These tasks are clustered into easy cooking task, normal cooking task, and hard cooking task based on the number of high-level actions/sub-tasks each task has.

## 14.3 Qualitative example

In this section, we show a qualitative example of all models' output on the short-horizon task "cut two lettuces." Note that cutting lettuce has not been shown in the prompt, so the LLM needs to be able to generalize to this new task and properly use a for-loop to write efficient code. **1-layer** and **2-layer (Comp)** had 0.0 success rate. **2-layer (High** has 0.2 success rate, and `Demo2Code` has 1.0 success rate. All models are able to generate the for-loop, as evident in the outputted code below. However, all ablation models struggle to generate code that adheres to physical constraints and applies to any random environment.

### 14.3.1 Why 1-Layer failed?

**1-Layer** generates the correct code to cut the lettuce. Despite not seeing how the low-level action `cut()` should be used in the prompt, it correctly writes the code that will cut the lettuce until the lettuce becomes cut. However, the LLM misses the physical constraints associated with cooking, e.g. in order to cut the lettuce at a cutting board, the robot needs to make sure that there is a lettuce at the cutting board before starting to cut. If there is no lettuce on the cutting board, the robot first needs to go pick up a lettuce and bring it to the cutting board. Another issue that **1-Layer** encounters is reaching the LLM maximum token length. For longer tasks, it cannot finish generating all the code, so its overall execution success rate is only 0.04.

```
1  # Get a list of all the lettuces in the kitchen.
2  lettuces = get_all_obj_names_that_match_type('lettuce')
3  # Get a list of all the cutting boards in the kitchen.
4  cutting_boards = get_all_location_names_that_match_type('cutting_board
       ')
5
6  # Repeat the following twice
7  for i in range(2):
8      # Decide a cutting board to use.
9      cutting_board_to_use = cutting_boards[i]
10     # Decide a lettuce to cut.
11     lettuce_to_cut = lettuces[i]
12     # Cut that lettuce at that cutting board.
13     # To cut lettuce_to_cut, the robot first needs to be at the same
       location as cutting_board_to_use
14     if get_curr_location() != cutting_board_to_use:
15         move(get_curr_location(), cutting_board_to_use)
16     while not is_cut(lettuce_to_cut):
17         cut(lettuce_to_cut)
```

### 14.3.2 Why 2-Layers (Comp) failed?

**2-Layer (Comp)** faces the same issue of not adhering to all the physical constraints. For example, it successfully identifies that it needs to pick up the lettuce before going to the cutting board and cutting it. However, it does not place the lettuce on the cutting board before it starts cutting (see line 27).

In addition, all the composite functions defined in the recursive step are technically correct even though they are not efficient or compact. For example, `move_then_cut` unnecessarily moves to the object to cut before actually starting to cut (see line 19). It then moves to the cut location inside the while loop of checking if the object has been cut yet or not (see line 22) A more optimal code will be to move to the cut location before the while loop, then just have the call to `cut` function in the while loop.

A core issue with **2-Layer (Comp)** is that the LLM needs to account for the physical constraints when generating the code in the first step. However, the specification that the LLM receives only includes information about the order between high-level actions and high-level control flows (e.g. for loops). The LLM is more prone to mistakes when it has to extrapolate the physical constraints from the high-level specification. Meanwhile, **2-Layer (High)**, which has the same number of recursive steps, has generated the correct initial code that the LLM can expand on. Its initial code aligns closer to the specification (see the code in section 14.3.3).

```python
1  # Get a list of all the lettuces in the kitchen.
2  lettuces = get_all_obj_names_that_match_type('lettuce')
3  # Get a list of all the cutting boards in the kitchen.
4  cutting_boards = get_all_location_names_that_match_type('cutting_board
       ')
5
6  # Decide a cutting board to use.
7  cutting_board_to_use = cutting_boards[0]
8
9  # Repeat the following twice
10 for i in range(2):
11     # Decide a lettuce to cut.
12     lettuce_to_cut = lettuces[i]
13     # Cut that lettuce at that cutting board.
14     # To cut lettuce_to_cut, the robot first needs to be holding
       lettuce_to_cut
15     if not is_holding(lettuce_to_cut):
16         # If the robot is not holding lettuce_to_cut, there are 2
       scenarios:
17         #   (1) if lettuce_to_cut is in a stack ,unstack
       lettuce_to_cut
18         #   (2) else, pick up lettuce_to_cut.
19         if is_in_a_stack(lettuce_to_cut):
20             # Because lettuce_to_cut is in a stack, robot need to move
        then unstack the lettuce_to_cut from the obj_at_bottom first
21             obj_at_bottom = get_obj_that_is_underneath(obj_at_top=
       lettuce_to_cut)
22             move_then_unstack(obj_to_unstack=lettuce_to_cut,
       obj_at_bottom=obj_at_bottom, unstack_location=get_obj_location(
       obj_at_bottom))
23         else:
24             # Since lettuce_to_cut is not in a stack, robot can just
       move then pick it up
25             move_then_pick(obj=lettuce_to_cut)
26     # move to the cutting_board_to_use's location then cut
       lettuce_to_cut
27     move_then_cut(obj_to_cut=lettuce_to_cut, cut_location=
       cutting_board_to_use)
28
29 def move_then_unstack(obj_to_unstack, obj_at_bottom, unstack_location)
       :
30     # For unstacking, we need to move to the location of the bottom
       object
31     bottom_location = get_obj_location(obj_at_bottom)
```

```
32      if get_curr_location() != bottom_location:
33          move(get_curr_location(), bottom_location)
34      unstack(obj_to_unstack, obj_at_bottom)
35      if get_curr_location() != unstack_location:
36          move(get_curr_location(), unstack_location)
37
38  def move_then_pick(obj):
39      obj_location = get_obj_location(obj)
40      if get_curr_location() != obj_location:
41          move(get_curr_location(), obj_location)
42      pick_up(obj, obj_location)
43
44  def move_then_cut(obj_to_cut, cut_location):
45      obj_location = get_obj_location(obj_to_cut)
46      if get_curr_location() != obj_location:
47          move(get_curr_location(), obj_location)
48      while not is_cut(obj_to_cut):
49          if get_curr_location() != cut_location:
50              move(get_curr_location(), cut_location)
51          cut(obj_to_cut)
```

### 14.3.3  Why 2-Layers (High) failed?

**2-Layers (High)** generated the same initial high level code as `Demo2Code`. However, the definition of `cut_object_at_location` in the recursive step is not exactly correct because the LLM is not taking account of all the physical constraints. It again assumes that the object to cut is already on the cutting board to cut. It moves to the object's location before starting to cut the object (line 18). Although **2-Layers (High)** has 0.2 unit test success rate, it achieved this because many of the lettuces happened to be placed on cutting boards in the randomly initialized environments.

```
1  # Get a list of all the lettuces in the kitchen.
2  lettuces = get_all_obj_names_that_match_type('lettuce')
3  # Get a list of all the cutting boards in the kitchen.
4  cutting_boards = get_all_location_names_that_match_type('cutting_board
       ')
5
6  # Repeat the following twice:
7  for i in range(2):
8      # Decide a lettuce to cut.
9      lettuce_to_cut = lettuces[i]
10     # Decide a cutting board to use.
11     cutting_board_to_use = cutting_boards[i]
12     # Cut that lettuce at that cutting board.
13     cut_object_at_location(obj=lettuce_to_cut, location=
       cutting_board_to_use)
14
15 def cut_object_at_location(obj, location):
16     # To cut an object, the robot first needs to be at the same
       location as obj
17     if get_curr_location() != get_obj_location(obj):
18         move(get_curr_location(), get_obj_location(obj))
19     # cut the object
20     while not is_cut(obj):
21         cut(obj)
```

### 14.3.4  Demo2Code Output

`Demo2Code` is able to adhere to physical constraints and correctly defines all the helper function in its recursive expansion steps.

```
1  def cut_object_at_location(obj, location):
2      # To cut an object, the robot first needs to be holding obj
3      if not is_holding(obj):
4          # If the robot is not holding obj, there are 2 scenarios:
```

```
5            #   (1) if obj is in a stack , unstack obj
6            #   (2) else , pick up obj.
7            if is_in_a_stack ( obj ):
8                # Because obj is in a stack , robot need to move then
     unstack the obj from the obj_at_bottom first
9                obj_at_bottom = get_obj_that_is_underneath ( obj_at_top = obj )
10               move_then_unstack ( obj_to_unstack = obj , obj_at_bottom =
     obj_at_bottom , unstack_location = get_obj_location ( obj_at_bottom ))
11           else :
12               # Since obj is not in a stack , robot can just move then
     pick it up
13               move_then_pick ( obj = obj )
14           # place the object at the location to cut at
15           move_then_place ( obj = obj , place_location = location )
16           # cut the object
17           cut_until_is_cut ( obj = obj )
18
19 def cut_until_is_cut ( obj ):
20     while not is_cut ( obj ):
21         cut ( obj )
22
23 def move_then_unstack ( obj_to_unstack , obj_at_bottom , unstack_location )
     :
24     # For unstacking , we need to move to the location of the bottom
     object
25     if get_curr_location () != get_obj_location ( obj_at_bottom ):
26         move ( get_curr_location () , get_obj_location ( obj_at_bottom ))
27     unstack ( obj_to_unstack , obj_at_bottom )
28     # After unstacking , we need to move to the unstack_location
29     if get_curr_location () != unstack_location :
30         move ( get_curr_location () , unstack_location )
31
32 def move_then_pick ( obj ):
33     obj_location = get_obj_location ( obj )
34     if get_curr_location () != obj_location :
35         move ( get_curr_location () , obj_location )
36     pick_up ( obj , obj_location )
37
38 def move_then_place ( obj , place_location ):
39     if get_curr_location () != place_location :
40         move ( get_curr_location () , place_location )
41     place ( obj , place_location )
```

## 15 Broader Impact

Our approach is a step towards making collaborative robots more accessible in different settings, such as homes, factories, and logistics operations. The broader implications of our research are manifold, touching upon societal, economic, ethical, technical, and educational aspects.

**Societal Impacts:** Currently, robots can only be programmed by engineers, limiting the tasks the robot can do to design choices made by engineers. This limits the accessibility for everyday users who require *personalization* to their individual needs and use cases. Our work tackles this problem head-on by allowing robots to be easily programmable via intuitive, user-friendly interfaces like vision and language. This could lead to increased creativity in the types of tasks that robots can be used for, potentially enabling novel applications and solutions.

**Economic Impacts:** Our approach has the potential to dramatically decrease the cost of programming robots, thus lowering the barriers to entry for businesses interested in incorporating robotics into their workflows. This could boost productivity and efficiency across a variety of industries, such as manufacturing, agriculture, and healthcare, among others. In the long term, this may contribute to economic growth and job creation in sectors that are currently behind on automation.

**Ethical and Legal Impacts:** Our approach has many ethical and legal considerations that need to be addressed. For instance, the widespread use of automation may lead to job displacement in certain sectors. Furthermore, it's crucial to ensure that the task code generated from demonstrations respects privacy laws and does not inadvertently encode biased or discriminatory behavior. Future work in this area will require close collaboration with ethicists, policymakers, and legal experts to navigate these issues.

**Technical Impacts:** Our work has the potential to accelerate the development of more intuitive and efficient human-robot interaction paradigms. For instance, demonstrations can be extended to interventions and corrections of the user. On the algorithmic side, our approach is a first step to connecting LLMs to Inverse Reinforcement Learning, and can spur advances in the fields of imitation learning, reinforcement learning, and natural language processing.

**Educational Impacts:** Lastly, our work could contribute to educational initiatives. The ability to generate task code from demonstrations could be utilized as an effective teaching tool in schools and universities, promoting a more experiential and intuitive approach to learning about robotics and coding. This could inspire and enable more students to pursue careers in STEM fields.

## 16 Reproducibility

We ran all our experiments using GPT-3.5 (`gpt-3.5-turbo`) with temperature 0. We have provided our codebase in https://github.com/portal-cornell/demo2code and all the prompts in section 18. The queries used to produce our results are available in the code base. Note that although we set the temperature to 0, which will make the LLM output mostly deterministic, the output might still slightly vary even for identical input.

## 17 `Demo2Code` Example Output

We provide an example for each domain and explain `Demo2Code`'s intermediate and final output.

### 17.1 Tabletop Simulator Example

This section shows an example of how `Demo2Code` solves the task: stacking all cylinders into one stack. The hidden user preference is that certain objects might need to have a fixed stack order, while other objects can be unordered. For this example, two objects ('blue cylinder' and 'purple cylinder') must follow the order that 'blue cylinder' should always be directly under the 'purple cylinder.'

#### 17.1.1 Query

The query has 3 components: (1) a list of objects that are in the environment, (2) a language instruction describing the goal of the task, and (3) two demonstrations.

```
objects=['pink block', 'yellow block', 'purple cylinder', 'cyan
    cylinder', 'pink cylinder', 'blue cylinder']
"""
[Scenario 1]
Stack all cylinders into one stack, while enforcing the order between
    cylinders if there is a requirement.

State 2:
'blue cylinder' has moved

State 3:
'purple cylinder' is not on top of 'table'
'purple cylinder' has moved
'purple cylinder' is on top of 'blue cylinder'

State 4:
'cyan cylinder' is not on top of 'table'
'cyan cylinder' has moved
```

```
'cyan cylinder' is on top of 'purple cylinder'
'cyan cylinder' is on top of 'blue cylinder'

State 5:
'pink cylinder' is not on top of 'table'
'pink cylinder' has moved
'pink cylinder' is on top of 'purple cylinder'
'pink cylinder' is on top of 'cyan cylinder'
'pink cylinder' is on top of 'blue cylinder'

[Scenario 2]
Stack all cylinders into one stack, while enforcing the order between
    cylinders if there is a requirement.

State 2:
'cyan cylinder' is not on top of 'table'
'cyan cylinder' has moved
'cyan cylinder' is on top of 'pink cylinder'

State 3:
'blue cylinder' is not on top of 'table'
'blue cylinder' has moved
'blue cylinder' is on top of 'cyan cylinder'
'blue cylinder' is on top of 'pink cylinder'

State 4:
'purple cylinder' is not on top of 'table'
'purple cylinder' has moved
'purple cylinder' is on top of 'cyan cylinder'
'purple cylinder' is on top of 'pink cylinder'
'purple cylinder' is on top of 'blue cylinder'
"""
```

### 17.1.2 Intermediate output

In stage 1 recursive summarization, for each demonstration, Demo2Code summarized the trajectories into low-level actions (e.g. moved 'blue cylinder', picked up and placed 'red cylinder' on top of 'blue cylinder'). Then, because tabletop's domain has short demonstrations, the LLM determined that a low-level action trajectory was sufficiently summarized.

Then, Demo2Code concatenates all summarized trajectories together to identify the user's preference and generate an appropriate task specification. Specifically, the LLM first identified the order of the stack in each scenario. Then, it compared the two orders in the two scenarios before concluding that the order that stays the same is ['blue cylinder', 'purple cylinder']. It finally produced the correct specification that will get passed to stage 2 to produce the correct code.

**Stage 1 Recursive Summary: recursive call # 1 with state-based trajectories as input**

```
===========================summarizing demo===========================
[[Reasoning:]]
The input trajectory contains state predicates because the trajectory
    talks about the status of the objects ('red cylinder' is not on
    top of 'table').
I will summarize the state trajectory into low-level actions.
The low-level actions are the most compact form, so I should stop
    summarizing after this.
The new trajectory will be sufficiently summarized.
[[Is the new trajectory sufficiently summarized? (yes/no):]]
yes
[[Summarized trajectory:]]
* In [Scenario 1], at state 2, the robot moved 'blue cylinder'.
* At state 3, the robot picked up and placed 'red cylinder' on top of
    'blue cylinder'.
```

```
* At state 4, the robot picked up and placed 'orange cylinder' on top
    of 'red cylinder'.
* At state 5, the robot picked up and placed 'cyan cylinder' on top of
     'orange cylinder'.

==========================summarizing demo==========================
[[Reasoning:]]
The input trajectory contains state predicates because the trajectory
    talks about the status of the objects ('orange cylinder' has moved
    ).
I will summarize the state trajectory into low-level actions.
The low-level actions are the most compact form, so I should stop
    summarizing after this.
The new trajectory will be sufficiently summarized.
[[Is the new trajectory sufficiently summarized? (yes/no):]]
yes
[[Summarized trajectory:]]
* In [Scenario 2], at state 2, the robot moved 'orange cylinder'.
* At state 3, the robot picked up and placed 'blue cylinder' on top of
     'orange cylinder'.
* At state 4, the robot picked up and placed 'red cylinder' on top of
    'blue cylinder'.
* At state 5, the robot picked up and placed 'cyan cylinder' on top of
     'red cylinder'.
```

**Summarized Demonstrations -> Task Specification**

```
[[Reasoning:]]
Because the task is "Stack all cylinders into one stack, while
    enforcing the order...", we focus on whether there is any ordering
     requirement.

In [Scenario 1], based on States 2-5, in the first stack, the
    cylinders from bottom to top is ['blue cylinder', 'red cylinder',
    'orange cylinder', 'cyan cylinder'].
In [Scenario 2], based on States 2-5, in the first stack, the
    cylinders from bottom to top is ['orange cylinder', 'blue cylinder
    ', 'red cylinder', 'cyan cylinder'].

Between the order of cylinders in these two scenarios, the ordering
    that stays the same is ['blue cylinder', 'red cylinder', 'cyan
    cylinder'].
[[Task Specification:]]
Stack all cylinders into one stack. Also make sure that the stacking
    enforces the bottom to top order between these cylinders: ['blue
    cylinder', 'red cylinder', 'cyan cylinder'].
```

### 17.1.3   Output Code

This is the final output of `Demo2Code`, which is the task code to solve this problem. The code illustrates the LLM's ability to write complex code that has control flows (e.g. for loop and if-else statements.)

```
# Based on the instruction, first define the cylinders that have
    strict ordering
cylinder_names_with_strick_order_from_bottom_to_top = ['blue cylinder'
    , 'purple cylinder']
# Then, find the rest of the cylinders that don't have strict order
cylinder_names = get_all_obj_names_that_match_type(type_name="cylinder
    ", objects_list=get_obj_names())
cylinder_names_without_order = []
for cylinder in cylinder_names:
  if cylinder not in
    cylinder_names_with_strick_order_from_bottom_to_top:
```

```
      cylinder_names_without_order.append(cylinder)
# Call helper function to determine the final stacking order before
    stacking the cylinders
stack_order_from_bottom_to_top = determine_final_stacking_order(
    objects_to_enforce_order=
    cylinder_names_with_strick_order_from_bottom_to_top,
    objects_without_order=cylinder_names_without_order)
stack_without_height_limit(objects_to_stack=
    stack_order_from_bottom_to_top)
```

## 17.2  Robotouille **Example**

### 17.2.1  Query

The example we will consider is *Make a burger stacking patty atop lettuce immediately.* This is a task that contains many different actions and is successfully solved by Demo2Code, making it a comprehensive example. The query for this task consists of two demonstrations each of which has the same language instruction to make a burger. Because each demonstration has 25 states, we have placed the query at the end of the appendix in section 19.1.

### 17.2.2  Intermediate output

Below, in stage 1 recursive summary, for each demonstration, Demo2Code summarizes each trajectory from a state-based trajectory to a low-level action trajectory to a high-level subtask trajectory, before the LLM decides that the trajectory is sufficiently summarized.

Then, Demo2Code concatenates all summarized trajectories together to identify the user's preference and generate an appropriate task specification. Specifically, the LLM first identified the order of the subtasks in each scenario. Then, it determined that the lettuce needs to be cut and stacked on top of the bottom bun before cooking the patty and stacking the patty on top of the lettuce. It finally produced the correct specification that will get passed to stage 2 to produce the correct code.

**Stage 1 Recursive Summary: recursive call # 1 with state-based trajectories as input**

```
==========================summarizing demo==========================
[[Reasoning:]]
The input trajectory contains state predicates because the trajectory
    talks about the status of the robot ('robot1' is at 'table2') and
    the status of the objects ('lettuce1' is not at 'table2').
I will summarize the state trajectory into low-level actions.
Low-level actions are not compacted enough yet because low-level
    actions can still be combined into high-level subtasks.
The new trajectory will NOT be sufficiently summarized.
[[Is the new trajectory sufficiently summarized? (yes/no):]]
no
[[Summarized trajectory:]]
* In [Scenario 1], at state 2, the robot moved from 'table1' to '
    table2'.
* At state 3, the robot picked up 'lettuce1'.
* At state 4, the robot moved from 'table2' to 'cutting_board1'.
* At state 5, the robot placed 'lettuce1' at location 'cutting_board1
    '.
* At state 6-8, the robot had cut 'lettuce1'.
* At state 9, the robot picked up 'lettuce1'.
* At state 10, the robot moved from 'cutting_board1' to 'table3'.
* At state 11, the robot placed 'lettuce1' on top of 'bottom_bun1' at
    location 'table3'.
* At state 12, the robot moved from 'table3' to 'stove1'.
* At state 13, the robot picked up 'patty1'.
* At state 14, the robot moved from 'stove1' to 'stove4'.
* At state 15, the robot placed 'patty1' at 'stove4'.
* At state 16-19, the robot had cooked 'patty1'.
* At state 20, the robot picked up 'patty1'.
```

```
* At state 21, the robot moved from 'stove4' to 'table3'.
* At state 22, the robot placed 'patty1' on top of 'lettuce1' at
    location 'table3'.
* At state 23, the robot moved from 'table3' to 'table4'.
* At state 24, the robot picked up 'top_bun1'.
* At state 25, the robot moved from 'table4' to 'table3'.
* At state 26, the robot placed 'top_bun1' on top of 'patty1' at
    location 'table3'.

============================summarizing demo============================
[[Reasoning:]]
The input trajectory contains state predicates because the trajectory
    talks about the status of the robot ('robot1' is at '
    cutting_board5') and the status of the objects ('lettuce3' is not
    at 'cutting_board5').
I will summarize the state trajectory into low-level actions.
Low-level actions are not compacted enough yet because low-level
    actions can still be combined into high-level subtasks.
The new trajectory will NOT be sufficiently summarized.
[[Is the new trajectory sufficiently summarized? (yes/no):]]
no
[[Summarized trajectory:]]
* In [Scenario 2], at state 2, the robot moved from 'table7' to '
    cutting_board5'.
* At state 3, the robot picked up 'lettuce3'.
* At state 4, the robot moved from 'cutting_board5' to 'cutting_board3
    '.
* At state 5, the robot placed 'lettuce3' at location 'cutting_board3
    '.
* At state 6-8, the robot had cut 'lettuce3'.
* At state 9, the robot picked up 'lettuce3'.
* At state 10, the robot moved from 'cutting_board3' to 'table4'.
* At state 11, the robot placed 'lettuce3' on top of 'bottom_bun3' at
    location 'table4'.
* At state 12, the robot moved from 'table4' to 'table7'.
* At state 13, the robot picked up 'patty3'.
* At state 14, the robot moved from 'table7' to 'stove3'.
* At state 15, the robot placed 'patty3' on 'stove3'.
* At state 16-19, the robot had cooked 'patty3'.
* At state 20, the robot picked up 'patty3'.
* At state 21, the robot moved from 'stove3' to 'table4'.
* At state 22, the robot placed 'patty3' on top of 'lettuce3' at
    location 'table4'.
* At state 23, the robot moved from 'table4' to 'cutting_board4'.
* At state 24, the robot picked up 'top_bun3'.
* At state 25, the robot moved from 'cutting_board4' to 'table4'.
* At state 26, the robot placed 'top_bun3' on top of 'patty3' at
    location 'table4'.
```

**Stage 1 Recursive Summary: recursive call # 2 with low-level action trajectories as input**

```
============================summarizing demo============================
[[Reasoning:]]
The input trajectory contains low-level actions because the trajectory
     mentions "moved", "picked up", etc.
I will summarize the low-level action trajectory into high-level
    subtasks.
High-level subtask trajectory is the most compacted form that cannot
    be summarized anymore.
The new trajectory will be sufficiently summarized.
[[Is the new trajectory sufficiently summarized? (yes/no):]]
yes
[[Summarized trajectory:]]
* In [Scenario 1], at state 2-8, the subtask is "cut lettuce". This
    subtask contains: 1. moving from 'table1' to 'cutting_board1' (
```

state 2-4) 2. placing 'lettuce1' at 'cutting_board1' (state 5) 3.
cutting 'lettuce1' until it is cut (state 6-8)
* At state 9-11, the subtask is "stack lettuce on top of bottom bun".
This subtask contains: 1. picking up 'lettuce1' (state 9) 2.
moving to stack 'lettuce1' on 'bottom_bun1' (state 10-11)
* At state 12-19, the subtask is "cook patty". This subtask contains:
1. moving from 'stove1' to 'stove4' (state 12-14) 2. placing '
patty1' at 'stove4' (state 15) 3. cooking 'patty1' until it is
cooked (state 16-19)
* At state 20-22, the subtask is "stack patty on top of lettuce". This
subtask contains: 1. picking up 'patty1' (state 20) 2. moving to
stack 'patty1' on 'lettuce1' (state 21-22)
* At state 23-26, the subtask is "stack top bun on top of patty". This
subtask contains: 1. moving from 'table3' to 'table4' (state 23)
2. picking up 'top_bun1' (state 24) 3. moving to stack 'top_bun1'
on 'patty1' (state 25-26)

==========================summarizing demo==========================
[[Reasoning:]]
The input trajectory contains low-level actions because the trajectory
mentions "moved", "picked up", etc.
I will summarize the low-level action trajectory into high-level
subtasks.
High-level subtask trajectory is the most compacted form that cannot
be summarized anymore.
The new trajectory will be sufficiently summarized.
[[Is the new trajectory sufficiently summarized? (yes/no):]]
yes
[[Summarized trajectory:]]
* In [Scenario 2], at state 2-8, the subtask is "cut lettuce". This
subtask contains: 1. moving to pick up 'lettuce3' (state 2-3) 2.
moving to place 'lettuce3' on 'cutting_board3' (state 4-5) 3.
cutting 'lettuce3' until it is cut (state 6-8)
* At state 9-11, the subtask is "stack lettuce on top of bottom bun".
This subtask contains: 1. picking up 'lettuce3' (state 9) 2.
moving to stack 'lettuce3' on 'bottom_bun3' (state 10-11)
* At state 12-19, the subtask is "cook patty". This subtask contains:
1. moving to pick up 'patty3' (state 12-13) 2. moving to place '
patty3' on 'stove3' (state 14-15) 3. cooking 'patty3' until it is
cooked (state 16-19)
* At state 20-22, the subtask is "stack patty on top of lettuce". This
subtask contains: 1. picking up 'patty3' (state 20) 2. moving to
stack 'patty3' on 'lettuce3' (state 21-22)
* At state 23-26, the subtask is "stack top bun on top of patty". This
subtask contains: 1. moving to pick up 'top_bun3' (state 23-24)
2. moving to stack 'top_bun3' on 'patty3' (state 25-26)

**Summarized Demonstrations -> Task Specification**

[[Reasoning:]]
* In [Scenario 1]'s unique kitchen environment, the subtasks were
executed in this order: ['cut lettuce', 'stack lettuce on top of
bottom bun', 'cook patty', 'stack patty on top of lettuce', 'stack
top bun on top of patty'].
* In [Scenario 2]'s unique kitchen environment, the subtasks were
executed in this order: ['cut lettuce', 'stack lettuce on top of
bottom bun', 'cook patty', 'stack patty on top of lettuce', 'stack
top bun on top of patty'].
* Both scenarios are executing the subtasks in the same order.
* There is no repetition or loop in the subtask ordering for any
individual scenario.
[[Task Specification:]]
Make a burger.

```
Specifically:
Get a list of all the lettuces in the kitchen.
Get a list of all the bottom buns in the kitchen.
Get a list of all the patties in the kitchen.
Get a list of all the top buns in the kitchen.

Decide a lettuce to use.
Get a list of all the available cutting boards in the kitchen.
Decide a cutting board to go to.
Cut that lettuce at that cutting board.

Decide a bottom bun to use.
Stack the lettuce on top of the bottom bun.

Decide a patty to use.
Get a list of all the available stoves in the kitchen.
Decide a stove to go to.
Cook that patty at that stove.

Stack the patty on top of the lettuce.

Decide a top bun to use.
Stack the top bun on top of the patty.
```

### 17.2.3  Output Code

This is the final output of the LLM, which is the task code to solve this problem. The code illustrates the LLM's ability to write complex code that has control flows (e.g. for loop and if-else statements.)

**Task Specification -> High-Level Code**

```
# Get a list of all the lettuces in the kitchen.
lettuces = get_all_obj_names_that_match_type('lettuce')
# Get a list of all the bottom buns in the kitchen.
bottom_buns = get_all_obj_names_that_match_type('bottom_bun')
# Get a list of all the patties in the kitchen.
patties = get_all_obj_names_that_match_type('patty')
# Get a list of all the top buns in the kitchen.
top_buns = get_all_obj_names_that_match_type('top_bun')

# Decide a lettuce to use.
lettuce_to_use = lettuces[0]
# Get a list of all the available cutting boards in the kitchen.
cutting_boards = get_all_location_names_that_match_type('cutting_board
    ')
# Decide a cutting board to go to.
cutting_board_to_cut_at = cutting_boards[0]
# Cut that lettuce at that cutting board.
cut_object_at_location(obj=lettuce_to_use, location=
    cutting_board_to_cut_at)

# Decide a bottom bun to use.
bottom_bun_to_use = bottom_buns[0]
# Stack the lettuce on top of the bottom bun.
# obj1 should be the lettuce, obj2 should be the bottom bun.
stack_obj1_on_obj2(obj1=lettuce_to_use, obj2=bottom_bun_to_use)

# Decide a patty to use.
patty_to_use = patties[0]
# Get a list of all the available stoves in the kitchen.
stoves = get_all_location_names_that_match_type('stove')
# Decide a stove to go to.
stove_to_cook_at = stoves[0]
# Cook that patty at that stove.
```

```python
cook_object_at_location(obj=patty_to_use, location=stove_to_cook_at)

# Stack the patty on top of the lettuce.
# obj1 should be the patty, obj2 should be the lettuce.
stack_obj1_on_obj2(obj1=patty_to_use, obj2=lettuce_to_use)

# Decide a top bun to use.
top_bun_to_use = top_buns[0]
# Stack the top bun on top of the patty.
# obj1 should be the top bun, obj2 should be the patty.
stack_obj1_on_obj2(obj1=top_bun_to_use, obj2=patty_to_use)
```

**Step 2: Define composite actions**

```python
def cook_object_at_location(obj, location):
    # To cook an object, the robot first needs to be holding obj
    if not is_holding(obj):
        # If the robot is not holding obj, there are 2 scenarios:
        #    (1) if obj is in a stack ,unstack obj
        #    (2) else, pick up obj.
        if is_in_a_stack(obj):
            # Because obj is in a stack, robot need to move then
    unstack the obj from the obj_at_bottom first
            obj_at_bottom = get_obj_that_is_underneath(obj_at_top=obj)
            # move then unstack: first you move to the obj_at_bottom's
     location, then you unstack obj from obj_at_bottom
            move_then_unstack(obj_to_unstack=obj, obj_at_bottom=
    obj_at_bottom, unstack_location=get_obj_location(obj_at_bottom))
        else:
            # move_then_pick: first you move to obj's location, then
    you pick obj up
            move_then_pick(obj=obj, pick_location=get_obj_location(obj
    ))
    # move then place: first you move to the location to cook at, then
     you place obj at that location
    move_then_place(obj=obj, place_location=location)
    # cook the object until it is cooked
    cook_until_is_cooked(obj=obj)

def stack_obj1_on_obj2(obj1, obj2):
    # To stack obj1 on obj2, the robot needs to be holding obj1
    if not is_holding(obj1):
        # If the robot is not holding obj1, there are 2 scenarios:
        #    (1) if obj1 is in a stack ,unstack obj1
        #    (2) else, pick up obj1.
        if is_in_a_stack(obj1):
            # Because obj1 is in a stack, robot need to move then
    unstack the obj from the obj_at_bottom first
            obj_at_bottom = get_obj_that_is_underneath(obj_at_top=obj1
    )
            # move then unstack: first you move to the obj_at_bottom's
     location, then you unstack obj from obj_at_bottom
            move_then_unstack(obj_to_unstack=obj1, obj_at_bottom=
    obj_at_bottom, unstack_location=get_obj_location(obj_at_bottom))
        else:
            # move_then_pick: first you move to obj's location, then
    you pick obj up
            move_then_pick(obj=obj1, pick_location=get_obj_location(
    obj1))
    # determine the location of obj2 to stack on
    obj2_location = get_obj_location(obj2)
    # move then stack: first you move to obj2's location, then you
    stack obj1 on obj2
    move_then_stack(obj_to_stack=obj1, obj_at_bottom=obj2,
    stack_location=obj2_location)
```

```python
def cut_object_at_location(obj, location):
    # To cut an object, the robot first needs to be holding obj
    if not is_holding(obj):
        # If the robot is not holding obj, there are 2 scenarios:
        #    (1) if obj is in a stack ,unstack obj
        #    (2) else, pick up obj.
        if is_in_a_stack(obj):
            # Because obj is in a stack, robot need to move then
    unstack the obj from the obj_at_bottom first
            obj_at_bottom = get_obj_that_is_underneath(obj_at_top=obj)
            # move then unstack: first you move to the obj_at_bottom's
     location, then you unstack obj from obj_at_bottom
            move_then_unstack(obj_to_unstack=obj, obj_at_bottom=
    obj_at_bottom, unstack_location=get_obj_location(obj_at_bottom))
        else:
            # move_then_pick: first you move to obj's location, then
    you pick obj up
            move_then_pick(obj=obj, pick_location=get_obj_location(obj
    ))
    # move then place: first you move to the location to cut at, then
    you place obj at that location
    move_then_place(obj=obj, place_location=location)
    # cut the object until it is cut
    cut_until_is_cut(obj=obj)
```

**Step 3: Define low-level actions**

```python
def move_then_unstack(obj_to_unstack, obj_at_bottom, unstack_location)
    :
    if get_curr_location() != unstack_location:
        move(get_curr_location(), unstack_location)
    unstack(obj_to_unstack, obj_at_bottom)

def move_then_pick(obj, pick_location):
    if get_curr_location() != pick_location:
        move(get_curr_location(), pick_location)
    pick_up(obj, pick_location)

def move_then_place(obj, place_location):
    if get_curr_location() != place_location:
        move(get_curr_location(), place_location)
    place(obj, place_location)

def cook_until_is_cooked(obj):
    start_cooking(obj)
    while not is_cooked(obj):
        noop()

def move_then_stack(obj_to_stack, obj_at_bottom, stack_location):
    if get_curr_location() != stack_location:
        move(get_curr_location(), stack_location)
    stack(obj_to_stack, obj_at_bottom)

def cut_until_is_cut(obj):
    while not is_cut(obj):
        cut(obj)
```

### 17.3 `EPIC-Kitchens` Example

#### 17.3.1 Query

We present an example of the demonstration given as a query to the LLM. The annotations are made for video id $P22\_07$. This example shows the LLM generalizing to multiple new objects, new

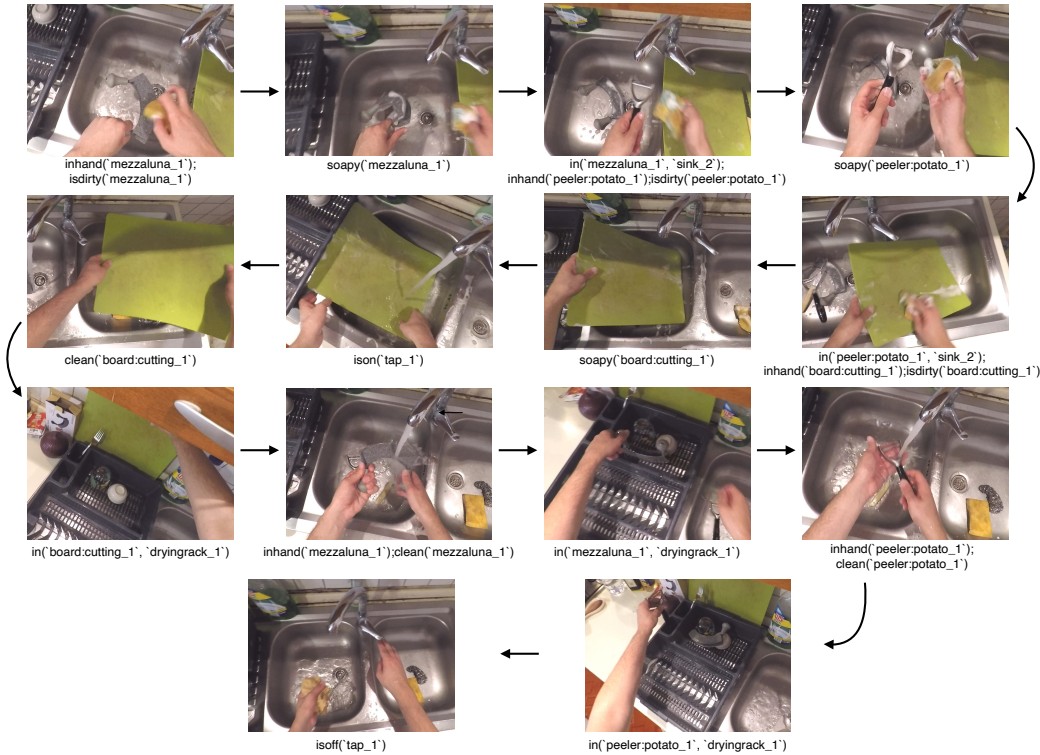

Figure 14: Example of annotations for video P22_07 in 14 frames

locations, and a different dishwashing style. Figure 14 shows visualizations of the video with the respective state annotations. Because each demonstration has 25 states, we have placed the query at the end of the appendix in section 19.2.

### 17.3.2 Intermediate Output

Below, in stage 1 recursive summary, for each demonstration, `Demo2Code` summarizes each trajectory from a state-based trajectory to a low-level action trajectory to a high-level subtask trajectory, before the LLM decides that the trajectory is sufficiently summarized.

Then, `Demo2Code` concatenates all summarized trajectories together to identify the user's preference and generate an appropriate task specification. Specifically, the LLM was able to identify that it should use a for-loop because [`pick up, soap, move object from A to B`] are repeated. It finally produced the correct specification that will get passed to stage 2 to produce the correct code.

**Stage 1 Recursive Summary: recursive call # 1 with state-based trajectories as input**

```
===========================summarizing demo===========================
[[Reasoning:]]
The input trajectory contains state predicates because the trajectory
    talks about the status of the objects (is_in_hand(`sponge_1`),
    dirty(`mezzaluna_1`), etc.).
I will summarize the state trajectory into low-level actions.
Low-level actions are not compacted enough yet because low-level
    actions can still be combined into high-level subtasks.
The new trajectory will NOT be sufficiently summarized.
[[Is the new trajectory sufficiently summarized? (yes/no):]]
no
[[Summarized trajectory:]]
* In [Scenario 1], at state 1-2, the robot picked up mezzaluna_1 from
    sink_1.
* At state 2-3, the robot soaped mezzaluna_1 until it became soapy.
```

```
* At state 3-4, the robot moved mezzaluna_1 from sink_1 to sink_2.
* At state 4-5, the robot moved peeler:potato_1 from sink_1 to sink_2.
* At state 5-6, the robot picked up peeler:potato_1.
* At state 6-7, the robot soaped peeler:potato_1 until it became soapy
   .
* At state 7-8, the robot moved peeler:potato_1 from sink_1 to sink_2.
* At state 8-9, the robot moved knife_1 from sink_1 to sink_2.
* At state 9-10, the robot picked up knife_1.
* At state 10-11, the robot soaped knife_1 until it became soapy.
* At state 11-12, the robot moved knife_1 from sink_1 to sink_2.
* At state 12-13, the robot moved board:cutting_1 from sink_1 to
   sink_2.
* At state 13-14, the robot picked up board:cutting_1.
* At state 14-15, the robot soaped board:cutting_1 until it became
   soapy.
* At state 15-16, the robot moved board:cutting_1 from sink_1 to
   sink_2.
* At state 16-17, the robot turned on tap_1.
* At state 17-18, the robot picked up board:cutting_1.
* At state 18-19, the robot rinsed board:cutting_1 until it became
   clean.
* At state 19-20, the robot placed board:cutting_1 in dryingrack_1.
* At state 20-21, the robot picked up mezzaluna_1.
* At state 21-22, the robot rinsed mezzaluna_1 until it became clean.
* At state 22-23, the robot placed mezzaluna_1 in dryingrack_1.
* At state 23-24, the robot picked up peeler:potato_1.
* At state 24-25, the robot rinsed peeler:potato_1 until it became
   clean.
* At state 25-26, the robot placed peeler:potato_1 in dryingrack_1.
* At state 26-27, the robot picked up knife_1.
* At state 27-28, the robot rinsed knife_1 until it became clean.
* At state 28-29, the robot placed knife_1 in dryingrack_1.
* At state 29-30, the robot turned off tap_1.
```

**Stage 1 Recursive Summary: recursive call # 2 with low-level action trajectories as input**

```
==========================summarizing demo==========================
[[Reasoning:]]
The input trajectory contains low-level actions because the trajectory
    mentions "picked up", "moved", "soaped", "rinsed", etc.
I will summarize the low-level action trajectory into high-level
   subtasks.
High-level subtasks are compacted enough because they cannot be
   combined together without losing important information.
The new trajectory will be sufficiently summarized.
[[Is the new trajectory sufficiently summarized? (yes/no):]]
yes
[[Summarized trajectory:]]
* In [Scenario 1], at state 1-3, the robot brought mezzaluna_1 from
   sink_1 to sink_2.
* At state 3-4, the robot brought peeler:potato_1 from sink_1 to
   sink_2.
* At state 4-5, the robot picked up peeler:potato_1.
* At state 5-6, the robot soaped peeler:potato_1 until it became soapy
   .
* At state 6-7, the robot moved peeler:potato_1 from sink_1 to sink_2.
* At state 7-8, the robot moved knife_1 from sink_1 to sink_2.
* At state 8-9, the robot picked up knife_1.
* At state 9-10, the robot soaped knife_1 until it became soapy.
* At state 10-11, the robot moved knife_1 from sink_1 to sink_2.
* At state 11-12, the robot moved board:cutting_1 from sink_1 to
   sink_2.
* At state 12-13, the robot picked up board:cutting_1.
* At state 13-14, the robot soaped board:cutting_1 until it became
   soapy.
```

* At state 14-15, the robot moved board:cutting_1 from sink_1 to
    sink_2.
* At state 15-16, the robot turned on tap_1.
* At state 16-19, the robot rinsed board:cutting_1, mezzaluna_1,
    peeler:potato_1, and knife_1 until they became clean.
* At state 19-20, the robot placed board:cutting_1 in dryingrack_1.
* At state 20-21, the robot picked up mezzaluna_1.
* At state 21-22, the robot rinsed mezzaluna_1 until it became clean.
* At state 22-23, the robot placed mezzaluna_1 in dryingrack_1.
* At state 23-24, the robot picked up peeler:potato_1.
* At state 24-25, the robot rinsed peeler:potato_1 until it became
    clean.
* At state 25-26, the robot placed peeler:potato_1 in dryingrack_1.
* At state 26-27, the robot picked up knife_1.
* At state 27-28, the robot rinsed knife_1 until it became clean.
* At state 28-29, the robot placed knife_1 in dryingrack_1.
* At state 29-30, the robot turned off tap_1.

**Summarized Demonstrations -> Task Specification**

```
============================summary -> spec============================
[[Reasoning:]]
* There are 4 objects that need to be washed: [mezzaluna_1, peeler:
    potato_1, knife_1, board:cutting_1]
* The list of high-level actions that happened in order is: [move
    object from A to B, pick up object, soap object, move object from
    A to B, turn on tap, rinse objects, place object in drying rack,
    turn off tap]
* The actions [pick up, soap, move object from A to B] are repeated
    for each object, so we can use a loop.
* Rinsing involves the use of tap water, which is why it is turned on
    at some point before the rinse cycle, and turned off after.
[[Task Specification:]]
Wash objects in sink. All clean objects must be placed in drying rack.
Specifically -

Get a list of all objects to wash
For each object in all objects:
  Move object from sink_1 to sink_2
  Pick up object
  Soap object
  Move object from sink_1 to sink_2
Turn on tap_1
For each object in all objects:
  Rinse object
  Place object in dryingrack_1
Turn off tap_1
```

### 17.3.3 Output Code

This is the final output of Demo2Code, which is the task code to solve this problem. The LLM
captures that the objects are soapped first, then rinsed one by one after. The code also illustrates the
LLM's ability to write complex code that has control flows (e.g. for loop and if-else statements.)

```
objects = get_all_objects()
for object in objects:
  bring_objects_to_loc([object], "sink_2")
  pick_up(object)
  soap(object)
  bring_objects_to_loc([object], "sink_1")
turn_on("tap_1")
for object in objects:
  rinse(object)
  place(object, "dryingrack_1")
```

```
turn_off("tap_1")
```

# 18 Prompts

All prompts are in yaml file representation. For the most updated prompt, please refer to our code base: https://portal-cornell.github.io/demo2code/.

## 18.1 Tabletop Manipulation Task Prompts

### 18.1.1 Recursive Summarization

```
recursive_summarization:
  main: |
    You are a helpful summarizer that recursively summarizes a
    trajectory into a more and more compacted form.
    <end_of_system_message>
    You are given a trajectory. Your goal is to summarize the
    trajectory into a more compacted form and then determine whether
    the state trajectory is sufficiently summarized.

    You must respond using the following form (you must generate [[end
     of response]] at the very end):
    [[Reasoning:]]
    You should first identify what type of trajectory this is. Then,
    you should determine what type you will summarize the trajectory
    into. Finally, you should determine whether the new type of
    trajectory is sufficiently summarized or not.
    [[Is the new trajectory sufficiently summarized? (yes/no):]]
    You should only respond using either "yes" or "no", and nothing
    else.
    [[Summarized trajectory:]]
    You should actually summarize the input trajectory and output it
    here.
    [[end of response]]

    You must follow these rules when you are summarizing a trajectory.
    Rules:
    * You must slowly summarize the trajectory from one type to
    another following this order: a state trajectory    > a low-level
    action trajectory.
    * A low-level action trajectory is represented as an unordered
    list. Each line in the unordered list should only contain one low-
    level action.
    * The low-level actions must be one of the following: "move <item1
    >", "picked up and placed <item1> <relative_position_relation> <
    item2>", "picked up and placed <item1> at <loc1>", "picked up and
    placed <item1> on top of 'table'", "picked up and placed <item1>
    on top of <item2>". You should never define new low-level actions.
  examples:
    - |
      [[Input Trajectory:]]
      [Scenario 1]
      Place the green cylinder next to the yellow cylinder.

      State 2:
      'yellow cylinder' is to the right of 'green cylinder'
      'green cylinder' has moved
      'green cylinder' is to the left of 'yellow cylinder'
      <end_of_example_user_query>
      [[Reasoning:]]
      The input trajectory contains state predicates because the
    trajectory talks about the status of the objects ('yellow cylinder
    ' is to the right of 'green cylinder').
```

I will summarize the state trajectory into low-level actions.
The low-level actions are the most compact form, so I should
stop summarizing after this.
The new trajectory will be sufficiently summarized.
[[Is the new trajectory sufficiently summarized? (yes/no):]]
yes
[[Summarized trajectory:]]
* In [Scenario 1], at state 2, the robot picked up and placed '
green cylinder' to the left of 'yellow cylinder'.
[[end of response]]
- |
[[Input Trajectory:]]
[Scenario 1]
Place the purple cylinder next to the orange block.

State 2:
'orange block' is in front of 'purple cylinder'
'purple cylinder' has moved
'purple cylinder' is at 'top edge'
'purple cylinder' is behind 'orange block'
<end_of_example_user_query>
[[Reasoning:]]
The input trajectory contains state predicates because the
trajectory talks about the status of the objects ('orange block'
is in front of 'purple cylinder').
I will summarize the state trajectory into low-level actions.
The low-level actions are the most compact form, so I should
stop summarizing after this.
The new trajectory will be sufficiently summarized.
[[Is the new trajectory sufficiently summarized? (yes/no):]]
yes
[[Summarized trajectory:]]
* In [Scenario 1], at state 2, the robot picked up and placed '
purple cylinder' behind 'orange block'.
[[end of response]]
- |
[[Input Trajectory:]]
[Scenario 2]
Place the blue block at the corner.

State 2:
'blue block' has moved
'blue block' is at 'top right corner'
<end_of_example_user_query>
[[Reasoning:]]
The input trajectory contains state predicates because the
trajectory talks about the status of the objects ('blue block' has
 moved).
I will summarize the state trajectory into low-level actions.
The low-level actions are the most compact form, so I should
stop summarizing after this.
The new trajectory will be sufficiently summarized.
[[Is the new trajectory sufficiently summarized? (yes/no):]]
yes
[[Summarized trajectory:]]
* In [Scenario 1], at state 2, the robot picked up and placed '
blue block' at 'top right corner'.
[[end of response]]
- |
[[Input Trajectory:]]
[Scenario 1]
Place the cyan block on the blue block.

State 2:
'orange cylinder' is on top of 'table'

```
'orange cylinder' has moved
'orange cylinder' is not on top of 'red block'
'orange cylinder' is not on top of 'cyan block'
'orange cylinder' is not on top of 'yellow block'

State 3:
'yellow block' is on top of 'table'
'yellow block' has moved
'yellow block' is not on top of 'red block'
'yellow block' is not on top of 'cyan block'

State 4:
'red block' is on top of 'table'
'red block' has moved
'red block' is not on top of 'cyan block'
'red block' is to the right of 'orange cylinder'
'orange cylinder' is to the left of 'red block'

State 5:
'cyan block' is not on top of 'table'
'cyan block' has moved
'cyan block' is on top of 'blue block'
<end_of_example_user_query>
[[Reasoning:]]
The input trajectory contains state predicates because the
trajectory talks about the status of the objects ('orange cylinder
' is on top of 'table').
I will summarize the state trajectory into low-level actions.
The low-level actions are the most compact form, so I should
stop summarizing after this.
The new trajectory will be sufficiently summarized.
[[Is the new trajectory sufficiently summarized? (yes/no):]]
yes
[[Summarized trajectory:]]
* In [Scenario 1], at state 2, the robot picked up and placed
the 'orange cylinder' on top of 'table'.
* At state 3, the robot picked up and placed 'yellow block' on
top of 'table'.
* At state 4, the robot picked up and placed 'red block' on top
of 'table'.
* At state 5, the robot picked up and placed 'cyan block' on top
 of 'blue block'.
[[end of response]]
- |
[[Input Trajectory:]]
[Scenario 1]
Stack all blocks, where enfocing the maximum stack height if
there is a requirement.

State 2:
'orange block' has moved
'orange block' is not behind 'cyan cylinder'
'cyan cylinder' is not in front of 'orange block'

State 3:
'blue block' is not on top of 'table'
'blue block' has moved
'blue block' is on top of 'orange block'

State 4:
'brown block' is not on top of 'table'
'brown block' has moved
'brown block' is on top of 'orange block'
'brown block' is on top of 'blue block'
```

```
   State 5:
   'yellow block' has moved
   'yellow block' is behind 'cyan cylinder'
   'cyan cylinder' is in front of 'yellow block'

   State 6:
   'gray block' is not on top of 'table'
   'gray block' has moved
   'gray block' is on top of 'yellow block'
   <end_of_example_user_query>
   [[Reasoning:]]
   The input trajectory contains state predicates because the
trajectory talks about the status of the objects ('orange block'
has moved).
   I will summarize the state trajectory into low-level actions.
   The low-level actions are the most compact form, so I should
stop summarizing after this.
   The new trajectory will be sufficiently summarized.
   [[Is the new trajectory sufficiently summarized? (yes/no):]]
   yes
   [[Summarized trajectory:]]
   * In [Scenario 1], at state 2, the robot moved 'orange block'.
   * At state 3, the robot picked up and placed 'blue block' on top
 of 'orange block'.
   * At state 4, the robot picked up and placed 'brown block' on
top of 'blue block'.
   * At state 5, the robot moved 'yellow block'.
   * At state 6, the robot picked up and placed 'gray block' on top
 of 'yellow block'.
   [[end of response]]
- |
   [[Input Trajectory:]]
   [Scenario 2]
   Stack all blocks into one stack, while enforcing the order
between blocks if there is a requirement.

   State 2:
   'blue block' has moved

   State 3:
   'red block' is not on top of 'table'
   'red block' has moved
   'red block' is on top of 'blue block'
   'red block' is not to the right of 'red cylinder'
   'red cylinder' is not to the left of 'red block'

   State 4:
   'yellow block' is not on top of 'table'
   'yellow block' has moved
   'yellow block' is on top of 'red block'
   'yellow block' is on top of 'blue block'

   State 5:
   'cyan block' is not on top of 'table'
   'cyan block' has moved
   'cyan block' is on top of 'yellow block'
   'cyan block' is on top of 'red block'
   'cyan block' is on top of 'blue block'
   <end_of_example_user_query>
   [[Reasoning:]]
   The input trajectory contains state predicates because the
trajectory talks about the status of the objects ('pink block' has
 moved).
   I will summarize the state trajectory into low-level actions.
```

The low-level actions are the most compact form, so I should
stop summarizing after this.
   The new trajectory will be sufficiently summarized.
   [[Is the new trajectory sufficiently summarized? (yes/no):]]
   yes
   [[Summarized trajectory:]]
   * In [Scenario 2], at state 2, the robot moved 'blue block'.
   * At state 3, the robot picked up and placed 'red block' on top
of 'blue block'.
   * At state 4, the robot picked up and placed 'yellow block' on
top of 'red block'.
   * At state 5, the robot picked up and placed 'cyan block' on top
 of 'yellow block'.
   [[end of response]]
- |
   [[Input Trajectory:]]
   [Scenario 1]
   Stack all objects into two stacks, and objects may need to be
categorized in the stacks.

   State 2:
   'brown block' is to the left of 'brown cylinder'
   'brown cylinder' has moved
   'brown cylinder' is to the right of 'brown block'
   'brown cylinder' is to the left of 'blue cylinder'
   'blue cylinder' is to the right of 'brown cylinder'

   State 3:
   'orange block' is not on top of 'table'
   'orange block' has moved
   'orange block' is on top of 'brown cylinder'

   State 4:
   'yellow block' is not on top of 'table'
   'yellow block' has moved
   'yellow block' is on top of 'orange block'
   'yellow block' is on top of 'brown cylinder'

   State 5:
   'pink cylinder' is not on top of 'table'
   'pink cylinder' has moved
   'pink cylinder' is on top of 'yellow block'
   'pink cylinder' is on top of 'orange block'
   'pink cylinder' is on top of 'brown cylinder'

   State 6:
   'blue block' has moved

   State 7:
   'brown cylinder' is not to the left of 'blue cylinder'
   'blue cylinder' is not on top of 'table'
   'blue cylinder' has moved
   'blue cylinder' is on top of 'blue block'
   'blue cylinder' is not to the right of 'brown cylinder'

   State 8:
   'orange cylinder' is not on top of 'table'
   'orange cylinder' has moved
   'orange cylinder' is on top of 'blue block'
   'orange cylinder' is on top of 'blue cylinder'

   State 9:
   'brown block' is not on top of 'table'
   'brown block' has moved
   'brown block' is on top of 'blue block'

```
    'brown block' is not to the left of 'brown cylinder'
    'brown block' is on top of 'orange cylinder'
    'brown block' is on top of 'blue cylinder'
    'brown cylinder' is not to the right of 'brown block'
    <end_of_example_user_query>
    [[Reasoning:]]
    The input trajectory contains state predicates because the
 trajectory talks about the status of the objects ('brown block' is
 to the left of 'brown cylinder').
    I will summarize the state trajectory into low-level actions.
    The low-level actions are the most compact form, so I should
 stop summarizing after this.
    The new trajectory will be sufficiently summarized.
    [[Is the new trajectory sufficiently summarized? (yes/no):]]
    yes
    [[Summarized trajectory:]]
    * In [Scenario 1], at state 2, the robot moved 'brown cylinder'.
    * At state 3, the robot picked up and placed 'orange block' on
 top of 'brown cylinder'.
    * At state 4, the robot picked up and placed 'yellow block' on
 top of 'orange block'.
    * At state 5, the robot picked up and placed 'pink cylinder' on
 top of 'yellow block'.
    * At state 6, the robot moved 'blue block'.
    * At state 7, the robot picked up and placed 'blue cylinder' on
 top of 'blue block'.
    * At state 8, the robot picked up and placed 'orange cylinder'
 on top of 'blue cylinder'.
    * At state 9, the robot picked up and placed 'brown block' on
 top of 'orange cylinder'.
    [[end of response]]
```

## Summarized Demonstrations -> Task Specification

```
summary_2_spec:
  main: |
    You are a helpful assistant who analyzes the trajectories and
    summarizes them into a concise pseudocode-style task specification
    .
    <end_of_system_message>
    You are given (1) a high-level goal and (2) one or more
    trajectories where each one represents a different scenario. Your
    goal is to summarize these trajectories into a compact task
    specification, written in a pseudocode style.

    You must respond using the following format (you must generate [[
    end of response]] at the very end):
    [[Reasoning:]]
    You should first list out the order of the high-level subtask
    trajectories in all scenarios. Then, you should consider if
    certain sections of the subtasks are repeated in the scenario and
    can be represented by a loop. Two scenarios having exactly the
    same list does not mean that there is a loop. A loop only exists
    among subtasks in one individual scenario.
    Overall, your task specification should work for all scenarios.
    You should make sure that the task specification matches the
    subtasks ordering across all scenarios. You should also make sure
    that the task specification uses loops when there is any
    repetition.
    [[Task Specification:]]
    You should first state the high-level goal. Then, you should say "
    Specifically:" before outputting the pseudocode-style task
    specification.
```

```
[[end of response]]
```

You must follow these rules when you are writing the task
specifications.
* You must write the task specifications in pseudocode style. You
should not write the task specification as a list. You cannot
number any line.
examples:
- |
  ```
  [[High-Level Goal:]]
  ```
  Place the green cylinder next to the yellow cylinder.
  ```
  [[Trajectories:]]
  ```
  * In [Scenario 1], at state 2, the robot picked up and placed '
  green cylinder' to the left of 'yellow cylinder'.
  ```
  <end_of_example_user_query>
  [[Reasoning:]]
  ```
  With only 1 scenario, we assume that the 'green cylinder' must
  always be to the left of 'yellow cylinder'.
  ```
  [[Task Specification:]]
  ```
  Place the green cylinder to the left of the yellow cylinder.
  ```
  [[end of response]]
  ```
- |
  ```
  [[High-Level Goal:]]
  ```
  Place the purple cylinder next to the orange block.
  ```
  [[Trajectories:]]
  ```
  * In [Scenario 1], at state 2, the robot picked up and placed
  the 'purple cylinder' behind 'orange block'.
  * In [Scenario 2], at state 2, the robot picked up and placed
  the 'purple cylinder' in front of 'orange block'.
  ```
  <end_of_example_user_query>
  [[Reasoning:]]
  ```
  Because two scenarios have two different relative positions
  between 'purple cylinder' and 'orange block', we assume that the '
  purple cylinder' can be anywhere next to 'orange block'.
  ```
  [[Task Specification:]]
  ```
  Place the purple cylinder at anywhere next to the orange block.
  ```
  [[end of response]]
  ```
- |
  ```
  [[High-Level Goal:]]
  ```
  Place the blue block at the corner.
  ```
  [[Trajectories:]]
  ```
  * In [Scenario 1], at state 2, the robot picked up and placed
  the 'blue block' at 'bottom right corner'.
  * In [Scenario 2], at state 2, the robot picked up and placed
  the 'blue block' at 'top right corner'
  ```
  <end_of_example_user_query>
  [[Reasoning:]]
  ```
  Because two scenarios have two corners ('bottom right corner', '
  top right corner'), we assume that the 'blue block' can be at any
  random corner.
  ```
  [[Task Specification:]]
  ```
  Place the blue block at any random corner.
  ```
  [[end of response]]
  ```
- |
  ```
  [[High-Level Goal:]]
  ```
  Place the cyan block on the blue block.
  ```
  [[Trajectories:]]
  ```
  * In [Scenario 1], at state 2, the robot picked up and placed
  the 'orange cylinder' on top of 'table'.
  * At state 3, the robot picked up and placed 'yellow block' on
  top of 'table'.
  * At state 4, the robot picked up and placed 'red block' on top
  of 'table'.
  * At state 5, the robot picked up and placed 'cyan block' on top
  of 'blue block'.

```
<end_of_example_user_query>
[[Reasoning:]]
Although the goal is to place "cyan block" on the "blue block",
the trajectories show that it needs to move other blocks ('orange
cylinder', 'yellow block', 'red block') before finally placing '
cyan block' on top of 'blue block'.
[[Task Specification:]]
1. place the orange cylinder on the table
2. place the yellow block on the table
3. place the red block on the table
4. place the cyan block on the blue block
[[end of response]]
- |
[[High-Level Goal:]]
Stack all blocks, where enforcing the maximum stack height if
there is a requirement.
[[Trajectories:]]
* In [Scenario 1], at state 2, the robot moved 'orange block'.
* At state 3, the robot picked up and placed 'blue block' on top
of 'orange block'.
* At state 4, the robot picked up and placed 'brown block' on
top of 'blue block'.
* At state 5, the robot moved 'yellow block'.
* At state 6, the robot picked up and placed 'gray block' on top
of 'yellow block'.
<end_of_example_user_query>
[[Reasoning:]]
Because the task is "Stack all blocks, where enfocing the
maximum stack height...", we focus how high the stacks are.

Based on States 2-4, in the first stack, the blocks from bottom
to top is ['orange block', 'blue block', 'brown block']. This is 3
blocks high.
Based on States 5-6, in the second stack, the blocks from bottom
to top is ['yellow block', 'gray block']. This is 2 blocks high.

Because there are 2 stacks and the tallest stack is 3 block high
, we assume that each stack needs to be at most 3 blocks high.
[[Task Specification:]]
Stack all blocks. However, the maximum height of a stack is 3.
[[end of response]]
- |
[[High-Level Goal:]]
Stack all blocks into one stack, while enforcing the order
between blocks if there is a requirement.
[[Trajectories:]]
* In [Scenario 1], at state 2, the robot moved 'red block'.
* At state 3, the robot picked up and placed 'yellow block' on
top of 'red block'.
* At state 4, the robot picked up and placed 'cyan block' on top
of 'yellow block'.
* At state 5, the robot picked up and placed 'blue block' on top
of 'cyan block'.
* In [Scenario 2], at state 2, the robot moved 'blue block'.
* At state 3, the robot picked up and placed 'red block' on top
of 'blue block'.
* At state 4, the robot picked up and placed 'yellow block' on
top of 'red block'.
* At state 5, the robot picked up and placed 'cyan block' on top
of 'yellow block'.
<end_of_example_user_query>
[[Reasoning:]]
Because the task is "Stack all blocks, while enforcing the order
...", we focus on whether there is any ordering requirement.
```

In [Scenario 1], based on States 2-5, in the first stack, the
blocks from bottom to top is ['red block', 'yellow block', 'cyan
block', 'blue block'].
In [Scenario 2], based on States 2-5, in the first stack, the
blocks from bottom to top is ['blue block', 'red block', 'yellow
block', 'cyan block'].

Between the order of blocks in these two scenarios, the ordering
that stays the same is ['red block', 'yellow block', 'cyan block
'].
[[Task Specification:]]
Stack all blocks into one stack. Also make sure that the
stacking enforces the bottom to top order between these objects:
['red block', 'yellow block', 'cyan block'].
[[end of response]]
- |
[[High-Level Goal:]]
Stack all objects into two stacks, and objects may need to be
categorized in the stacks.
[[Trajectories:]]
* In [Scenario 1], at state 2, the robot moved 'brown cylinder'.
* At state 3, the robot picked up and placed 'orange block' on
top of 'brown cylinder'.
* At state 4, the robot picked up and placed 'yellow block' on
top of 'orange block'.
* At state 5, the robot picked up and placed 'pink cylinder' on
top of 'yellow block'.
* At state 6, the robot moved 'blue block'.
* At state 7, the robot picked up and placed 'blue cylinder' on
top of 'blue block'.
* At state 8, the robot picked up and placed 'orange cylinder'
on top of 'blue cylinder'.
* At state 9, the robot picked up and placed 'brown block' on
top of 'orange cylinder'.
<end_of_example_user_query>
[[Reasoning:]]
Because the task is "Stack all objects into two stacks, and
objects may need to be categorized in the stacks", we focus on
whether the objects are stacked by type.

Based on States 2-5, in the first stack, the blocks from bottom
to top is ['brown cylinder', 'orange block', 'yellow block', 'pink
cylinder'].
Based on States 6-9, in the first stack, the blocks from bottom
to top is ['blue block', 'blue cylinder', 'orange cylinder', '
brown block'].

Because each stack has both blocks and cylinders, we assume that
it doesn't matter whether the objects are categorized.
[[Task Specification:]]
Stack all objects into two stacks. It doesn't matter whether the
objects are categorized.
[[end of response]]

### 18.1.2 Recursive Expansion

**Task Specification -> High-Level Code**

```
spec_2_highlevelcode:
  main: |
    You are a Python code generator for robotics. The users will first
     provide the imported Python modules. Then, for each code they
    want you to generate, they provide the requirements and pseudocode
     in Python docstrings.
    <end_of_system_message>
```

You need to write robot control scripts in Python code. The Python code should be general and applicable to different environments.

Below are the imported Python libraries and functions that you can use. You CANNOT import new libraries.
```
# Python kitchen robot control script
import numpy as np
from robot_utils import put_first_on_second,
stack_without_height_limit, stack_with_height_limit
from env_utils import get_obj_names,
get_all_obj_names_that_match_type, determine_final_stacking_order,
 parse_position
ALL_CORNERS_LIST = ['top left corner', 'top right corner', 'bottom
 left corner', 'bottom right corner']
ALL_EDGES_LIST = ['top edge', 'bottom edge', 'left edge', 'right
edge']
ALL_POSITION_RELATION_LIST = ['left of', 'right of', 'behind', 'in
 front of']
```
Below shows the docstrings for these imported library functions that you must follow. You CANNOT add additional parameters to these functions.
* robot_utils Specifications:
put_first_on_second(arg1, arg2)
"""
You must not write things like:
  put_first_on_second("red block", "left of yellow block")
  put_first_on_second("red block", "top left corner")
  put_first_on_second("red block", "top edge")
You can write something like:
  put_first_on_second("red block", "yellow block")
  put_first_on_second("red block", "table")

Pick up an object (arg1) and put it at arg2. If arg2 is an object,
 arg1 will be on top of arg2. If arg2 is 'table', arg1 will be
somewhere random on the table. If arg2 is a list, arg1 will be
placed at location [x, y].

Parameters:
 arg1 (str): A string that defines the object by its color and type
 (which is either "block" or "cylinder"). For example, "red block
", "orange cylinder".
 arg2 (list or str): If it's a list, it needs to be a list of
floats, and it represents the [x, y] position to place arg1. If it
's a string, it can either be "table" or a string that defines the
 object by its color and type. arg2 must not be a free-from string
 that represents a description of a position. For example, it
cannot be relative position (e.g. "left of yellow block"), or
corner name (e.g "top left corner"), or edge name (e.g. "top edge
").
"""
stack_without_height_limit(objects_to_stack)
"""
Stack the list of objects_to_stack into one stack without
considering height limit. The first object (which is the object at
 the bottom of the stack) will also get moved and placed somewhere
 on the table.

Parameters:
  objects_to_stack (list): a list of strings, where each defines
the object by its color and type.
"""
stack_with_height_limit(objects_to_stack, height_limit)
"""

```
 Stack the list of objects_to_stack into potentially multiple
stacks, and each stack has a maximum height based on height_limit.
 The first object (which is the object at the bottom of the stack)
 will also get moved and placed somewhere on the table.

 Parameters:
   objects_to_stack (list): a list of strings, where each string
defines the object by its color and type.
   height_limit (int): an integer representing the maximum height
for each stack.
 """
 * env_utils Specifications:
 get_obj_names()
 """
 Return:
   a list of objects in the environment
 """
 get_all_obj_names_that_match_type(type_name, objects_list)
 """
 Return:
   a list of objects in the environment that match the type_name
 """
 determine_final_stacking_order(objects_to_enforce_order,
 objects_without_order)
 """
 Return:
   a sorted list of objects to stack. The first object in the list
 would be at the bottom of the stack.
 """
 parse_position(description)
 """
 You must use parse_postion for
   - relative position (e.g. "left of yellow block")
   - corner position (e.g. "top left corner")
   - edge position (e.g. "top edge")
 Return:
   a list [x, y] that represents the position described by the
 description.
 """
 You must follow these rules when you are generating Python code.
 * You MUST ONLY use Python library functions imported above. You
 MUST follow the docstrings and specification for these functions.
 * You must follow the instructions provided by the user. You
 CANNOT add additional steps, conditionals, or loops that are not
 in the instruction.
examples:
 - |
   ```
   objects=['orange block', 'yellow cylinder', 'green cylinder']
   """
   Place the green cylinder to the left of the yellow cylinder.
   """
   ```

   <end_of_example_user_query>
   ```
   # must use parse position to get relative position
   location_pos = parse_position('left of the yellow cylinder')
   put_first_on_second('green cylinder', location_pos)
   ```
 - |
   ```
   objects=['orange block', 'purple cylinder']
   """
   Place the purple cylinder at anywhere next to the orange block.
   """
```

```
    ```
    <end_of_example_user_query>
    ```
    position_relation = np.random.choice(ALL_POSITION_RELATION_LIST)
    # must use parse position to get relative position
    location_pos = parse_position(f'{position_relation} the orange
block')
    put_first_on_second('purple cylinder', location_pos)
    ```
- |
    ```
    objects=['blue block', 'brown block', 'orange cylinder', 'pink
cylinder', 'purple cylinder']
    """
    Place the blue block at any random corner.
    """
    ```
    <end_of_example_user_query>
    ```
    corner_name = np.random.choice(ALL_CORNERS_LIST)
    # must use parse position to get corner position
    corner_pos = parse_position(corner_name)
    put_first_on_second('blue block', corner_pos)
    ```
- |
    ```
    objects=['red block', 'blue block', 'cyan block', 'yellow block'
, 'orange cylinder']
    """
    1. place the orange cylinder on the table
    2. place the yellow block on the table
    3. place the red block on the table
    4. place the cyan block on the blue block
    """
    ```
    <end_of_example_user_query>
    ```
    items_to_place_first_in_order = ['orange cylinder', 'yellow
block', 'red block']
    for item in items_to_place_first_in_order:
      put_first_on_second(item, "table")
    put_first_on_second('cyan block', 'blue block')
    ```
- |
    ```
    objects=['orange block', 'blue block', 'brown block', 'yellow
block', 'gray block', 'pink block', 'cyan cylinder', 'blue
cylinder']
    """
    Stack all blocks. However, the maximum height of a stack is 3.
    """
    ```
    <end_of_example_user_query>
    ```
    block_names = get_all_obj_names_that_match_type(type_name="block
", objects_list=get_obj_names())
    stack_with_height_limit(objects_to_stack=block_names,
height_limit=3)
    ```
- |
    ```
    objects=['yellow block', 'cyan block', 'red block', 'blue block'
, 'green cylinder', 'red cylinder', 'blue cylinder']
    """
```

```
    Stack all blocks into one stack. Also make sure that the
stacking enforces the bottom to top order between these objects:
['red block', 'yellow block', 'cyan block'].
    """
    ```
    <end_of_example_user_query>
    ```
    # Based on the instruction, first define the blocks that have
strict ordering
    block_names_with_strick_order_from_bottom_to_top = ['red block',
'yellow block', 'cyan block']
    # Then, find the rest of the blocks that don't have strict order
    block_names = get_all_obj_names_that_match_type(type_name="block
", objects_list=get_obj_names())
    block_names_without_order = []
    for block in block_names:
      if block not in
block_names_with_strick_order_from_bottom_to_top:
            block_names_without_order.append(block)
    # Call helper function to determine the final stacking order
before stacking the blocks
    stack_order_from_bottom_to_top = determine_final_stacking_order(
objects_to_enforce_order=
block_names_with_strick_order_from_bottom_to_top,
objects_without_order=block_names_without_order)
    stack_without_height_limit(objects_to_stack=
stack_order_from_bottom_to_top)
    ```
  - |
    ```
    objects=['blue block', 'yellow block', 'brown block', 'orange
block', 'pink cylinder', 'brown cylinder', 'orange cylinder', '
blue cylinder']
    """
    Stack all objects into two stacks (where each stack has maximum
height of 4). It doesn't matter whether the objects are
categorized.
    """
    ```
    <end_of_example_user_query>
    ```
    object_names = get_obj_names()
    # split the objects into 2 stacks.
    stack_1 = object_names[:4]
    stack_2 = object_names[4:]
    stack_without_height_limit(objects_to_stack=stack_1)
    stack_without_height_limit(objects_to_stack=stack_2)
    ```
```

## 18.2 Robotouille Task Prompts

### 18.2.1 Recursive Summarization

```
recursive_summarization:
  main: |
    You are a helpful summarizer that recursively summarizes a
    trajectory into a more and more compacted form.
    <end_of_system_message>
    You are given a trajectory. Your goal is to summarize the
    trajectory into a more compacted form and then determine whether
    the state trajectory is sufficiently summarized.

    You must respond using the following form (you must generate [[end
     of response]] at the very end):
```

```
 [[Reasoning:]]
 You should first identify what type of trajectory this is. Then,
 you should determine what type you will summarize the trajectory
 into. Finally, you should determine whether the new type of
 trajectory is sufficiently summarized or not.
 [[Is the new trajectory sufficiently summarized? (yes/no):]]
 You should only respond using either "yes" or "no", and nothing
 else.
 [[Summarized trajectory:]]
 You should actually summarize the input trajectory and output it
 here.
 [[end of response]]

 In summary, you must follow these rules when you are summarizing a
  trajectory.
 Rules:
 * You must slowly summarize the trajectory from one type to
 another following this order: a state trajectory    > a low-level
 action trajectory --> a high-level subtask trajectory.
 * You cannot skip a type (e.g. you cannot directly summarize a low
 -level action trajectory into a high-level subtask trajectory).
 * A low-level action trajectory is represented as an unordered
 list. Each line in the unordered list should only contain one low-
 level action.
 * The low-level actions must be one of the following: "move from
 location1 to location2", "pick up item1", "place down item1 at
 location1 ", "stack item1 on top of item2", "unstack item1 from
 item 2", "cook item1", "cut item1". You should never define new
 low-level actions.
 * A high-level subtask trajectory is represented as an unordered
 list. Each line in the unordered list should only contain one high
 -level subtask. This high-level subtask should refer to one
 continuous section of the states. For example, you cannot say "at
 states 1-5, and states 10-15, the robot did    ". There can only
 be one interval of states.
 * The high-level subtask must be one of the following: "cook [ITEM
 ]", "cut [ITEM]", "stack [ITEM] on top of [ITEM]", and "unstack [
 ITEM] from [ITEM]". [ITEM] must be one of the following: "patty",
 "lettuce", "top bun", "bottom bun", "cheese", "tomato".
 * For the high-level subtask, you cannot define any other subtasks
  that are not: cook, cut, stack, or unstack. You must use choose
  from the list above.
examples:
 - |
   [[Input Trajectory:]]
   [Scenario 2]
   Cook a patty and cut a lettuce.

   State 2:
   'robot1' is at 'table6'
   'robot1' is not at 'table7'

   State 3:
   'lettuce6' is not at 'table6'
   'robot1' is holding 'lettuce6'

   State 4:
   'robot1' is at 'cutting_board6'
   'robot1' is not at 'table6'

   State 5:
   'lettuce6' is at 'cutting_board6'
   'robot1' is not holding 'lettuce6'

   State 6:
```

```
State 7:

State 8:
'lettuce6' is cut

State 9:
'robot1' is not at 'cutting_board6'
'robot1' is at 'table7'

State 10:
'patty6' is not at 'table7'
'robot1' is holding 'patty6'

State 11:
'robot1' is at 'stove6'
'robot1' is not at 'table7'

State 12:
'patty6' is at 'stove6'
'robot1' is not holding 'patty6'

State 13:

State 14:

State 15:

State 16:
'patty6' is cooked
<end_of_example_user_query>
[[Reasoning:]]
The input trajectory contains state predicates because the
trajectory talks about the status of the robot ('robot1' is at '
table6') and the status of the objects ('lettuce6' is not at '
table6').
I will summarize the state trajectory into low-level actions.
Low-level actions are not compacted enough yet because low-level
 actions can still be combined into high-level subtasks.
The new trajectory will NOT be sufficiently summarized.
[[Is the new trajectory sufficiently summarized? (yes/no):]]
no
[[Summarized trajectory:]]
  * In [Scenario 2], at state 2, the robot moved from 'table7' to
'table 6'.
  * At state 3, the robot picked up 'lettuce6'.
  * At state 4, the robot moved from 'table6' to 'cutting_board6'.
  * At state 5, the robot placed 'lettuce6' at location '
cutting_board6'.
  * At state 6-8, the robot had cut 'lettuce6'.
  * At state 9, the robot moved from 'cutting_board6' to 'table7'.
  * At state 10, the robot picked up 'patty6'.
  * At state 11, the robot moved from 'table7' to 'stove6'.
  * At state 12, the robot placed 'patty6' at 'stove6'.
  * At state 13-16, the robot had cooked 'patty6'
  [[end of response]]
- |
  [[Input Trajectory:]]
  * In [Scenario 2], at state 2, the robot moved from 'table5' to
'table9'.
```

```
     * At state 3, the robot picked up 'patty9'.
     * At state 4, the robot moved from 'table9' to 'stove5'.
     * At state 5, the robot placed 'patty9' on 'stove5'.
     * At state 6-9, the robot had cooked 'patty9'.
     * At state 10, the robot moved from 'stove5' to 'table5'.
     * At state 11, the robot picked up 'patty3'.
     * At state 12, the robot moved from 'table5' to 'stove8'.
     * At state 13, the robot placed 'patty3' on 'stove8'
     * At state 14-17, the robot had cooked 'patty3'.
     <end_of_example_user_query>
     [[Reasoning:]]
     The input trajectory contains low-level actions because the
trajectory mentions "moved", "picked up", etc.
     I will summarize the low-level action trajectory into high-level
 subtasks.
     High-level subtask trajectory is the most compacted form that
cannot be summarized anymore.
     The new trajectory will be sufficiently summarized.
     [[Is the new trajectory sufficiently summarized? (yes/no):]]
     yes
     [[Summarized trajectory:]]
     * In [Scenario 2], at state 2-9, the subtask is "cook patty".
This subtask contains: 1. moving to pick up 'patty9' (state 2-3)
2. moving to place 'patty9' on 'stove5' (state 4-5) 3. cooking '
patty9' until it is cooked (state 6-9)
     * At state 10-17, the subtask is "cook patty". This subtask
contains: 1. moving to pick up 'patty3' (state 10-11) 2. moving to
 place 'patty3' on 'stove8' (state 12-13) 3. cooking 'patty3'
until it is cooked (state 14-17)
     [[end of response]]
 - |
     [[Input Trajectory:]]
     [Scenario 1]
     Cut a lettuce before stacking it on top of a bottom bun. Then
stack a top bun on top of the lettuce.

     State 2:
     'lettuce1' is not at 'table6'
     'robot1' is holding 'lettuce1'

     State 3:
     'robot1' is not at 'table6'
     'robot1' is at 'cutting_board1'

     State 4:
     'robot1' is not holding 'lettuce1'
     'lettuce1' is at 'cutting_board1'

     State 5:

     State 6:

     State 7:
     'lettuce1' is cut

     State 8:
     'lettuce1' is not at 'cutting_board1'
     'robot1' is holding 'lettuce1'

     State 9:
     'robot1' is not at 'cutting_board1'
     'robot1' is at 'table2'
```

```
State 10:
'lettuce1' is at 'table2'
'lettuce1' is on top of 'bottom_bun1'
'robot1' is not holding 'lettuce1'

State 11:
'robot1' is not at 'table2'
'robot1' is at 'table5'

State 12:
'top_bun2' is not at 'table5'
'robot1' is holding 'top_bun2'

State 13:
'robot1' is not at 'table5'
'robot1' is at 'table2'

State 14:
'top_bun2' is at 'table2'
'top_bun2' is on top of 'lettuce1'
'robot1' is not holding 'top_bun2'
<end_of_example_user_query>
[[Reasoning:]]
The input trajectory contains state predicates because the
trajectory talks about the status of the robot ('robot1' is
holding 'lettuce1') and the status of the objects ('lettuce1' is
not at 'table6').
I will summarize the state trajectory into low-level actions.
Low-level actions are not compacted enough yet because low-level
actions can still be combined into high-level subtasks.
The new trajectory will NOT be sufficiently summarized.
[[Is the new trajectory sufficiently summarized? (yes/no):]]
no
[[Summarized trajectory:]]
* In [Scenario 1], at state 2, the robot picked up 'lettuce1'.
* At state 3, the robot moved from 'table6' to 'cutting_board1'.
* At state 4, the robot placed 'lettuce1' at location '
cutting_board1'.
* At state 5-7, the robot had cut 'lettuce1'.
* At state 8, the robot picked up 'lettuce1'.
* At state 9, the robot moved from 'cutting_board1' to 'table2'.
* At state 10, the robot placed 'lettuce1' on top of '
bottom_bun1' at location 'table2'.
* At state 11, the robot moved from 'table2' to 'table5'.
* At state 12, the robot picked up 'top_bun2'.
* At state 13, the robot moved from 'table5' to 'table2'.
* At state 14, the robot placed 'top_bun2' on top of 'lettuce1'
at location 'table2'.
[[end of response]]
- |
[[Input Trajectory:]]
* In [Scenario 1], at state 2, the robot picked up 'lettuce1'.
* At state 3, the robot moved from 'table6' to 'cutting_board1'.
* At state 4, the robot placed 'lettuce1' at location '
cutting_board1'.
* At state 5-7, the robot had cut 'lettuce1'.
* At state 8, the robot picked up 'lettuce1'.
* At state 9, the robot moved from 'cutting_board1' to 'table2'.
* At state 10, the robot placed 'lettuce1' on top of '
bottom_bun1' at location 'table2'.
* At state 11, the robot moved from 'table2' to 'table5'.
* At state 12, the robot picked up 'top_bun2'.
* At state 13, the robot moved from 'table5' to 'table2'.
* At state 14, the robot placed 'top_bun2' on top of 'lettuce1'
at location 'table2'.
```

```
      <end_of_example_user_query >
      [[Reasoning:]]
     The input trajectory contains low-level actions because the
   trajectory mentions "picked up", "moved", etc.
     I will summarize the low-level action trajectory into high-level
    subtasks.
     High-level subtask trajectory is the most compacted form that
   cannot be summarized anymore.
     The new trajectory will be sufficiently summarized.
     [[Is the new trajectory sufficiently summarized? (yes/no):]]
     yes
     [[Summarized trajectory:]]
     * In [Scenario 1], at state 2-7, the subtask is "cut lettuce".
   This subtask contains: 1. pick up 'lettuce1' (state 2) 2. moving
   to place 'lettuce1' on 'cutting_board1' (state 3-4) 3. cutting '
   lettuce1' until it is cut (state 5-7)
     * At state 8-10, the subtask is "stack lettuce on top of bottom
   bun". This subtask contains: 1. picking up 'lettuce1' (state 8) 2.
    moving to stack 'lettuce1' on 'bottom_bun1' (state 9-10)
     * At state 11-14, the subtask is "stack top bun on top of
   lettuce". This subtask contains: 1. moving to pick up 'top_bun2' (
   state 11-12) 2. moving to stack 'top_bun2' on 'lettuce1' (state
   13-14)
      [[end of response]]
```

## Summarized Demonstrations -> Task Specification

```
summary_2_spec:
  main: |
    You are a helpful assistant that analyzes the high-level subtask
    trajectories and summarizes them into a concise pseudocode-style
    task specification.
    <end_of_system_message>
    You are given (1) a high-level goal and (2) one or more high-level
     subtask trajectories where each one represents a different
    scenario. Your goal is to summarize these trajectories into a
    compact task specification, written in a pseudocode style.

    You must respond using the following format (you must generate [[
    end of response]] at the very end):
    [[Reasoning:]]
    You should first list out the order of the high-level subtask
    trajectories in all scenarios. Then, you should consider if
    certain sections of the subtasks are repeated in the scenario and
    can be represented by a loop. Two scenarios having exactly the
    same list does not mean that there is a loop. A loop only exists
    among subtasks in one individual scenario.
    Overall, your task specification should work for all scenarios.
    You should make sure that the task specification matches the
    subtasks ordering across all scenarios. You should also make sure
    that the task specification uses loops when there is any
    repetition.
    [[Task Specification:]]
    You should first state the high-level goal. Then, you should say "
    Specifically:" before outputting the pseudocode-style task
    specification.
    [[end of response]]

    You must follow these rules when you are writing the task
    specifications.
    Rules:
```

* You must write the task specifications in pseudocode style. You
 should not write the task specification as a list. You cannot
 number any line.
 * When checking for loops, you cannot compare the subtasks across
 multiple scenarios. Even if two scenarios have the exact same list
  of subtasks, there is NOT any loop. Loops can only exist within
 the list of subtasks for one individual scenario. Do not consider
 loops across multiple scenarios.
examples:
 - |
   [[High-Level Goal:]]
   Cook a patty and cut a lettuce.
   [[Trajectories:]]
   * In [Scenario 2], at state 2-8, the subtask is "cut lettuce".
 This subtask contains: 1. moving to pick up 'lettuce4' (state 2-3)
  2. moving to place 'lettuce4' on 'cutting_board3' (state 4-5) 3.
 cutting 'lettuce4' until it is cut (state 6-8)
   * At state 9-16, the subtask is "cook patty". This subtask
 contains: 1. moving to picking up 'patty2' (state 9-10) 2. moving
 to place 'patty2' at location 'stove8' (state 11-12) 3. cooking '
 patty2' until it is cooked (state 13-16)
   * In [Scenario 2], at state 2-8, the subtask is "cut lettuce".
 This subtask contains: 1. moving to pick up 'lettuce6' (state 2-3)
  2. moving to place 'lettuce6' on 'cutting_board6' (state 4-5) 3.
 cutting 'lettuce6' until it is cut (state 6-8)
   * At state 9-16, the subtask is "cook patty". This subtask
 contains: 1. moving to picking up 'patty6' (state 9-10) 2. moving
 to place 'patty6' at location 'stove6' (state 11-12) 3. cooking '
 patty6' until it is cooked (state 13-16)
   <end_of_example_user_query>
   [[Reasoning:]]
   * In [Scenario 1]'s unique kitchen environment, the subtasks
 were executed in this order: ['cut lettuce', 'cook patty'].
   * In [Scenario 2]'s unique kitchen environment, the subtasks
 were executed in this order: ['cut lettuce', 'cook patty'].
   * Both scenarios are executing the subtasks in the same order.
   * There is no repetition or loop in the subtask ordering for any
  individual scenario.
   [[Task Specification:]]
   Cook a patty and cut a lettuce.

   Specifically:
   Get a list of all the lettuces in the kitchen.
   Get a list of all the patties in the kitchen.

   Decide a lettuce to use.
   Get a list of all the available cutting boards in the kitchen.
   Decide a cutting board to go to.
   Cut that lettuce at that cutting board.

   Decide a patty to use.
   Get a list of all the available stoves in the kitchen.
   Decide a stove to go to.
   Cook that patty at that stove.
 - |
   [[High-Level Goal:]]
   Cook two patties.
   [[Trajectories:]]
   * In [Scenario 1], at state 2-9, the subtask is "cook patty".
 This subtask contains: 1. moving to pick up 'patty4' (state 2-3)
 2. moving to place 'patty4' on 'stove10' (state 4-5) 3. cooking '
 patty4' until it is cooked (state 6-9)
   * At state 10-17, the subtask is "cook patty". This subtask
 contains: 1. moving to pick up 'patty6' (state 10-11) 2. moving to

place 'patty6' on 'stove11' (state 12-13) 3. cooking 'patty6'
until it is cooked (state 14-17)
   * In [Scenario 2], at state 2-9, the subtask is "cook patty".
This subtask contains: 1. moving to pick up 'patty9' (state 2-3)
2. moving to place 'patty9' on 'stove5' (state 4-5) 3. cooking '
patty9' until it is cooked (state 6-9)
   * At state 10-17, the subtask is "cook patty". This subtask
contains: 1. moving to pick up 'patty3' (state 10-11) 2. moving to
 place 'patty3' on 'stove8' (state 12-13) 3. cooking 'patty3'
until it is cooked (state 14-17)
   <end_of_example_user_query>
   [[Reasoning:]]
   * In [Scenario 1]'s unique kitchen environment, the subtasks
were executed in this order: ['cook patty', 'cook patty'].
   * In [Scenario 2]'s unique kitchen environment, the subtasks
were executed in this order: ['cook patty', 'cook patty'].
   * Both scenarios are executing the subtasks in the same order.
   * In both scenarios, we see that the subset ['cook patty'] got
repeated 2 times consecutively, so we can use a for-loop in our
specification.
   [[Task Specification:]]
   Cook two patties.

   Specifically:
   Get a list of all the patties in the kitchen.

   For two patties, do:
     Decide a patty to use.
     Get a list of all the available stoves in the kitchen.
     Decide a stove to go to.
     Cook that patty at that stove.
 - |
   [[High-Level Goal:]]
   Cut a lettuce before stacking it on top of a bottom bun. Then
stack a top bun on top of the lettuce.
   [[Trajectories:]]
   * In [Scenario 1], at state 2-7, the subtask is "cut lettuce".
This subtask contains: 1. pick up 'lettuce1' (state 2) 2. moving
to place 'lettuce1' on 'cutting_board1' (state 3-4) 3. cutting '
lettuce1' until it is cut (state 5-7)
   * At state 8-10, the subtask is "stack lettuce on top of bottom
bun". This subtask contains: 1. picking up 'lettuce1' (state 8) 2.
 moving to stack 'lettuce1' on 'bottom_bun1' (state 9-10)
   * At state 11-14, the subtask is "stack top bun on top of
lettuce". This subtask contains: 1. moving to pick up 'top_bun2' (
state 11-12) 2. moving to stack 'top_bun2' on 'lettuce1' (state
13-14)
   * In [Scenario 2], at state 2-7, the subtask is "cut lettuce".
This subtask contains: 1. pick up 'lettuce4' (state 2) 2. moving
to place 'lettuce4' on 'cutting_board2' (state 3-4) 3. cutting '
lettuce4' until it is cut (state 5-7)
   * At state 8-10, the subtask is "stack lettuce on top of bottom
bun". This subtask contains: 1. picking up 'lettuce4' (state 8) 2.
 moving to stack 'lettuce4' on 'bottom_bun3' (state 9-10)
   * At state 11-14, the subtask is "place top bun on top of
lettuce". This subtask contains: 1. moving to pick up 'top_bun1' (
state 11-12) 2. moving to place 'top_bun1' on top of 'lettuce4' (
state 13-14)
   <end_of_example_user_query>
   [[Reasoning:]]
   * In [Scenario 1]'s unique kitchen environment, the subtasks
were executed in this order: ['cut lettuce', 'stack lettuce on top
 of bottom bun', 'stack top bun on top of lettuce'].

```
   * In [Scenario 2]'s unique kitchen environment, the subtasks
were executed in this order: ['cut lettuce', 'stack lettuce on top
 of bottom bun', 'place top bun on top of lettuce'].
   * 'stack lettuce on top of bottom bun' and 'place top bun on top
 of lettuce' are essentially the same subtask.
   * Both scenarios are executing the subtasks in the same order.
   * There is no repetition or loop in the subtask ordering for any
 individual scenario.
  [[Task Specification:]]
  Cut a lettuce before stacking it on top of a bottom bun. Then
stack a top bun on top of the lettuce.

  Specifically:
  Get a list of all the lettuces in the kitchen.
  Get a list of all the bottom buns in the kitchen.
  Get a list of all the top buns in the kitchen.

  Decide a lettuce to use.
  Get a list of all the available cutting boards in the kitchen.
  Decide a cutting board to go to.
  Cut that lettuce at that cutting board.

  Decide a bottom bun to use.
  Stack the lettuce on top of the bottom bun.

  Decide a top bun to use.
  Stack the top bun on top of the lettuce.
```

### 18.2.2 Recursive Expansion

```
spec_2_highlevelcode:
  main: |
    You are a Python code generator for robotics. The users will first
     provide the imported Python modules. Then, for each code they
    want you to generate, they provide the requirements and pseudocode
     in Python docstrings.
    <end_of_system_message>
    You need to write kitchen robot control scripts in Python code.
    The Python code should be general and applicable to different
    kitchen environments.

    Below are the imported Python libraries and functions that you can
     use. You CANNOT import new libraries.
    ```
    # Python kitchen robot control script
    from env_utils import get_all_obj_names_that_match_type,
    get_all_location_names_that_match_type, is_cut, is_cooked
    ```
    Below shows the docstrings for these imported library functions
    that you must follow. You CANNOT add additional parameters to
    these functions.
    * env_utils Specifications:
    get_all_obj_names_that_match_type(obj_type)
    """
    Parameters:
    obj_type (str): A string that has to be one of 'patty', 'lettuce',
     'bottom bun' ,'top bun'.
    Return:
         (list) a list of string of objects that match the obj_type
    """
    get_all_location_names_that_match_type(location_type)
    """
    Parameters:
```

```
      location_type (str): A string that has to be one of 'table',
 'cutting_board', 'stove'.
 Return:
      (list) a list of string of locations that match the
location_type
"""
is_cut(obj)
"""
Parameters:
      obj (str): A string that has the type of a cuttable object (
must be 'lettuce') and an id. For example: 'lettuce3', 'lettuce1'
 Return:
      (boolean) true if obj is cut
"""
is_cooked(obj)
"""
Parameters:
      obj (str): A string that has the type of a cookable object (
must be 'patty') and an id. For example: 'patty1', 'patty2'
 Return:
      (boolean) true if obj is cooked
"""
```

 You must follow these rules when you are generating Python code.
 * You can use Python library functions imported above. You MUST
follow the docstrings and specification for these functions.
 * You can also call new functions that are not yet defined.
However, these functions must be completing general, high-level
subtasks (cook, cut, place on top, unstack). These functions
cannot be doing low-level actions.
 * You must follow the pseudocode provided by the user. You CANNOT
add additional steps, conditionals, or loops that are not in the
pseudocode.
examples:
 - |
    ```
    """
    Cook a patty and cut a lettuce.

    Specifically:
    Get a list of all the lettuces in the kitchen.
    Get a list of all the patties in the kitchen.

    Decide a lettuce to use.
    Get a list of all the available cutting boards in the kitchen.
    Decide a cutting board to go to.
    Cut that lettuce at that cutting board.

    Decide a patty to use.
    Get a list of all the available stoves in the kitchen.
    Decide a stove to go to.
    Cook that patty at that stove.
    """
    ```

    <end_of_example_user_query>
    ```
    # Get a list of all the lettuces in the kitchen.
    lettuces = get_all_obj_names_that_match_type('lettuce')
    # Get a list of all the patties in the kitchen.
    patties = get_all_obj_names_that_match_type('patty')

    # Decide a lettuce to use.
    lettuce_to_use = lettuces[0]
    # Get a list of all the available cutting boards in the kitchen.
```

```
    cutting_boards = get_all_location_names_that_match_type('
cutting_board')
    # Decide a cutting board to go to.
    cutting_board_to_cut_at = cutting_boards[0]
    # Cut that lettuce at that cutting board.
    cut_object_at_location(obj=lettuce_to_use, location=
cutting_board_to_cut_at)

    # Decide a patty to use.
    patty_to_use = patties[0]
    # Get a list of all the available stoves in the kitchen.
    stoves = get_all_location_names_that_match_type('stove')
    # Decide a stove to go to.
    stove_to_cook_at = stoves[0]
    # Cook that patty at that stove.
    cook_object_at_location(obj=patty_to_use, location=
stove_to_cook_at)
    ```
- |
    ```
    """
    Cook two patties.

    Specifically:
    Get a list of all the patties in the kitchen.

    Decide a patty to use.
    Get a list of all the available stoves in the kitchen.
    Decide a stove to go to.
    Cook that patty at that stove.

    Repeat the above steps for the second patty.
    """
    ```
    <end_of_example_user_query>
    ```
    # Get a list of all the patties in the kitchen.
    patties = get_all_obj_names_that_match_type('patty')

    # For two patties, do
    for i in range(2):
      # Decide a patty to use.
      patty_to_use = patties[i]
      # Get a list of all the available stoves in the kitchen.
      stoves = get_all_location_names_that_match_type('stove')
      # Decide a stove to go to.
      stove_to_cook_at = stoves[i]
      # Cook that patty at that stove.
      cook_object_at_location(obj=patty_to_use, location=
stove_to_cook_at)
    ```
- |
    ```
    """
    Cut a lettuce before stacking it on top of a bottom bun. Then
stack a top bun on top of the lettuce.

    Specifically:
    Get a list of all the lettuces in the kitchen.
    Get a list of all the bottom buns in the kitchen.
    Get a list of all the top buns in the kitchen.

    Decide a lettuce to use.
    Get a list of all the available cutting boards in the kitchen.
    Decide a cutting board to go to.
```

```
        Cut that lettuce at that cutting board.

        Decide a bottom bun to use.
        Stack the lettuce on top of the bottom bun.

        Decide a top bun to use.
        Stack the top bun on top of the lettuce.
        """
        ```
        <end_of_example_user_query>
        ```
        # Get a list of all the lettuces in the kitchen.
        lettuces = get_all_obj_names_that_match_type('lettuce')
        # Get a list of all the bottom buns in the kitchen.
        bottom_buns = get_all_obj_names_that_match_type('bottom_bun')
        # Get a list of all the top buns in the kitchen.
        top_buns = get_all_obj_names_that_match_type('top_bun')

        # Decide a lettuce to use.
        lettuce_to_use = lettuces[0]
        # Get a list of all the available cutting boards in the kitchen.
        cutting_boards = get_all_location_names_that_match_type('
    cutting_board')
        # Decide a cutting board to go to.
        cutting_board_to_cut_at = cutting_boards[0]
        # Cut that lettuce at that cutting board.
        cut_object_at_location(obj=lettuce_to_use, location=
    cutting_board_to_cut_at)

        # Decide a bottom bun to use.
        bottom_bun_to_use = bottom_buns[0]
        # Stack the lettuce on top of the bottom bun.
        # obj1 should be the lettuce, obj2 should be the bottom bun.
        stack_obj1_on_obj2(obj1=lettuce_to_use, obj2=bottom_bun_to_use)

        # Decide a top bun to use.
        top_bun_to_use = top_buns[0]
        # Stack that top bun on top of the lettuce.
        # obj1 should be the top bun, obj2 should be the lettuce.
        stack_obj1_on_obj2(obj1=top_bun_to_use, obj2=lettuce_to_use)
```

**Step 2: Define composite actions**  Given a function header, the LLM outputs code that may contain undefined functions.

```
step2:
  main: |
    You are a Python code generator for robotics. The users will first
     provide the imported Python modules. Then, for each code that
    they want you to generate, they provide the requirement in Python
    docstrings.
    <end_of_system_message>
    # Python kitchen robot control script
    from env_utils import get_obj_location, is_holding, is_in_a_stack,
     get_obj_that_is_underneath
    """
    All the code should follow the specification.

    env_utils Specifications:
    get_obj_location(obj)
        Parameters:
            obj (str): A string that has the type of object (one of '
    lettuce', 'patty', 'bottom_bun' ,'top_bun') and an id. For example
    : 'lettuce5', 'patty7', 'bottom_bun1', 'top_bun4'
```

```
        Return:
            (str) location that the object is currently at. A string
    that has the type of location (one of 'table', 'cutting_board', '
    stove') and an id. For example: 'table2', 'cutting_board1', '
    stove5'
     is_holding(obj)
        Parameters:
            obj (str): A string that has the type of object (one of '
    lettuce', 'patty', 'bottom_bun' ,'top_bun') and an id. For example
    : 'lettuce5', 'patty7', 'bottom_bun1', 'top_bun4'
        Return:
            (bool) true if the robot is currently holding obj
     is_in_a_stack(obj)
        Parameters:
            obj (str): A string that has the type of object (one of '
    lettuce', 'patty', 'bottom_bun' ,'top_bun') and an id. For example
    : 'lettuce5', 'patty7', 'bottom_bun1', 'top_bun4'
        Return:
            (bool) true if the obj is in a stack and the robot needs
    to unstack the obj
     get_obj_that_is_underneath(obj_at_top)
        Parameters:
            obj_at_top (str): A string that has the type of object (
    one of 'lettuce', 'patty', 'bottom_bun' ,'top_bun') and an id. For
     example: 'lettuce5', 'patty7', 'bottom_bun1', 'top_bun4'
        Return:
            (str) The name of the object that is directly underneath
    the obj_at_top. A string that has the type of object (one of '
    lettuce', 'patty', 'bottom_bun' ,'top_bun') and an id. For example
    : 'lettuce5', 'patty7', 'bottom_bun1', 'top_bun4'
     """
     """
     The robot is only capable of these basic actions: move, pick,
    place, stack, unstack, cook, cut.
     These functions require the robot to hold the object: place, stack
     These functions require the robot to not hold anything: pick,
    unstack

     You can define functions that combine two of these actions, but
     you cannot define new actions.
     """
examples:
  - |
    ```
    """
    You can use functions imported above and also call new functions

    The robot might not be near any object or near any location
    specified in the function parameters.

    Define the function: cut_object_at_location(obj, location)
    """
    ```

    <end_of_example_user_query>
    ```

    def cut_object_at_location(obj, location):
      # To cut an object, the robot first needs to be holding obj
      if not is_holding(obj):
          # If the robot is not holding obj, there are 2 scenarios:
          #   (1) if obj is in a stack ,unstack obj
          #   (2) else, pick up obj.
          if is_in_a_stack(obj):
              # Because obj is in a stack, robot need to move then
    unstack the obj from the obj_at_bottom first
```

```
                obj_at_bottom = get_obj_that_is_underneath(obj_at_top=
obj)
                # move then unstack: first you move to the
obj_at_bottom's location, then you unstack obj from obj_at_bottom
                move_then_unstack(obj_to_unstack=obj, obj_at_bottom=
obj_at_bottom, unstack_location=get_obj_location(obj_at_bottom))
            else:
                # move_then_pick: first you move to obj's location,
then you pick obj up
                move_then_pick(obj=obj, pick_location=get_obj_location
(obj))
        # move then place: first you move to the location to cut at,
then you place obj at that location
        move_then_place(obj=obj, place_location=location)
        # cut the object until it is cut
        cut_until_is_cut(obj=obj)
    ```
  - |
        ```
        """
        You can use functions imported above and also call new
functions

        The robot might not be near any object or near any location
specified in the function parameters.

        Define the function: stack_obj1_on_obj2(obj1, obj2)
        """
        ```
        <end_of_example_user_query>
        ```
        def stack_obj1_on_obj2(obj1, obj2):
            # To stack obj1 on obj2, the robot needs to be holding
obj1
            if not is_holding(obj1):
                # If the robot is not holding obj1, there are 2
scenarios:
                #   (1) if obj1 is in a stack ,unstack obj1
                #   (2) else, pick up obj1.
                if is_in_a_stack(obj1):
                    # Because obj1 is in a stack, robot need to move
then unstack the obj from the obj_at_bottom first
                    obj_at_bottom = get_obj_that_is_underneath(
obj_at_top=obj1)
                    # move then unstack: first you move to the
obj_at_bottom's location, then you unstack obj from obj_at_bottom
                    move_then_unstack(obj_to_unstack=obj1,
obj_at_bottom=obj_at_bottom, unstack_location=get_obj_location(
obj_at_bottom))
                else:
                    # move_then_pick: first you move to obj's location
, then you pick obj up
                    move_then_pick(obj=obj, pick_location=
get_obj_location(obj))
            # determine the location of obj2 to stack on
            obj2_location = get_obj_location(obj2)
            # move then stack: first you move to obj2's location, then
 you stack obj1 on obj2
            move_then_stack(obj_to_stack=obj1, obj_at_bottom=obj2,
stack_location=obj2_location)
        ```
  - |
        ```
        """
    You can use functions imported above and also call new functions
```

```
   The robot might not be near any object or near any location
specified in the function parameters.

   Define the function: unstack_obj1_from_obj2(obj1, obj2)
   """
   ```

   <end_of_example_user_query>
   ```

   def unstack_obj1_from_obj2(obj1, obj2):
       # To unstack obj1 from obj2, the robot needs to not hold
anything yet.
       if is_holding(obj1):
           # Because the robot is holding obj1, unstacking must
have been successful already
           return
       # determine the location of obj2 to unstack from
       obj2_location = get_obj_location(obj2)
       # move then unstack: first you move to obj2's location, then
 you unstack obj1 from obj2
       move_then_unstack(obj_to_unstack=obj1, obj_at_bottom=obj2,
unstack_location=obj2_location)
   ```
 - |
   ```
   """
   You can use functions imported above and also call new functions

   The robot might not be near any object or near any location
specified in the function parameters.

   Define the function: cook_object_at_location(obj, cook_location)
   """
   ```

   <end_of_example_user_query>
   ```

   def cook_object_at_location(obj, cook_location):
     # To cook an object, the robot first needs to be holding obj
     if not is_holding(obj):
         # If the robot is not holding obj, there are 2 scenarios:
         #   (1) if obj is in a stack ,unstack obj
         #   (2) else, pick up obj.
         if is_in_a_stack(obj):
             # Because obj is in a stack, robot need to move then
unstack the obj from the obj_at_bottom first
             obj_at_bottom = get_obj_that_is_underneath(obj_at_top=
obj)
             # move then unstack: first you move to the
obj_at_bottom's location, then you unstack obj from obj_at_bottom
             move_then_unstack(obj_to_unstack=obj, obj_at_bottom=
obj_at_bottom, unstack_location=get_obj_location(obj_at_bottom))
         else:
             # move_then_pick: first you move to obj's location,
then you pick obj up
             move_then_pick(obj=obj, pick_location=get_obj_location
(obj))
     # move then place: first you move to the location to cook at,
then you place obj at that location
     move_then_place(obj=obj, place_location=cook_location)
     # cook the object until it is cooked
     cook_until_is_cooked(obj=obj)
   ```

step3:
```

```
main: |
  You are a Python code generator for robotics. The users will first
   provide the imported Python modules. Then, for each code that
  they want you to generate, they provide the requirement in Python
  docstrings.
  <system_message_separator>
  # Python kitchen robot control script
  import numpy as np
  from robot_utils import move, pick_up, place, cut, start_cooking,
  noop, stack, unstack
  from env_utils import is_cut, is_cooked, get_obj_location,
  get_curr_location
  """
  All the code should follow the specification.

  robot_utils Specifications:
  move(curr_loc, target_loc)
      Requirement:
          The curr_loc cannot be the same as target_loc.

      Move from the curr_loc to the target_loc.

      Parameters:
          curr_loc (str): a string that has the type of location (
  one of 'table', 'cutting_board', 'stove') and an id. For example:
  'table2', 'cutting_board1', 'stove5'
          target_loc (str): a string that has the type of location (
  one of 'table', 'cutting_board', 'stove') and an id. For example:
  'table2', 'cutting_board1', 'stove5'
  pick_up(obj, loc)
      Requirement:
          The robot must have moved to loc already, and it cannot be
   holding anything else.

      Pick up the obj from the loc.

      Parameters:
          obj (str): object to pick. A string that has the type of
  object (one of 'lettuce', 'patty', 'bottom bun' ,'top bun') and an
   id. For example: 'lettuce5', 'patty7', 'bun1'
          loc (str): location to pick the object from. a string that
   has the type of location (one of 'table', 'cutting_board', 'stove
  ') and an id. For example: 'table2', 'cutting_board1', 'stove5'
  place(obj, loc)
      Requirement:
          The robot must have moved to loc already, and it cannot be
   holding anything else.

      Place the obj on the loc.

      Parameters:
          obj (str): object to place. A string that has the type of
  object (one of 'lettuce', 'patty', 'bottom bun' ,'top bun') and an
   id. For example: 'lettuce5', 'patty7', 'bun1'
          loc (str): location to place the object at. a string that
  has the type of location (one of 'table', 'cutting_board', 'stove
  ') and an id. For example: 'table2', 'cutting_board1', 'stove5'
  cut(obj)
      Requirement:
          The robot must be at the same location as obj.

      Make progress on cutting the obj. You need to call this
  function multiple times to finish cutting the obj.

      Parameters:
```

```
            obj (str): object to cut. A string that has the type of a
cuttable object (must be 'lettuce') and an id. For example: '
lettuce3', 'lettuce1'
start_cooking(obj)
    Requirement:
        The robot must be at the same location as obj.

    Start cooking the obj. You only need to call this once. The
obj will take an unknown amount before it is cooked.

    Parameters:
        obj (str): object to cook. A string that has the type of a
cookable object (must be 'patty') and an id. For example: 'patty1
', 'patty5'
noop()
    Do nothing
stack(obj_to_stack, obj_at_bottom)
    Requirement:
        The robot must be at the same location as obj_at_bottom.

    Stack obj_to_stack on top of obj_at_bottom

    Parameters:
        obj_to_stack (str): object to stack. A string that has the
type of object (one of 'lettuce', 'patty', 'bottom bun' ,'top bun
') and an id. For example: 'lettuce5', 'patty7', 'bun1'
        obj_at_bottom (str): object to stack on top of. A string
that has the type of object (one of 'lettuce', 'patty', 'bottom
bun' ,'top bun') and an id. For example: 'lettuce5', 'patty7', '
bun1'
unstack(obj_to_unstack, obj_at_bottom)
    Requirement:
        The robot must be at the same location as obj_at_bottom.

    Unstack obj_to_unstack from obj_at_bottom

    Parameters:
        obj_to_unstack (str): object to unstack. A string that has
the type of object (one of 'lettuce', 'patty', 'bottom bun' ,'top
bun') and an id. For example: 'lettuce5', 'patty7', 'bun1'
        obj_at_bottom (str): object to unstack from. A string that
has the type of object (one of 'lettuce', 'patty', 'bottom bun'
,'top bun') and an id. For example: 'lettuce5', 'patty7', 'bun1'

env_utils Specifications:
is_cut(obj)
    Parameters:
        obj (str): A string that has the type of a cuttable object
(must be 'lettuce') and an id. For example: 'lettuce3', 'lettuce1
'

    Return:
        (boolean)
is_cooked(obj)
    Parameters:
        obj (str): A string that has the type of a cookable object
(must be 'patty') and an id. For example: 'patty1', 'patty2'
    Return:
        (boolean)
get_curr_location()
    Return:
        (str) location that the robot is currently at. A string
that has the type of location (one of 'table', 'cutting_board', '
stove') and an id. For example: 'table2', 'cutting_board1', '
stove5'
get_obj_location(obj)
```

```
        Parameters:
            obj (str): A string that has the type of a cuttable object
   (must be 'lettuce') and an id. For example: 'lettuce3', 'lettuce1
'
        Return:
            (str) location that the object is currently at. A string
  that has the type of location (one of 'table', 'cutting_board', '
  stove') and an id. For example: 'table2', 'cutting_board1', '
  stove5'
  """
examples:
  - |
    ```
    """
    You can only use the functions imported in the header.

    Define the function: basic_move(target_loc)
        Move to any location specified the target_loc.
    """
    ```
    <end_of_example_user_query>
    ```
    def basic_move(target_loc):
        if get_curr_location() != target_loc:
            move(get_curr_location(), target_loc)
    ```
  - |
    ```
    """
    You can only use the functions imported in the header.

    Define the function: cook_until_is_cooked(obj)
    """
    ```
    <end_of_example_user_query>
    ```
    def cook_until_is_cooked(obj):
        start_cooking(obj)
        while not is_cooked(obj):
            noop()
    ```
  - |
    ```
    """
    You can only use the functions imported in the header.

    Define the function: move_then_stack(obj_to_stack, obj_at_bottom
)
    """
    ```
    <end_of_example_user_query>
    ```
    def move_then_stack(obj_to_stack, obj_at_bottom):
        # For stacking, because the location is not provided, we
  need to determine the stack_location
        stack_location = get_obj_location(obj_at_bottom)
        if get_curr_location() != stack_location:
            move(get_curr_location(), stack_location)
        stack(obj_to_stack, obj_at_bottom)
    ```
```

## 18.3 EPIC Kitchens Task Prompts

### 18.3.1 Recursive Summarization

```
recursive_summarization:
  main: |
    You are a helpful summarizer that recursively summarizes a
    trajectory into a more and more compacted form.
    <end_of_system_message>
    You are given a trajectory. Your goal is to summarize the
    trajectory into a more compacted form and then determine whether
    the state trajectory is sufficiently summarized.

    You must respond using the following form (you must generate [[end
     of response]] at the very end):
    [[Reasoning:]]
    You should first identify what type of trajectory this is. Then,
    you should determine what type you will summarize the trajectory
    into. Finally, you should determine whether the new type of
    trajectory is sufficiently summarized or not.
    [[Is the new trajectory sufficiently summarized? (yes/no):]]
    You should only respond using either "yes" or "no", and nothing
    else.
    [[Summarized trajectory:]]
    You should actually summarize the input trajectory and output it
    here.
    [[end of response]]

    You must follow these rules when you are summarizing a trajectory.
    Rules:
    * You must slowly summarize the trajectory from one type to
    another following this order: a state trajectory    > a low-level
    action trajectory --> a high-level subtask trajectory.
    * You cannot skip a type (e.g. you cannot directly summarize a low
    -level action trajectory into a high-level subtask trajectory).
    * A low-level action trajectory is represented as an unordered
    list.
    * The low-level actions must be one of the following: "the robot
    moved from location1 to location2", "the robot picked up item1", "
    the robot placed down item1 at location1 ", "the robot soaped
    item1 until it became soapy because item1 was dirty", "the robot
    rinsed item1 until it became clean", "the robot turned on tap1", "
    the robot turned off tap2". You should never define new low-level
    actions.
    * A high-level subtask trajectory is represented as an unordered
    list. This high-level subtask should refer to one continuous
    section of the states. For example, you cannot say "at states 1-5,
     and states 10-15, the robot did    ". There can only be one
    interval of states.
    * The high-level subtask must be one of the following: "brought [
    ITEMS] from [LOCATION 1] to [LOCATION 2]", "soaped [ITEM] until it
     became soap", "rinsed [ITEM] until it became clean", "picked up [
    ITEM]", "placed [ITEMS] in [LOCATION]", "turned on [TAP]", "turned
     off [TAP]". You should never define new high-level subtasks. You
    must choose from the list above.
  examples:
    - |
      [[Input Trajectory:]]
      [Scenario 1]
      Wash objects at a sink. All cleaned objects should be placed in
    the sink.
      Objects to clean
        - spoon_1 at countertop_1
```

```
- fork_1 at countertop_1

Initial Condition (State 1):
at(`countertop_1`)
is_dirty(`spoon_1`)
is_dirty(`fork_1`)
near(`spoon_1`)
near(`fork_1`)

State 2:
is_in_hand(`fork_1`)
is_in_hand(`spoon_1`)
far(`sink_1`)

State 3:
at(`sink_1`)

State 4:
is_soapy(`fork_1`)
is_on((`tap_1`)

State 5:
is_soapy(`spoon_1`)

State 6:
is_clean(`spoon_1`)

State 7
is_clean(`fork_1`)

State 8
is_off(`tap_1`)
in(`fork_1`, `sink_1`)
in(`spoon_1`, `sink_1`)
<end_of_example_query>
[[Reasoning:]]
  The input trajectory contains state predicates because the
trajectory talks about the status of robot (at(`countertop_1`))
and the objects (is_dirty(`spoon_1`)).
  I will summarize the state trajectory into low-level actions.
  Low-level actions are not compacted enough yet because low-level
 actions can still be combined into high-level subtasks.
  The new trajectory will NOT be sufficiently summarized.
  [[Is the new trajectory sufficiently summarized? (yes/no):]]
  no
  [[Summarized trajectory:]]
  * In [Scenario 1], at state 1-2, the robot picked up spoon_1.
The robot picked up fork_1.
  * At state 2-3, the robot moved from countertop_1 to sink_1.
  * At state 3-4, the robot turned on tap_1. The robot soaped
fork_1 until it became soapy because fork_1 was dirty.
  * At state 4-5, the robot soaped spoon_1 until it became soapy
because spoon_1 was dirty.
  * At state 5-6, the robot rinsed spoon_1 until it became clean.
  * At state 6-7, the robot rinsed fork_1 until it became clean.
  * At state 7-8, the robot turned off tap_1. The robot placed
spoon_1 in sink_1. The robot placed fork_1 in sink_1.
  [[end of response]]
- |
  [[Input Trajectory:]]
  [Scenario 1]
  Wash objects at a sink. All cleaned objects should be placed in
the sink.
  Objects to clean
    - spoon_1 at dishwasher_1
```

```
  - plate_1 at sink_1
  - plate_2 at sink_1

Initial Condition (State 1):
in(`spoon_1`, `dishwasher_1`)
in(`plate_1`, `sink_1`)
in(`plate_2`, `sink_1`)

State 2:
is_in_hand(`spoon_1`)

State 3:
at(`sink_1`)
is_dirty('spoon_1')

State 4:
is_on((`tap_1`)

State 5:
is_foamy(`sponge_1`)

State 6:
is_soapy(`spoon_1`)

State 7:
is_clean(`spoon_1`)

State 8:
in(`spoon_1`, `sink_1`)

State 9:
is_off(`tap_1`)
is_in_hand(`plate_1`)
is_dirty(`plate_1`)

State 10:
is_on((`tap_1`)
is_in_hand(`sponge_1`)
is_foamy(`sponge_1`)

State 11:
is_soapy(`plate_1`)

State 12:
is_clean(`plate_1`)

State 13:
in(`plate_1`, `sink_1`)

State 14:
is_off(`tap_1`)

State 15:
is_in_hand(`plate_2`)
is_dirty(`plate_2`)

State 16:
is_on((`tap_1`)

State 17:
is_foamy(`sponge_1`)

State 18:
is_soapy(`plate_2`)
```

```
   State 19:
   is_clean(`plate_2`)
   in(`plate_2`, `sink_1`)
   <end_of_example_query>
   [[Reasoning:]]
   The input trajectory contains state predicates because the
trajectory talks about the status of the objects (in(`spoon_1`, `
dishwasher_1`)).
   I will summarize the state trajectory into low-level actions.
   Low-level actions are not compacted enough yet because low-level
 actions can still be combined into high-level subtasks.
   The new trajectory will NOT be sufficiently summarized.
   [[Is the new trajectory sufficiently summarized? (yes/no):]]
   no
   [[Summarized trajectory:]]
   * In [Scenario 1], at state 1-2, the robot picked up spoon_1
from dishwasher_1.
   * At state 2-3, the robot moved from dishwasher_1 to sink_1.
   * At state 3-4, the robot turned on tap_1.
   * At state 4-5, the robot picked up sponge_1, which was foamy.
   * At state 5-6, the robot soaped spoon_1 until it became soapy
because spoon_1 was dirty.
   * At state 6-7, the robot rinsed spoon_1 until it became clean.
   * At state 7-8, the robot placed spoon_1 in sink_1. The robot
placed sponge_1 in sink_1.
   * At state 8-9, the robot turned off tap_1. The robot picked up
plate_1.
   * At state 9-10, the robot turned on tap_1. The robot picked up
sponge_1, which was foamy.
   * At state 10-11 ,the robot soaped plate_1 until it became soapy
 because plate_1 was dirty.
   * At state 11-12, the robot rinsed plate_1 until it became clean
.
   * At state 12-13, the robot placed plate_1 in sink_1. The robot
placed sponge_1 in sink_1.
   * At state 13-14, the robot turned off tap_1.
   * At state 14-15, the robot picked up plate_2.
   * At state 15-16, the robot turned on tap_1.
   * At state 16-17, the robot picked up sponge_1, which was foamy.
   * At state 17-18, the robot soaped plate_2 until it became soapy
 because plate_2 was dirty.
   * At state 18-19, the robot rinsed plate_2 until it became clean
. The robot placed plate_2 in sink_1. The robot turned off tap_1.
   [[end of response]]
 - |
   [[Input Trajectory:]]
   [Scenario 1]
   * In [Scenario 1], at state 1-2, the robot picked up spoon_1.
The robot picked up fork_1.
   * At state 2-3, the robot moved from countertop_1 to sink_1.
   * At state 3-4, the robot turned on tap_1. The robot soaped
fork_1 until it became soapy because fork_1 was dirty.
   * At state 4-5, the robot soaped spoon_1 until it became soapy
because spoon_1 was dirty.
   * At state 5-6, the robot rinsed spoon_1 until it became clean.
   * At state 6-7, the robot rinsed fork_1 until it became clean.
   * At state 7-8, the robot turned off tap_1. The robot placed
spoon_1 in sink_1. The robot placed fork_1 in sink_1.
   <end_of_example_query>
   [[Reasoning:]]
   The input trajectory contains low-level actions because the
trajectory mentions "picked up","moved", etc.
   I will summarize the low-level action trajectory into high-level
 subtasks.
```

High-level subtasks are compacted enough because they cannot be
combined together without losing important information.
  The new trajectory will be sufficiently summarized.
  [[Is the new trajectory sufficiently summarized? (yes/no):]]
  yes
  [[Summarized trajectory:]]
  * In [Scenario 1], at state 1-3, the robot brought spoon_1 and
fork_1 from countertop_1 to the sink_1.
  * At state 3-4, turned on tap_1.
  * At state 3-4, soaped fork_1 until it became soapy
  * At state 4-5, soaped spoon_1 until it became soapy
  * At state 5-6, rinsed spoon_1 until it became clean.
  * At state 6-7, rinsed fork_1 until it became clean.
  * At state 7-8, turned off tap_1, and placed spoon_1 and fork_1
in sink_1.
  [[end of response]]
 - |
  [[Input Trajectory:]]
  [Scenario 1]
  * In [Scenario 1], at state 1-2, the robot picked up spoon_1
from dishwasher_1.
  * At state 2-3, the robot moved from dishwasher_1 to sink_1.
  * At state 3-4, the robot turned on tap_1.
  * At state 4-5, the robot picked up sponge_1, which was foamy.
  * At state 5-6, the robot soaped spoon_1 until it became soapy
because spoon_1 was dirty.
  * At state 6-7, the robot rinsed spoon_1 until it became clean.
  * At state 7-8, the robot placed spoon_1 in sink_1. The robot
placed sponge_1 in sink_1.
  * At state 8-9, the robot turned off tap_1. The robot picked up
plate_1.
  * At state 9-10, the robot turned on tap_1. The robot picked up
sponge_1, which was foamy.
  * At state 10-11 ,the robot soaped plate_1 until it became soapy
 because plate_1 was dirty.
  * At state 11-12, the robot rinsed plate_1 until it became clean
.
  * At state 12-13, the robot placed plate_1 in sink_1. The robot
placed sponge_1 in sink_1.
  * At state 13-14, the robot turned off tap_1.
  * At state 14-15, the robot picked up plate_2.
  * At state 15-16, the robot turned on tap_1.
  * At state 16-17, the robot picked up sponge_1, which was foamy.
  * At state 17-18, the robot soaped plate_2 until it became soapy
 because plate_2 was dirty.
  * At state 18-19, the robot rinsed plate_2 until it became clean
. The robot placed plate_2 in sink_1. The robot turned off tap_1.
  <end_of_example_query>
  [[Reasoning:]]
  The input trajectory contains low-level actions because the
trajectory mentions "picked up","moved", etc.
  I will summarize the low-level action trajectory into high-level
 subtasks.
  High-level subtasks are compacted enough because they cannot be
combined together without losing important information.
  The new trajectory will be sufficiently summarized.
  [[Is the new trajectory sufficiently summarized? (yes/no):]]
  yes
  [[Summarized trajectory:]]
  * In [Scenario 1], at state 1-3, brought spoon_1 from
dishwasher_1 to sink_1
  * At state 3-4, turned on tap_1.
  * At state 4-6, soaped spoon_1 until it became soapy
  * At state 6-7, rinsed spoon_1 until it became clean
  * At state 7-8, placed spoon_1 in sink_1

```
      * At state 8-9, turned off tap_1 and picked up plate_1
      * At state 9-10, turned on tap_1.
      * At state 10-11, soaped plate_1 until it became soapy
      * At state 11-12, rinsed plate_1 until it became clean
      * At state 12-13, placed plate_1 in sink_1
      * At state 13-15, turned off tap_1 and picked up plate_2
      * At state 15-16, turned on tap_1
      * At state 16-18, soaped plate_2 until it became soapy
      * At state 18-19, rinsed plate_2 until it became clean, turned
   off tap_1, and placed plate_2 in sink_1
      [[end of response]]
```

## Summarized Demonstrations -> Task Specification

```
summary_2_spec:
  main: |
    You are a helpful assistant who analyzes the trajectories and
    summarizes them into a concise pseudocode-style task specification
    .
    <end_of_system_message>
    You are given (1) a high-level goal and (2) one or more
    trajectories where each one represents a different scenario. Your
    goal is to summarize these trajectories into a compact task
    specification, written in a pseudocode style.

    You must respond using the following format (you must generate [[
    end of response]] at the very end):
    [[Reasoning:]]
    You should first list out the order of the high-level subtask
    trajectories in all scenarios. Then, you should consider if
    certain sections of the subtasks are repeated in the scenario and
    can be represented by a loop. Two scenarios having exactly the
    same list does not mean that there is a loop. A loop only exists
    among subtasks in one individual scenario.
    Overall, your task specification should work for all scenarios.
    You should make sure that the task specification matches the
    subtasks ordering across all scenarios. You should also make sure
    that the task specification uses loops when there is any
    repetition.
    [[Task Specification:]]
    You should first state the high-level goal. Then, you should say "
    Specifically:" before outputting the pseudocode-style task
    specification.
    [[end of response]]

    You must follow these rules when you are writing the task
    specifications.
    * You must write the task specifications in pseudocode style. You
    should not write the task specification as a list. You cannot
    number any line.
  examples:
    - |
      [[High-Level Goal:]]
      Wash objects at a sink. All cleaned objects should be placed in
    the sink.
      Objects to clean
        - spoon_1 at countertop_1
        - fork_1 at countertop_1

      Initial Condition (State 1):
      at(`countertop_1`)
      is_dirty(`spoon_1`)
      is_dirty(`fork_1`)
```

```
    near(`spoon_1`)
    near(`fork_1`)
    [[Trajectories:]]
    * In [Scenario 1], at state 1-3, the robot brought spoon_1 and
fork_1 from countertop_1 to the sink_1.
    * At state 3-4, turned on tap_1
    * At state 3-4, soaped fork_1 until it became soapy
    * At state 4-5, soaped spoon_1 until it became soapy
    * At state 5-6, rinsed spoon_1 until it became clean.
    * At state 6-7, rinsed fork_1 until it became clean.
    * At state 7-8, turned off tap_1, and placed spoon_1 and fork_1
in sink_1.
    <end_of_example_user_query>
    [[Reasoning:]]
    * There are 2 dishes that got washed: [fork_1, spoon_1]
    * The list of high level actions that happened in order is: [
move dishes from A to B, turn on tap, soap, soap, rinse, rinse,
turn off tap, placed dishes in sink_1]
    * Because in [soap, soap], the high-level action of soapping got
 repeated twice (once for each dish we brought) we can use a for-
loop.
    * Because in [rinse, rinse], the high-level action rinsing gets
repeated twice (once for each dish), we can use a for-loop.
    * Since the dishes can be kept in hand, there is no     place
and   pick    u p   before soaping or rinsing.
    * Rinsing involves the use of tap water, which is why it was
turned on at some point before the rinse cycle, and turned off
after.
    [[Task Specification:]]
    Wash objects at a sink. All cleaned objects should be placed in
the sink.
    Specifically -

    Get a list of all objects to wash
    Bring all objects from countertop_1 to sink_1
    Turn on tap
    For each object
        Soap object
    For each object
        Rinse object
    Turn off tap
    For each object
        Place object in sink_1
    [[end of response]]
 - |
    [[High-Level Goal:]]
    Wash objects at a sink. All cleaned objects should be placed in
the sink.
    Objects to clean
      - spoon_1 at countertop_1
      - fork_1 at countertop_1

    Initial Condition (State 1):
    in(`spoon_1`, `dishwasher_1`)
    in(`plate_1`, `sink_1`)
    in(`plate_2`, `sink_1`)
    [[Trajectories:]]
    * In [Scenario 1], at state 1-3, brought spoon_1 from
dishwasher_1 to sink_1
    * At state 3-4, turned on tap_1.
    * At state 4-6, soaped spoon_1 until it became soapy
    * At state 6-7, rinsed spoon_1 until it became clean
    * At state 7-8, placed spoon_1 in sink_1
    * At state 8-9, turned off tap_1 and picked up plate_1
    * At state 9-10, turned on tap_1.
```

```
   * At state 10-11, soaped plate_1 until it became soapy
   * At state 11-12, rinsed plate_1 until it became clean
   * At state 12-13, placed plate_1 in sink_1
   * At state 13-15, turned off tap_1 and picked up plate_2
   * At state 15-16, turned on tap_1
   * At state 16-18, soaped plate_2 until it became soapy
   * At state 18-19, rinsed plate_2 until it became clean, turned
off tap_1, and placed plate_2 in sink_1
   <end_of_example_user_query>
   [[Reasoning:]]
   * There are 3 dishes got washed: [spoon_1, plate_1, plate_2]
   * The list of high level actions that happened in order is: [
move dish from A to B, turn on tap, soap, rinse, place, turn off
tap, pick up, turn on tap, soap, rinse, place, turn off tap, pick
up, turn on tap, soap, rinse, place, turn off tap]
   * Only spoon_1 is brought to the sink from the dishwasher, other
 dishes are already in the sink.
   * The spoon_1 does not have a pick_up action associated with it
because its already in hand when brought from dishwasher_1 to
sink_1. The action can be added to the code for generalizing
without a side effect.
   * The actions [pick_up, turn on tap, soap, rinse, place, turn
off tap] are repeated for each dish, so we can use a loop.
   * Rinsing involves the use of tap water, which is why it is
turned on at some point before the rinse cycle, and turned off
after.
   [[Task Specification:]]
   Wash objects at a sink. All cleaned objects should be placed in
the sink.
   Specifically -

   Get a list of all objects to wash
   Bring all objects to sink_1
   For each object in all objects:
     Pick_up object
     Turn on tap_1
     Soap object
     Rinse object
     Place object in sink_1
     Turn off tap_1
   [[end of response]]
```

### 18.3.2 Recursive Expansion

```
spec_2_highlevelcode:
  main: |
    You are a Python code generator for robotics. The users will first
     provide the imported Python modules. Then, for each code they
    want you to generate, they provide the requirements and pseudocode
     in Python docstrings.
    <end_of_system_message>
    You need to write robot control scripts in Python code. The Python
     code should be general and applicable to different environments.

    Below are the imported Python libraries and functions that you can
     use. You CANNOT import new libraries.
    ```
    # Python kitchen robot control script
    from env_utils import get_all_objects
    from robot_utils import bring_objects_to_loc, turn_off, turn_on,
    soap, rinse, pick_up, place, go_to, clean_with
    ```
```

Below shows the docstrings `for` these imported library functions
that you must follow. You CANNOT add additional parameters to
these functions.
```
"""
All the code should follow the specification. Follow descriptions
when writing code.

Specifications:
get_all_objects()
    Return:
        (list) a list of string of objects that need to be cleaned
bring_objects_to_loc(obj, loc)
    involves calling pick_up, go_to and place within the function.
    Parameters:
        obj (List[str]): Strings of the form "object_id" (e.g. "
plate_1") that need to be brought to the location loc
        loc(str): location string of the form "loc_id" (e.g. "sink_1
")
    Return:
        (void)
turn_off(tap_name)
    turns off tap
    Parameters:
        tap_name (str): A string that has the type of a tap (must be
 'tap') and an id. For example: 'tap1', 'tap2'
turn_on(tap_name)
    turns on tap
    Parameters:
        tap_name (str): A string that has the type of a tap (must be
 'tap') and an id. For example: 'tap1', 'tap2'
soap(obj)
    soap the object "obj". It must be in hand before calling this
function.
    Parameters:
        obj (str): String of the form "object_id" (e.g. "plate_1")
that needs to be saoped
rinse(obj)
    rinses the object "obj". It must be in hand before calling this
function, tap must be turned on before calling this function, and
switched off after
    Parameters:
        obj (str): String of the form "object_id" (e.g. "plate_1")
that needs to be rinsed
pick_up(obj)
    Parameters:
        obj (str): String of the form "object_id" (e.g. "plate_1")
to pick up
place(obj, loc)
    Parameters:
        obj (str): String of the form "object_id" (e.g. "plate_1")
to pick up
        loc (str): Location string (e.g. "sink_1") where obj is to
be placed
go_to(loc):
    Parameters
        loc (str): Location string (e.g. "sink_1") to go to from
current location.
clean_with(obj, tool)
    Parameters:
        obj (str): Object strings (e.g. "plate")  to clean
        tool (str): Tool strings (e.g. "sponge" or "towel")  to
clean with
"""
```

You must follow these rules when you are generating Python code.
* You MUST ONLY use Python library functions imported above. You
MUST follow the docstrings and specifications for these functions.
* You CANNOT call define new functions. You CANNOT use functions
that are NOT imported above.
* You must follow the instructions provided by the user. You
CANNOT add additional steps, conditionals, or loops that are not
in the instruction.
examples:
  - |
    ```
    """
    Wash objects at a sink. All cleaned objects should be placed in
    the sink.
    Specifically -

    Get a list of all objects to wash
    Bring all objects from countertop_1 to sink_1
    Turn on tap_1
    For each object
      Soap object
    For each object
      Rinse object
    Turn off tap_1
    For each object
      Place object in sink_1
    """
    ```
    <end_of_user_query>
    ```
    objects = get_all_objects()
    bring_objects_to_loc(objects, "sink_1")
    turn_on("tap_1")
    for object in objects:
      soap(object)
    for object in objects:
      rinse(object)
    turn_off("tap_1")
    for object in objects:
      place(object, "sink_1")
    ```
  - |
    ```
    """
    Wash objects at a sink. All cleaned objects should be placed in
    the sink.
    Specifically -

    Get a list of all objects to wash
    Bring all objects to sink_1
    For each object in all objects:
      Pick_up object
      Turn on tap_1
      Soap object
      Rinse object
      Place object in sink_1
      Turn off tap_1
    """
    ```
    <end_of_user_query>
    ```
    objects = get_all_objects()
    bring_objects_to_loc(objects, "sink_1")
    for object in objects:
    ```

```
        pick_up(object)
        turn_on("tap_1")
        soap(object)
        rinse(object)
        place(object, "sink_1")
        turn_off("tap_1")
    ```
```

# 19   Other Long Examples

## 19.1   Example Robotouille Query

```
[Scenario 1]
Make a burger.

State 2:
'patty1' is not at 'table1'
'robot1' is holding 'patty1'

State 3:
'robot1' is at 'stove2'
'robot1' is not at 'table1'

State 4:
'patty1' is at 'stove2'
'robot1' is not holding 'patty1'

State 5:

State 6:

State 7:

State 8:
'patty1' is cooked

State 9:
'patty1' is not at 'stove2'
'robot1' is holding 'patty1'

State 10:
'robot1' is not at 'stove2'
'robot1' is at 'table3'

State 11:
'patty1' is at 'table3'
'patty1' is on top of 'bottom_bun1'
'robot1' is not holding 'patty1'

State 12:
'robot1' is not at 'table3'
'robot1' is at 'table6'

State 13:
'tomato1' is not at 'table6'
'robot1' is holding 'tomato1'
```

```
State 14:
'robot1' is not at 'table6'
'robot1' is at 'cutting_board1'

State 15:
'tomato1' is at 'cutting_board1'
'robot1' is not holding 'tomato1'

State 16:

State 17:

State 18:
'tomato1' is cut

State 19:
'tomato1' is not at 'cutting_board1'
'robot1' is holding 'tomato1'

State 20:
'robot1' is at 'table3'
'robot1' is not at 'cutting_board1'

State 21:
'tomato1' is at 'table3'
'tomato1' is on top of 'patty1'
'robot1' is not holding 'tomato1'

State 22:
'robot1' is at 'table5'
'robot1' is not at 'table3'

State 23:
'lettuce1' is not at 'table5'
'robot1' is holding 'lettuce1'

State 24:
'robot1' is not at 'table5'
'robot1' is at 'cutting_board1'

State 25:
'lettuce1' is at 'cutting_board1'
'robot1' is not holding 'lettuce1'

State 26:

State 27:

State 28:
'lettuce1' is cut

State 29:
'lettuce1' is not at 'cutting_board1'
'robot1' is holding 'lettuce1'

State 30:
'robot1' is at 'table3'
'robot1' is not at 'cutting_board1'

State 31:
'lettuce1' is at 'table3'
```

```
'lettuce1' is on top of 'tomato1'
'robot1' is not holding 'lettuce1'

State 32:
'robot1' is at 'table4'
'robot1' is not at 'table3'

State 33:
'top_bun1' is not at 'table4'
'robot1' is holding 'top_bun1'

State 34:
'robot1' is not at 'table4'
'robot1' is at 'table3'

State 35:
'top_bun1' is at 'table3'
'top_bun1' is on top of 'lettuce1'
'robot1' is not holding 'top_bun1'

[Scenario 2]
Make a burger.

State 2:
'patty3' is not at 'table6'
'robot1' is holding 'patty3'

State 3:
'robot1' is at 'stove3'
'robot1' is not at 'table6'

State 4:
'patty3' is at 'stove3'
'robot1' is not holding 'patty3'

State 5:

State 6:

State 7:

State 8:
'patty3' is cooked

State 9:
'patty3' is not at 'stove3'
'robot1' is holding 'patty3'

State 10:
'robot1' is not at 'stove3'
'robot1' is at 'table5'

State 11:
'patty3' is at 'table5'
'patty3' is on top of 'bottom_bun3'
'robot1' is not holding 'patty3'

State 12:
'robot1' is at 'table3'
'robot1' is not at 'table5'

State 13:
```

```
'tomato3' is not at 'table3'
'robot1' is holding 'tomato3'

State 14:
'robot1' is not at 'table3'
'robot1' is at 'cutting_board3'

State 15:
'tomato3' is at 'cutting_board3'
'robot1' is not holding 'tomato3'

State 16:

State 17:

State 18:
'tomato3' is cut

State 19:
'tomato3' is not at 'cutting_board3'
'robot1' is holding 'tomato3'

State 20:
'robot1' is at 'table5'
'robot1' is not at 'cutting_board3'

State 21:
'tomato3' is at 'table5'
'tomato3' is on top of 'patty3'
'robot1' is not holding 'tomato3'

State 22:
'robot1' is at 'table7'
'robot1' is not at 'table5'

State 23:
'lettuce3' is not at 'table7'
'robot1' is holding 'lettuce3'

State 24:
'robot1' is not at 'table7'
'robot1' is at 'cutting_board3'

State 25:
'lettuce3' is at 'cutting_board3'
'robot1' is not holding 'lettuce3'

State 26:

State 27:

State 28:
'lettuce3' is cut

State 29:
'lettuce3' is not at 'cutting_board3'
'robot1' is holding 'lettuce3'

State 30:
'robot1' is at 'table5'
'robot1' is not at 'cutting_board3'
```

```
State 31:
'lettuce3' is at 'table5'
'lettuce3' is on top of 'tomato3'
'robot1' is not holding 'lettuce3'

State 32:
'robot1' is at 'table9'
'robot1' is not at 'table5'

State 33:
'top_bun3' is not at 'table9'
'robot1' is holding 'top_bun3'

State 34:
'robot1' is not at 'table9'
'robot1' is at 'table5'

State 35:
'top_bun3' is at 'table5'
'top_bun3' is on top of 'lettuce3'
'robot1' is not holding 'top_bun3'
```

## 19.2 Example EPIC-Kitchens Query

```
"""
[Scenario 1]
Wash objects in sink. All clean objects must be placed in drying rack.
Objects to clean
    - mezzaluna_1 in sink_1
    - peeler:potato_1 in sink_1
    - knife_1 in sink_1
    - board:cutting_1 in sink_1

Initial Condition (State 1):
is_in_hand(`sponge_1`)
in(`mezzaluna_1`, `sink_1`)
at(`sink_1`)

State 2:
is_in_hand(`mezzaluna_1`)
dirty(`mezzaluna_1`)

State 3:
is_soapy(`mezzaluna_1`)

State 4:
in(`mezzaluna_1`, `sink_2`)

State 5:
in(`peeler:potato_1`, `sink_1`)

State 6:
is_in_hand(`peeler:potato_1`)

State 7:
is_soapy(`peeler:potato_1`)

State 8:
in(`peeler:potato_1`, `sink_2`)
```

```
State 9:
in(`knife_1`, `sink_1`)

State 10:
is_in_hand(`knife_1`)

State 11:
is_soapy(`knife_1`)

State 12:
in(`knife_1`, `sink_2`)

State 13:
in(`board:cutting_1`, `sink_1`)

State 14:
is_in_hand(`board:cutting_1`)

State 15:
is_soapy(`board:cutting_1`)

State 16:
in(`board:cutting_1`, `sink_2`)

State 17:
is_on(`tap_1`)

State 18:
is_in_hand(`board:cutting_1`)

State 19:
is_clean(`board:cutting_1`)

State 20:
in(`board:cutting_1`, `dryingrack_1`)

State 21:
is_in_hand(`mezzaluna_1`)

State 22:
is_clean(`mezzaluna_1`)

State 23:
in(`mezzaluna_1`, `dryingrack_1`)

State 24:
is_in_hand(`peeler:potato_1`)

State 25:
is_clean(`peeler:potato_1`)

State 26:
in(`peeler:potato_1`, `dryingrack_1`)

State 27:
is_in_hand(`knife_1`)

State 28:
is_clean(`knife_1`)

State 29:
in(`knife_1`, `dryingrack_1`)

State 30:
is_off(`tap_1`)
```

```
"""
```

## 19.3 Intermediate Reasoning Ablation Helper Functions

### 19.3.1 No Reasoning Helper Functions

```python
1  def move_then_unstack(obj_to_unstack, obj_at_bottom, unstack_location)
       :
2      # For unstacking, we need to move to the location of the bottom
       object
3      if get_curr_location() != get_obj_location(obj_at_bottom):
4          move(get_curr_location(), get_obj_location(obj_at_bottom))
5      unstack(obj_to_unstack, obj_at_bottom)
6      # After unstacking, we need to move to the unstack_location
7      if get_curr_location() != unstack_location:
8          move(get_curr_location(), unstack_location)
9
10 def move_then_pick(obj):
11     obj_location = get_obj_location(obj)
12     if get_curr_location() != obj_location:
13         move(get_curr_location(), obj_location)
14     pick_up(obj, obj_location)
15
16 def move_then_place(obj, place_location):
17     if get_curr_location() != place_location:
18         move(get_curr_location(), place_location)
19     place(obj, place_location)
20
21 def cook_until_is_cooked(obj):
22     start_cooking(obj)
23     while not is_cooked(obj):
24         noop()
25
26 def cook_object_at_location(obj, location):
27     # To cook an object, the robot first needs to be holding obj
28     if not is_holding(obj):
29         # If the robot is not holding obj, there are 2 scenarios:
30         #    (1) if obj is in a stack ,unstack obj
31         #    (2) else, pick up obj.
32         if is_in_a_stack(obj):
33             # Because obj is in a stack, robot need to move then
       unstack the obj from the obj_at_bottom first
34             obj_at_bottom = get_obj_that_is_underneath(obj_at_top=obj)
35             move_then_unstack(obj_to_unstack=obj, obj_at_bottom=
       obj_at_bottom, unstack_location=get_obj_location(obj_at_bottom))
36         else:
37             # Since obj is not in a stack, robot can just move then
       pick it up
38             move_then_pick(obj=obj)
39     # place the object at the location to cook at
40     move_then_place(obj=obj, place_location=location)
41     # cook the object
42     cook_until_is_cooked(obj=obj)
43
44 def cut_until_is_cut(obj):
45     while not is_cut(obj):
46         cut(obj)
47
48 def cut_object_at_location(obj, location):
49     # To cut an object, the robot first needs to be holding obj
50     if not is_holding(obj):
```

```
51          # If the robot is not holding obj, there are 2 scenarios:
52          #   (1) if obj is in a stack ,unstack obj
53          #   (2) else, pick up obj.
54          if is_in_a_stack(obj):
55              # Because obj is in a stack, robot need to move then
        unstack the obj from the obj_at_bottom first
56              obj_at_bottom = get_obj_that_is_underneath(obj_at_top=obj)
57              move_then_unstack(obj_to_unstack=obj, obj_at_bottom=
        obj_at_bottom, unstack_location=get_obj_location(obj_at_bottom))
58          else:
59              # Since obj is not in a stack, robot can just move then
        pick it up
60              move_then_pick(obj=obj)
61          # place the object at the location to cut at
62          move_then_place(obj=obj, place_location=location)
63          # cut the object
64          cut_until_is_cut(obj=obj)
65
66  def move_then_stack(obj_to_stack, obj_at_bottom, stack_location):
67      if get_curr_location() != stack_location:
68          move(get_curr_location(), stack_location)
69      stack(obj_to_stack, obj_at_bottom)
70
71  def stack_obj1_on_obj2(obj1, obj2):
72      # To stack obj1 on obj2, the robot needs to be holding obj1
73      if not is_holding(obj1):
74          # If the robot is not holding obj1, there are 2 scenarios:
75          #   (1) if obj1 is in a stack ,unstack obj1
76          #   (2) else, pick up obj1.
77          if is_in_a_stack(obj1):
78              # Because obj1 is in a stack, robot need to move then
        unstack the obj from the obj_at_bottom first
79              obj_at_bottom = get_obj_that_is_underneath(obj_at_top=obj1
        )
80              move_then_unstack(obj_to_unstack=obj1, obj_at_bottom=
        obj_at_bottom, unstack_location=get_obj_location(obj_at_bottom))
81          else:
82              # Since obj1 is not in a stack, robot can just move then
        pick it up
83              move_then_pick(obj=obj1)
84      # determine the location of obj2 to stack on
85      obj2_location = get_obj_location(obj2)
86      # move to obj2's location then stack obj1 on obj2
87      move_then_stack(obj_to_stack=obj1, obj_at_bottom=obj2,
        stack_location=obj2_location)
```

### 19.3.2   Only List Helper Functions

```
1  def move_then_unstack(obj_to_unstack, obj_at_bottom, unstack_location)
        :
2      # For unstacking, we need to move to the location of the bottom
        object
3      if get_curr_location() != get_obj_location(obj_at_bottom):
4          move(get_curr_location(), get_obj_location(obj_at_bottom))
5      unstack(obj_to_unstack, obj_at_bottom)
6      # After unstacking, we need to move to the unstack_location
7      if get_curr_location() != unstack_location:
8          move(get_curr_location(), unstack_location)
9
10  def move_then_pick(obj):
11      obj_location = get_obj_location(obj)
12      if get_curr_location() != obj_location:
13          move(get_curr_location(), obj_location)
```

```
14      pick_up(obj, obj_location)

16  def move_then_place(obj, place_location):
17      if get_curr_location() != place_location:
18          move(get_curr_location(), place_location)
19      place(obj, place_location)

21  def cook_until_is_cooked(obj):
22      start_cooking(obj)
23      while not is_cooked(obj):
24          noop()

26  def cook_object_at_location(obj, location):
27      # To cook an object, the robot first needs to be holding obj
28      if not is_holding(obj):
29          # If the robot is not holding obj, there are 2 scenarios:
30          #    (1) if obj is in a stack ,unstack obj
31          #    (2) else, pick up obj.
32          if is_in_a_stack(obj):
33              # Because obj is in a stack, robot need to move then
        unstack the obj from the obj_at_bottom first
34              obj_at_bottom = get_obj_that_is_underneath(obj_at_top=obj)
35              move_then_unstack(obj_to_unstack=obj, obj_at_bottom=
        obj_at_bottom, unstack_location=get_obj_location(obj_at_bottom))
36          else:
37              # Since obj is not in a stack, robot can just move then
        pick it up
38              move_then_pick(obj=obj)
39      # place the object at the location to cook at
40      move_then_place(obj=obj, place_location=location)
41      # cook the object
42      cook_until_is_cooked(obj=obj)

44  def move_then_stack(obj_to_stack, obj_at_bottom, stack_location):
45      if get_curr_location() != stack_location:
46          move(get_curr_location(), stack_location)
47      stack(obj_to_stack, obj_at_bottom)

49  def stack_obj1_on_obj2(obj1, obj2):
50      # To stack obj1 on obj2, the robot needs to be holding obj1
51      if not is_holding(obj1):
52          # If the robot is not holding obj1, there are 2 scenarios:
53          #    (1) if obj1 is in a stack ,unstack obj1
54          #    (2) else, pick up obj1.
55          if is_in_a_stack(obj1):
56              # Because obj1 is in a stack, robot need to move then
        unstack the obj from the obj_at_bottom first
57              obj_at_bottom = get_obj_that_is_underneath(obj_at_top=obj1
        )
58              move_then_unstack(obj_to_unstack=obj1, obj_at_bottom=
        obj_at_bottom, unstack_location=get_obj_location(obj_at_bottom))
59          else:
60              # Since obj1 is not in a stack, robot can just move then
        pick it up
61              move_then_pick(obj=obj1)
62      # determine the location of obj2 to stack on
63      obj2_location = get_obj_location(obj2)
64      # move to obj2's location then stack obj1 on obj2
65      move_then_stack(obj_to_stack=obj1, obj_at_bottom=obj2,
        stack_location=obj2_location)

67  def move_to_location(location):
68      if get_curr_location() != location:
69          move(get_curr_location(), location)

70
```

```
71 def cut_object(obj, location):
72     # To cut an object, the robot first needs to be holding obj
73     if not is_holding(obj):
74         # If the robot is not holding obj, there are 2 scenarios:
75         #    (1) if obj is in a stack ,unstack obj
76         #    (2) else, pick up obj.
77         if is_in_a_stack(obj):
78             # Because obj is in a stack, robot need to move then
    unstack the obj from the obj_at_bottom first
79             obj_at_bottom = get_obj_that_is_underneath(obj_at_top=obj)
80             move_then_unstack(obj_to_unstack=obj, obj_at_bottom=
    obj_at_bottom, unstack_location=get_obj_location(obj_at_bottom))
81         else:
82             # Since obj is not in a stack, robot can just move then
    pick it up
83             move_then_pick(obj=obj)
84     # move to the location to cut at
85     move_to_location(location=location)
86     # cut the object
87     cut(obj=obj)
```

### 19.3.3 Full Helper Functions

```
1 def move_then_unstack(obj_to_unstack, obj_at_bottom, unstack_location)
    :
2     # For unstacking, we need to move to the location of the bottom
    object
3     if get_curr_location() != get_obj_location(obj_at_bottom):
4         move(get_curr_location(), get_obj_location(obj_at_bottom))
5     unstack(obj_to_unstack, obj_at_bottom)
6     # After unstacking, we need to move to the unstack_location
7     if get_curr_location() != unstack_location:
8         move(get_curr_location(), unstack_location)
9
10 def move_then_pick(obj):
11     obj_location = get_obj_location(obj)
12     if get_curr_location() != obj_location:
13         move(get_curr_location(), obj_location)
14     pick_up(obj, obj_location)
15
16 def move_then_place(obj, place_location):
17     if get_curr_location() != place_location:
18         move(get_curr_location(), place_location)
19     place(obj, place_location)
20
21 def cook_until_is_cooked(obj):
22     start_cooking(obj)
23     while not is_cooked(obj):
24         noop()
25
26 def cook_object_at_location(obj, location):
27     # To cook an object, the robot first needs to be holding obj
28     if not is_holding(obj):
29         # If the robot is not holding obj, there are 2 scenarios:
30         #    (1) if obj is in a stack ,unstack obj
31         #    (2) else, pick up obj.
32         if is_in_a_stack(obj):
33             # Because obj is in a stack, robot need to move then
    unstack the obj from the obj_at_bottom first
34             obj_at_bottom = get_obj_that_is_underneath(obj_at_top=obj)
35             move_then_unstack(obj_to_unstack=obj, obj_at_bottom=
    obj_at_bottom, unstack_location=get_obj_location(obj_at_bottom))
36         else:
```

```
37              # Since obj is not in a stack , robot can just move then
        pick it up
38              move_then_pick ( obj = obj )
39          # place the object at the location to cook at
40          move_then_place ( obj = obj , place_location = location )
41          # cook the object
42          cook_until_is_cooked ( obj = obj )
43
44  def move_then_stack ( obj_to_stack , obj_at_bottom , stack_location ):
45      if get_curr_location () != stack_location :
46          move ( get_curr_location () , stack_location )
47      stack ( obj_to_stack , obj_at_bottom )
48
49  def stack_obj1_on_obj2 ( obj1 , obj2 ):
50      # To stack obj1 on obj2 , the robot needs to be holding obj1
51      if not is_holding ( obj1 ):
52          # If the robot is not holding obj1 , there are 2 scenarios :
53          #    (1) if obj1 is in a stack , unstack obj1
54          #    (2) else , pick up obj1 .
55          if is_in_a_stack ( obj1 ):
56              # Because obj1 is in a stack , robot need to move then
        unstack the obj from the obj_at_bottom first
57              obj_at_bottom = get_obj_that_is_underneath ( obj_at_top = obj1
        )
58              move_then_unstack ( obj_to_unstack = obj1 , obj_at_bottom =
        obj_at_bottom , unstack_location = get_obj_location ( obj_at_bottom ))
59          else :
60              # Since obj1 is not in a stack , robot can just move then
        pick it up
61              move_then_pick ( obj = obj1 )
62      # determine the location of obj2 to stack on
63      obj2_location = get_obj_location ( obj2 )
64      # move to obj2 's location then stack obj1 on obj2
65      move_then_stack ( obj_to_stack = obj1 , obj_at_bottom = obj2 ,
        stack_location = obj2_location )
66
67  def cut_until_is_cut ( obj ):
68      while not is_cut ( obj ):
69          cut ( obj )
70
71  def cut_object_at_location ( obj , location ):
72      # To cut an object , the robot first needs to be holding obj
73      if not is_holding ( obj ):
74          # If the robot is not holding obj , there are 2 scenarios :
75          #    (1) if obj is in a stack , unstack obj
76          #    (2) else , pick up obj .
77          if is_in_a_stack ( obj ):
78              # Because obj is in a stack , robot need to move then
        unstack the obj from the obj_at_bottom first
79              obj_at_bottom = get_obj_that_is_underneath ( obj_at_top = obj )
80              move_then_unstack ( obj_to_unstack = obj , obj_at_bottom =
        obj_at_bottom , unstack_location = get_obj_location ( obj_at_bottom ))
81          else :
82              # Since obj is not in a stack , robot can just move then
        pick it up
83              move_then_pick ( obj = obj )
84      # place the object at the location to cut at
85      move_then_place ( obj = obj , place_location = location )
86      # cut the object
87      cut_until_is_cut ( obj = obj )
```

### 19.3.4 Only Analyze Helper Functions

```
1  def move_then_unstack(obj_to_unstack, obj_at_bottom, unstack_location)
       :
2      # For unstacking, we need to move to the location of the bottom
       object
3      if get_curr_location() != get_obj_location(obj_at_bottom):
4          move(get_curr_location(), get_obj_location(obj_at_bottom))
5      unstack(obj_to_unstack, obj_at_bottom)
6      # After unstacking, we need to move to the unstack_location
7      if get_curr_location() != unstack_location:
8          move(get_curr_location(), unstack_location)
9
10 def move_then_pick(obj):
11     obj_location = get_obj_location(obj)
12     if get_curr_location() != obj_location:
13         move(get_curr_location(), obj_location)
14     pick_up(obj, obj_location)
15
16 def move_then_place(obj, place_location):
17     if get_curr_location() != place_location:
18         move(get_curr_location(), place_location)
19     place(obj, place_location)
20
21 def cook_until_is_cooked(obj):
22     start_cooking(obj)
23     while not is_cooked(obj):
24         noop()
25
26 def cook_object_at_location(obj, location):
27     # To cook an object, the robot first needs to be holding obj
28     if not is_holding(obj):
29         # If the robot is not holding obj, there are 2 scenarios:
30         #   (1) if obj is in a stack ,unstack obj
31         #   (2) else, pick up obj.
32         if is_in_a_stack(obj):
33             # Because obj is in a stack, robot need to move then
       unstack the obj from the obj_at_bottom first
34             obj_at_bottom = get_obj_that_is_underneath(obj_at_top=obj)
35             move_then_unstack(obj_to_unstack=obj, obj_at_bottom=
       obj_at_bottom, unstack_location=get_obj_location(obj_at_bottom))
36         else:
37             # Since obj is not in a stack, robot can just move then
       pick it up
38             move_then_pick(obj=obj)
39     # place the object at the location to cook at
40     move_then_place(obj=obj, place_location=location)
41     # cook the object
42     cook_until_is_cooked(obj=obj)
43
44 def move_then_stack(obj_to_stack, obj_at_bottom, stack_location):
45     if get_curr_location() != stack_location:
46         move(get_curr_location(), stack_location)
47     stack(obj_to_stack, obj_at_bottom)
48
49 def stack_obj1_on_obj2(obj1, obj2):
50     # To stack obj1 on obj2, the robot needs to be holding obj1
51     if not is_holding(obj1):
52         # If the robot is not holding obj1, there are 2 scenarios:
53         #   (1) if obj1 is in a stack ,unstack obj1
54         #   (2) else, pick up obj1.
55         if is_in_a_stack(obj1):
56             # Because obj1 is in a stack, robot need to move then
       unstack the obj from the obj_at_bottom first
57             obj_at_bottom = get_obj_that_is_underneath(obj_at_top=obj1
       )
```

```
58                move_then_unstack(obj_to_unstack=obj1, obj_at_bottom=
         obj_at_bottom, unstack_location=get_obj_location(obj_at_bottom))
59            else:
60                # Since obj1 is not in a stack, robot can just move then
         pick it up
61                move_then_pick(obj=obj1)
62        # determine the location of obj2 to stack on
63        obj2_location = get_obj_location(obj2)
64        # move to obj2's location then stack obj1 on obj2
65        move_then_stack(obj_to_stack=obj1, obj_at_bottom=obj2,
         stack_location=obj2_location)

66
67  def cut_until_is_cut(obj):
68      while not is_cut(obj):
69          cut(obj)

70
71  def cut_object_at_location(obj, location):
72      # To cut an object, the robot first needs to be holding obj
73      if not is_holding(obj):
74          # If the robot is not holding obj, there are 2 scenarios:
75          #   (1) if obj is in a stack ,unstack obj
76          #   (2) else, pick up obj.
77          if is_in_a_stack(obj):
78              # Because obj is in a stack, robot need to move then
         unstack the obj from the obj_at_bottom first
79              obj_at_bottom = get_obj_that_is_underneath(obj_at_top=obj)
80              move_then_unstack(obj_to_unstack=obj, obj_at_bottom=
         obj_at_bottom, unstack_location=get_obj_location(obj_at_bottom))
81          else:
82              # Since obj is not in a stack, robot can just move then
         pick it up
83              move_then_pick(obj=obj)
84      # place the object at the location to cut at
85      move_then_place(obj=obj, place_location=location)
86      # cut the object
87      cut_until_is_cut(obj=obj)
```

