# OpenReview forum: "Demo2Code: From Summarizing Demonstrations to Synthesizing Code via Extended Chain-of-Thought"
_NeurIPS.cc/2023/Conference — NeurIPS 2023 poster_

### Official Review · Reviewer_yYo2 · 2023-06-15

**Soundness:** 3 good
**Presentation:** 3 good
**Contribution:** 3 good
**Rating:** 6
**Confidence:** 3

**Summary:**

This work presents a recursive method to summarize demonstrations into programs through LLM.  The idea is interesting in that it uses spec as the bottleneck to connect complex demonstrations and complex robot task code, encoded and decoded through chain of thoughts. The method is evaluated on three different benchmarks involving table-top manipulation, novel kitchen text tasks, and EpicKitchen. The method outperforms naive language-to-code baselines and can generalize to longer-horizon tasks as well as learning user intents.

**Strengths:**

1. The problem is challenging in that demonstrations and codes are both complex: demonstrations have lots of details and multimodality, yet codes are abstract and need to follow strict requirements.
2. The method is sound: it uses divide and conquer to tackle some limitations of the current LLM.
3. The new benchmark can be interesting to researchers that want to attempt the high-level planning problem in kitchen tasks.
4. The figures and the pseudo codes are helpful
5. The method generalize to longer-horizon tasks as well as learning user intents.


**Weaknesses:**

1. Have several concerns on the evaluation metrics
2. Need more details on discussing the tabletop benchmark, the EpicKitchen experiment and the new proposed benchmark. Looks like epic kitchen is closer to diverse raw data such as Youtube and yet the new kitchen simulator and the table-top tasks has more predicates as well as low-level relationships


**Questions:**

1. The formulation of MDP seem a bit unnecessary to me.
2. The spec is called “latent” multiple times in the paper, but shouldn’t it be clearly defined for each task?
3. To confirm on the evaluation metric: Is the unit-test pass rates the same as task success rates? I.e. does the generated code satisfy a set of instructions (task) defined with the demonstrations? Or is it more narrowed?
4. What is the quality of the spec2code, and why matching it is a necessary metric? It seems to me that there are many ways to write code for each task.
5. Is hallucination a problem? Because the method introduces such a pipeline instead of end-to-end, is it possible that the LLM introduces unnecessary steps in the instruction summary stage and then introduce more functions to complete those substeps in the expanding stage?
6. What is the reference code in EpicKitchen dataset in line 190. And what is the user scoring process?
7. Given that there could be multiple ways to do the breakdown and the LLM are not deterministic, is stochasty a concern? It seems that variance is not provided in the table for each experiment.
8. Have the authors tried the new GPT3.5 model with longer-context or GPT-4 model?

**Limitations:**

See above.

---

> ### Author Rebuttal · Authors · 2023-08-10
>
> We appreciate the reviewer for the questions which help us improve our paper's clarity. We are excited that the reviewer acknowledges how our approach solves a challenging problem and Robotouille as a promising benchmark. Please see below for our answers to the questions:
>
> ### Questions
> #### **Q2: Why is "latent task specification" not clearly defined for each task?**
> We wish to clarify the sentence (line 30-31): “while demonstrations are long and code is complex, they both share a latent task specification that the user had in mind.”
>
> We use "latent task specification" because:
> - “Latent” evokes the analogy with an encoder-decoder architecture that encodes demonstrations with the summarizer and decodes into code with the code expansion module.
> - "Task specification” refers to the detailed language description of how the task should be completed.
>
> This specification is not defined for each task because we assume that the user doesn't need to provide detailed instructions on how to complete a task. Instead, Demo2Code is responsible for "encoding" the demonstrations and generating this latent task specification.
>
> #### **Q3: Is the unit-test pass rates the same as the execution success rate?**
> No, they are different.
> - The execution success rate shows whether a code can be executed in the simulator without any error.
> - The unit-test pass rate is whether a code has completed the task in the same way that the user wants. The unit test checks whether the code has completed all the subgoals and satisfied all the constraints successfully.
>
> For example:
> - **When execution success rate = 0:** The generated code has syntax error, or if it violates the physical constraint in the simulator (e.g. trying to pick up item A from table B even though table B is empty).
> - **When unit-test pass rate = 0:** Consider a task to make a burger with a patty and lettuce. Not known to our pipeline, the user wants to prepare all the ingredients before assembling the burger, and they also want the lettuce to be on top of the patty. Even if a code makes a burger without having any error in the simulator (thereby having an execution success rate of 1), if it cooks the patty then immediately adds it to the burger instead of cutting the lettuce first, or if it puts the patty on top of the lettuce, the code will fail the unit test.
>
> #### **Q4: Why is matching Spec2Code a necessary metric?**
> Spec2Code assumes access to the latent task specification, so it knows exactly how to complete the task. We agree with the reviewer that matching Spec2Code isn't a necessary metric, as there are many possible ways to solve the task while satisfying the user's hidden requirements.
>
> However, we empirically found this metric to be a useful proxy metric of the quality of a code without having to actually run the code.
> - A high match rate indicates that the code is likely to succeed in execution and pass the unit test.
> - However, a low match rate doesn't imply the code is wrong. Hence, we rely on the execution success rate and unit-test pass rate to evaluate the correctness of the code.
>
> #### **Q5: Is hallucination a problem?**
> For our experiments, we didn't encounter explicit issues with hallucination although LLMs like GPT-3.5 do hallucinate at times [1]. We are able to avoid hallucination because when we show examples of how to summarize demonstrations, we make sure that the LLM would cite parts of the demonstrations that make up the summarized line.
>
> However, we do run into issues with the LLMs making mistakes during summarization, which consequently causes the task specification to be incorrect.
> - e.g. the LLMs can accidentally omit a high-level subtask like stacking a patty on top of the lettuce, thereby causing the specification to also miss this subtask.
>
> This limitation shows the value of future work to add a verification and improvement step so that our approach can catch errors it makes and improve its code [2-5].
>
> #### **Q6: What is the reference code and the pass rate for the EPIC-KITCHENS dataset?**
> The reference code for the EPIC-KITCHENS dataset is written by a human annotator after watching the demonstrations, and reviewed by another human to avoid inconsistency. Then, the LLM's generated code is compared against this reference code to compute the BLEU score (match score).
>
> The pass rate is similar to the unit-test pass rate in the tabletop environment and Robotouille. However, because we don't have access to a simulator to run the generated code, we rely on human annotators to check if the demonstrations fit as a trace of the generated code.
>
> #### **Q7: Is stochasticity a concern?**
> It is not a major concern because we use temperature = 0 for GPT-3.5, which keeps our generated output and the final code mostly deterministic.
>
> #### **Q8: Have the authors tried the new GPT-3.5 model with longer contexts or GPT-4 model?**
> While we didn't try the new GPT3.5 model with longer contexts or GPT-4, we hypothesize that the increased context length will help solve more complicated tasks with longer and larger amount of demonstrations.
>
> The multimodal reasoning capabilities of models like GPT-4 (which came out in parallel to us running the experiments) can also be more robust to noisy demonstrations and better solve real-world tasks like the ones in the EPIC-KITCHENS's dataset.
>
> [1] Mark Chen, et al. Evaluating large language models trained on code. arXiv:2107.03374, 2021.
>
> [2] Marta Skreta, et al. Errors are useful prompts: Instruction guided task programming with verifier-assisted iterative prompting, 2023.
>
> [3] Shreyas Sundara Raman, et al. Planning with large language models via corrective re-prompting, 2022.
>
> [4] Debjit Paul, et al. Refiner: Reasoning feedback on intermediate representations, 2023.
>
> [5] Noah Shinn, et al. Reflexion: Language agents with verbal reinforcement learning, 2023.

---

> > ### Author Response · Authors · 2023-08-21
> > **Thank you for your feedback**
> >
> > Hello! Thank you again for your feedback and questions! As the discussion period ends soon, please let us know if we can provide any additional clarifications or answers that would help you in your evaluation.

---

### Official Review · Reviewer_325h · 2023-07-05

**Soundness:** 3 good
**Presentation:** 3 good
**Contribution:** 2 fair
**Rating:** 6
**Confidence:** 4

**Summary:**

This paper presents demo2code, a framework that takes as input user's language instructions as well as demonstrations, and outputs synthesized code for completing the tasks. It first iteratively summarizes given demonstrations to a compact task specification, then reasons by incorporating user preferences etc, and lastly output expanded execution code. The method is evaluated on a range of tasks, including table top manipulation, a simple cooking simulator, and real-world epic kitchen dataset.

**Strengths:**

- The framework is novel in that it proposes to summarize demonstrations and language instructions using LLM, which is then used for action generation via code synthesis.
- the recursive and hierachical way of summarizing demonstrations and generating code is reasonable
- the idea that using LLM to reason about user preferences makes sense

**Weaknesses:**

- Upon reading the introduction i was excited to see how the approach is able to handle both instruction and demonstration: the latter usually comes in a visual space, but then i realize the authors made a big assumption that they can query the simulator to get state-based demonstration. This assumption presents a few issues: 1) such privileged information hinders the application in realistic settings. In fact, the author had to manually densely label the epic kitchen dataset, which leads to the question of how this can be used for real-world settings 2) even in simulation, it's not straightforawrd to obtain these state information. for example, as opposed to `op-top`, relations such as `in(obj, microwave)` is hard to obtain easily. Also, binary spatial relations loses dense geometric informations. 3) if access to step-by-step low-level state is assumed, and the LLM can summarize and generate step-by-step specifications, why not directly use task and motion planning(TAMP) to solve the task?
- how does the oracle spec2code works? does it use TAMP? if yes, what's the advantage of demo2code over it?
- the experiment which uses epickitchen, but need additional manual annotation, loses the point of evaluating on real-world dataset

**Questions:**

- the authors says the language instruction and demonstration shares a common latent space after summarizing, but i don't see any details on this. Did I miss something?
- also see weaknesses.

**Limitations:**

- assumption on access to privileged low-level state and relations
- additional annotation on real-world data
- other limitations are discussed in the last section

---

> ### Author Rebuttal · Authors · 2023-08-10
>
> We appreciate that the reviewer acknowledges our novelty in recursively summarizing demonstrations and hierarchically generating code. We would like to respond to the reviewer's helpful feedback and questions.
>
> ### Questions
> #### **Q1: Do you have any detail on the latent task specification?**
> In the paper's introduction section (line 30-31), we stated that "while demonstrations are long and code is complex, they both share a latent task specification that the user had in mind." Specifically, this task specification is a detailed language description of how the task should be completed. It is latent because we assume that the user does not directly provide it as the language instruction. Please refer to our global response for more details.
>
> ### Weakness
> #### **Clarification of assuming access to state-based demonstrations**
> We clarify the focus of our work in the global response. We envision that our pipeline will exist in conjunction with other perception modules that parse the environment and demonstrations into symbolic states. We make this assumption so that we can focus on our goal of generating robot task code from demonstrations. As a part of future work, we are integrating Demo2Code into an end-to-end robotics system.
>
> #### **Clarification on code v.s. using TAMP**
> We refer the reviewer to the global response on our justification of generating code. We chose code as a flexible closed-loop policy representation that can check conditionals, contain loops, and use existing utility functions. In contrast, TAMP produces an open loop plan that needs replanning if the environment changes.
>
> However, note that once Demo2Code's pipeline is used to summarize the demonstrations into a task specification, the downstream policy can be modified to other types of policy representation, e.g. PDDL planner, TAMP.
> - For example, we show the possibility of integrating Demo2Code with a symbolic planner LLM+P [1], a pipeline that uses LLMs to generate PDDL before calling a planner to solve the problem. Please refer to Figure 1 of the attached pdf for a qualitative example.
>
> #### **Clarification of the oracle Spec2Code**
> The oracle Spec2Code and our approach Demo2Code are different in the following way:
> - Spec2Code assumes having access to the latent task specification, which is the user's hidden information on how they want the tasks to be completed.
> - Demo2Code doesn't know the task specification. It needs to generate the specification by summarizing the input demonstrations.
>
> However, the second stage of Spec2Code and Demo2Code are the same: given a task specification, generating the robot task code and recursively defining any helper function.
>
> Spec2Code sets an upper bound for how good Demo2Code can be. Demo2Code attempts to generate a good enough task specification from demonstrations and language to achieve the same performance as Spec2Code.
>
> #### **What is the advantage of using Demo2Code over Spec2Code?**
> Using Spec2Code directly is cumbersome because it requires the human to give a very detailed description of how to do a task. In contrast, Demo2Code just needs the user to demonstrate the task. Thus, Demo2Code reduces the amount of effort that users need to spend in order to teach and interact with a robot.
>
> #### **Clarification of running experiments with EPIC-KITCHENS data**
> While we acknowledge relying on manual annotations for EPIC-KITCHEN is limiting, we argue that the experiments we provide are still quite valuable because:
> - The EPIC-KITCHENS is a real-world dataset, analogous to how a typical user would provide demonstrations once Demo2Code is integrated into an end-to-end robotics system.
> - The data covers a wide range of unique kitchens with different kitchenware and a diverse set of users who each has a preferred way of washing dishes.
> - The manually annotated states and actions have a wide range of objects, predicates, and actions.
>
> In our future work, we are working to integrate Demo2code with a vision language model that can automatically extract states and actions from the video data. In the meantime, our current experiments have verified that Demo2Code can extract different real-world users' unique preferences from the output of such perception models.
>
> We also ran ablation studies where we added noisy predicates (e.g. positions of irrelevant objects in the background) that a perception module might automatically identify in real-world demonstrations. Table 1 in Section G of the appendix shows that noisy demonstrations can worsen LLM's performance. Thus, the EPIC-KITCHENS dataset emphasizes the importance of future works on using feedback to iteratively improve Demo2Code's output [2-5].
>
> [1] Bo Liu, et al. LLM+P: Empowering large language models with optimal planning proficiency, 2023.
>
> [2] Marta Skreta, et al. Errors are useful prompts: Instruction guided task programming with verifier-assisted iterative prompting, 2023.
>
> [3] Shreyas Sundara Raman, et al. Planning with large language models via corrective re-prompting, 2022.
>
> [4] Debjit Paul, et al. Refiner: Reasoning feedback on intermediate representations, 2023.
>
> [5] Noah Shinn, et al. Reflexion: Language agents with verbal reinforcement learning, 2023.

---

> > ### Comment · Reviewer_325h · 2023-08-19
> >
> > I appreciate the authors' detailed response. I have increased my scores accordingly.

---

> > > ### Author Response · Authors · 2023-08-21
> > > **Thank you for your feedback**
> > >
> > > Thank you for your detailed review, questions, and suggestions!

---

### Official Review · Reviewer_ujZb · 2023-07-06

**Soundness:** 3 good
**Presentation:** 3 good
**Contribution:** 2 fair
**Rating:** 7
**Confidence:** 3

**Summary:**

This paper proposes Demo2Code, a new method for generating code given a natural language description and demonstrations of the task. Demo2Code recursively summarizes demonstrations using a language model (LM) to create a task specification. The task specification is concatenated to the description and then recursively synthesized into code using a LM. Demo2Code is shown to outperform previous SoTA on an object manipulation environment, an author-designed cooking simulator, and the EPIC Kitchens dataset. Finally, qualitative results are shown demonstrating OOD generalization, grounding, and understanding of user preferences.



**Strengths:**

*Originality:* a new long-range sequential decision-making benchmark, Robotouille, was developed that focuses more on high-level actions rather than manipulation and navigation. This may be useful for future work studying agents that learn from task specification.

*Quality:* the experiments are comprehensive and cover a range of tasks and domains, suggesting that the method is robust to distributions of tasks. Furthermore, the results show a marked improvement over the previous SoTA on all the tasks, which likely demonstrates the efficacy of leveraging demonstrations.

*Clarity:* the paper is well-written and the figures are self-explanatory.

*Significance:* generating code from demonstrations is an important step towards developing agents that can efficiently interact with humans. Moreover, this work demonstrates a working end-to-end setup from natural language demonstrations to code, likely encouraging future work in this area.

**Weaknesses:**

There are two implicit assumptions of the method that are not evaluated and may make the method difficult to use in practice. I believe that these assumptions may be hard to overcome, so I am leaning towards giving a 6.5 but am rounding up because of the high-quality execution.

1. *The method assumes that demonstrations are complete descriptions of each state in the trajectory.* I would imagine that in many real-world environments (which is the setting that such a method would likely be deployed in), obtaining a complete description of each state in the trajectory is noisy. Some actions may be occluded or unable to be clearly delineated. Moreover, it may not even be clear a priori how to canonically parse actions from video or a natural language description of the environment. Would it be possible to develop an ablation where some of the actions are noisily parsed or even entirely omitted from the demonstration? I suspect that GPT-3.5 may not be able to handle the perturbation, but GPT-4 may be able to.

2. *The method assumes that the task description is well-specified.* Humans can often provide demonstrations and descriptions of the task that are under-specified or misspecified. For example, when booking airline tickets, a human may forget to describe their preference for red-eye flights and there may not be enough demonstrations to determine their underlying preferences [1]. In its current form, the method appears unequipped to handle such cases and it is unclear what solution it would generate. As mentioned in the work, one possible remedy is to provide feedback to the LM, but it is unclear how successful such an approach would be.

[1]  Lin, J., Fried, D., Klein, D., & Dragan, A. (2022). Inferring Rewards from Language in Context. In Proceedings of the 60th Annual Meeting of the Association for Computational Linguistics (Volume 1: Long Papers) (pp. 8546–8560). Association for Computational Linguistics.



**Questions:**

Questions:
* Is there intuition for why the pass performance of DemoNoLang2Code is greater than the pass performance of Demo2Code on the "Make a burger stacking lettuce atop patty immediately" and the "Make two burgers stacking patty atop lettuce after preparation" tasks in Table 2?

**Limitations:**

In general the limitations are well-addressed. The following two limitations were also addressed as weaknesses above:
* Demonstrations are complete descriptions of low-level actions. In practice it seems difficult to obtain complete descriptions of all actions and state of an environment, as actions are often noisy and difficult to clearly delineate. E.g., given some video of a human washing the dishes, it would be unclear how to appropriately parse the action space (in the paper it was done by hand).
* The natural language description of the task may be misspecified by the human. This seems like a fundamental limitation of the current method, and might make the generated code incorrect or misspecified.

---

> ### Author Rebuttal · Authors · 2023-08-10
>
> We thank the reviewer's enthusiasm for our cooking game Robotouille and how our approach shows improvement over a wide range of tasks and domains. We also appreciate the reviewer's feedback on our assumptions. We would like to address the questions and concerns raised:
>
> ### Questions
> #### **Q1: Why is the unit pass rate of DemoNoLang2Code sometimes greater than Demo2Code?**
> The two instances in Table 2 of the paper where DemoNoLang2Code performed better than Demo2Code are due to errors made by the summarization module in Demo2Code. For example, when Demo2Code summarizes the demonstrations for the "make two burgers stacking patty atop lettuce after preparation" task, the LLM accidentally makes a mistake and misses the high-level subtask of stacking the patty on top of the lettuce. Consequently, the task specification also misses this subtask. We are working on better verification techniques to catch such mistakes and improve code generation.
>
> ### Weakness
> #### **Clarification of assuming access to state-based demonstrations**
> We refer the reviewer to the global response where we clarified the scope. We envision that our pipeline will exist in conjunction with other perception modules that parse the environment and demonstrations into symbolic states. As a part of future work, we are integrating Demo2Code into an end-to-end robotics system.
>
> #### **Ablations where some states are noisily parsed or entirely omitted?**
> Please refer to the appendix (section G Table 1), where we have run an ablation study with the EPIC-KITCHENS dataset.
> - We try adding noisy, distracting predicates (e.g. showing the position of additional objects in the scene) to at least two separate states in the demonstrations to confuse the LLM.
> - We find that Demo2Code's performance suffers from degradation.
> - Specifically, it originally can correctly extract 5 out of 7 users' preferences, but with noisy states, it can only correctly solve for 3 users.
>
> We also present new noisy state ablation experiments for Robotouille in the attached pdf (Table 1).
> - Specifically, for a demonstration, we study the effect of randomly removing 10% of the predicates and the effect of randomly removing 10% of states completely.
> - We find that missing 10% of the predicates in demonstrations only slightly worsens the performance (from 0.465 to 0.42).
> - Meanwhile, because removing 10% of the states essentially removed more than 10% of the predicates, it has more greatly affected the performance (from 0.465 to 0.327).
>
> Table 1 in the attached pdf contains detailed results for individual Robotouille tasks.
>
> #### **Clarification of using language description and demonstrations as input**
> We agree with the reviewer that language descriptions of a task can be under-specified or misspecified. We offer the following two arguments.
> 1. Demonstrations may capture complementary information that language description omits. Even if demonstrations and language are both noisy, as long as one modality captures the user’s preference, Demo2Code can recover it in the specification. Using the airline ticket booking examples, if a user shows a demonstration where a red-eye flight is picked over other ones that are cheaper, our pipeline can extract that preference from the demonstrations even if the user never explicitly states that they prefer red-eye flights.
> 2. In the event that both demonstrations and language miss such information, we would need feedback from the user to add what's missing. Recent works [1-4] have shown such feedback schemes are able to correct LLMs' output successfully. We should be able to extend Demo2Code to add such feedback, and we will explore this in future work.
>
> [1] Marta Skreta, et al. Errors are useful prompts: Instruction guided task programming with verifier-assisted iterative prompting, 2023.
>
> [2] Shreyas Sundara Raman, et al. Planning with large language models via corrective re-prompting, 2022.
>
> [3] Debjit Paul, et al. Refiner: Reasoning feedback on intermediate representations, 2023.
>
> [4] Noah Shinn, et al. Reflexion: Language agents with verbal reinforcement learning, 2023.

---

> > ### Comment · Reviewer_ujZb · 2023-08-10
> > **Thank you for the clarification**
> >
> > .

---

### Official Review · Reviewer_sxrs · 2023-07-06

**Soundness:** 2 fair
**Presentation:** 3 good
**Contribution:** 2 fair
**Rating:** 6
**Confidence:** 3

**Summary:**

The authors propose an LLM-based completion framework to translate natural language instructions, in addition to transcribed state sequences of demonstrations (as PDDL (or other strips-like) predicates), into code for executing the task with a robot.

The method is based on recursively summarizing the demonstrations into a 'specification', and then recursively expanding the 'specification' into python(?) code. The recusrsion stops in the first phase when no further summarization is possible, and in the code expansion loop, when no function is undefined.

The authors demonstrate this is three different domains with upto 10 distinct high level actions occuring in each demonstration.

**Strengths:**

Multimodal learning from demonstrations plus language instructions is an important problem that will enable consuming a wider modality of data sources to generate robot/agent programs and behavior. Using the summarization/translation capabilities of LLMs will likely be a key component of such a system.

**Weaknesses:**

**Methodology**: The paper does not define what a task specification is concretely, nor does it define what it means for the demonstrations to be adequately summarized, in particular in Alg 1 the function is_summarized() is not defined, nor is its operationalization defined anywhere in the text of the paper. Specification traditionally refers to a formal statement whose semantics are well known and the satisfaction of the specification with respect to an output is computationally well-defined and consistent. By not having a concrete definition of specification, the evaluations set up in the latter part of the paper suffer from lack of diversity and a lack of quantification of task difficulty.

**Code as output language**: This work relies on the output code being interpretable, and this involves providing a set of primitive parametric functions to the LLM to use. The part of sensing the environment, and acting using perception feedback is something that is already programmed into the primitives. Coming up with an adequate and a competent set of parameterized primitives is challenging, and in this case is entirely the responsibility of the system designer.

**Claims of generalization** Tthe generalization shown is only generalizing towards named entity substitutions, not to complex control flows and temporal specifications. Specifically, all the tasks shown here are a sequence of subgoals where performing any subgoal out of order will not preclude the agent from completing the subsequent actions. Further, there is no reactivity in the task specifications, and there are no avoidance tasks. All of these task specifications notions are very common in robotics and planning problems that this system has not been evaluated on. Refer to [1] for a survey on robotics mission types (also relevant to symbolic planning). All the tasks here are limited to the visit or sequenced visit type. Further while the submission claims that demo2code can generalize for complex long-horizon tasks, the maximum task length is quite smaller than state of the art for symbolic planning. Automatically translating textual domain descriptions to a formal domain description followed by the use of automated planners has already shown more reliable performance on harder problems [5]. Further there is quite a bit of evidence that LLMs cannot plan beyond the simplest of domains [6], and this line of research is unacknowledged in the submission.

**Issues on evaluations of learning from demonstrations:** Generalization from demonstrations and language is a tricky subject. Usually demonstrations and language contain complementary sources of information. Therefore none of the system behaviors are incorrect in Figure 5, the core issue is that inductive learning is by definition an ill-posed problem, and many approaches to inductive learning have relied on Bayesian inference in the past. [2],[3]. Specifically, the issue of where to place the purple block (fig 5a) given the language description is underspecified, and the system had to forcibly ground the placement to any of the valid options to generate a trajectory. Committing to a valid assignment as was done by the lang2code model is one approach, and asking for resolution of referential ambiguity is another approach [4]. One might argue that learning to overconstrain the output based on a single demonstration is an example of overspecification.



[1] - Menghi, C., Tsigkanos, C., Pelliccione, P., Ghezzi, C. and Berger, T., 2019. Specification patterns for robotic missions. IEEE Transactions on Software Engineering, 47(10), pp.2208-2224.

[2] - Tenenbaum, J.B., 1999. A Bayesian framework for concept learning (Doctoral dissertation, Massachusetts Institute of Technology).

[3] - Shah, A., Kamath, P., Shah, J.A. and Li, S., 2018. Bayesian inference of temporal task specifications from demonstrations. Advances in Neural Information Processing Systems, 31.

[4] - Williams, Tom, Rafael C. Núñez, Gordon Briggs, Matthias Scheutz, Kamal Premaratne, and Manohar N. Murthi. "A dempster-shafer theoretic approach to understanding indirect speech acts." In Advances in Artificial Intelligence--IBERAMIA 2014: 14th Ibero-American Conference on AI, Santiago de Chile, Chile, November 24-27, 2014, Proceedings 14, pp. 141-153. Springer International Publishing, 2014.

[5] - Liu, B., Jiang, Y., Zhang, X., Liu, Q., Zhang, S., Biswas, J. and Stone, P., 2023. Llm+ p: Empowering large language models with optimal planning proficiency. arXiv preprint arXiv:2304.11477.

[6] - Valmeekam, K., Olmo, A., Sreedharan, S. and Kambhampati, S., 2022. Large Language Models Still Can't Plan (A Benchmark for LLMs on Planning and Reasoning about Change). arXiv preprint arXiv:2206.10498.

**Questions:**

To strengthen the paper the authors should carefully define the following:
1. The role of the user/system developer. This includes definition of skill primitives that can be executed in the environment, definition of predicates that are sufficient to track the progress of the tasks. Developing perception systems that accurates record the predicate states
2. Quantify task complexity using metrics from logic (expressing instructions in temporal logic, and measuring the size of the automaton). Using number of predicates, and actions and reporting them for each planning domain.
3. Evaluate on diverse set of instructions taking inspiration from Menghi et al. [1] to come up with specification templates beyond sequenced visit.
4. Report comparative performance against state-of-the-art symbolic planners. in comparably sized planning domains.

**Limitations:**

Please see above

---

> ### Author Rebuttal · Authors · 2023-08-10
>
> We appreciate the reviewer's valuable feedback on how we can strengthen our work. We also thank the reviewer for seeing our approach's capability to summarize and handle learning from demonstrations!
>
> Please find below our responses to the questions and concerns:
>
> ## Questions
> #### **Q1:**
> We assume that a system developer does the following:
> - Provides a perception library. This library defines a set of predicates, similar to [8-9], and maintains this set based on sensor observations. It also contains helper functions that use these predicates, e.g. get_obj_location(obj), is_cooked(obj).
> - Provides an action library. This library defines a set of actions that correspond to low-level policies, similar to [1-7].
>
> These are common modules in a robotics stack and seem reasonable to assume. Given such a system, Demo2Code can take demonstrations from any user (not a system developer) to generate robot code that uses functions from these libraries. We list the set of low-level actions and predicate-based functions for each planning domain in Table 3 of the attached pdf.
>
> #### **Q2:**
> We present the number of unique predicates and low-level actions for each planning domain in Table 2 of the attached pdf. We also cluster the tasks based on their type to report the demonstration length and code complexity (by code length, average number of loops, average number of conditionals, and average number of functions defined) in Table 4.
>
> #### **Q3:**
> We clarify our rationale for our selection of tasks, which has a different motivation than Menghi et al. [10] Since we focus on learning tasks from demonstrations rather than task planning efficiency, we chose our tasks to test the following:
> - **Recovering hidden world constraints:** The tabletop tasks require identifying constraints not specified in language but implicit in the demonstrations.
> - **Inferring conditionals and control flows:** The Robotuille tasks require identifying conditionals (e.g. picking up the object if not holding it) and loops (e.g. cutting an object until it's cut, using for loops to cook multiple burgers, etc).
> - **Extract user preferences:** The Epic Kitchen tasks require identifying the user’s preference, e.g. different ways to wash dishes.
>
> #### **Q4:**
> We adapted our pipeline to run LLM with a symbolic planner, LLM+P [11]. Since LLM+P does not have a recursive summarization pipeline to handle long demonstrations, we provide the task specification generated by Demo2Code.
>
> We made these observations:
> - LLM+P fails to capture user preferences if they are not observable in the goal state. Fig. 1 in the attached pdf shows a qualitative example. LLM+P produces the left plan which has a lower cost, but misses the preference that the user wants the robot to prepare all ingredients before assembling them into a burger because the order of subtasks is not captured in the goal state.
> - LLM+P needs to be called every time the initial condition changes. In contrast, Demo2Code needs to generate the code once, which generalizes for different initial conditions.
>
> ## Weakness
> ### Methodology
> In our work, the task specification is described in language rather than a formal structure. We empirically found the following format to work reliably.
> - Header: 1-2 sentences that define the overall goal to ground the code generation step.
> - Body: descriptions in pseudocode format to help the LLM generate the high-level task code more easily.
>
> A demonstration is sufficiently summarized when it has been distilled down to a task specification that can “explain away the demos”, i.e. P(code | spec) = P(code | spec, demo).
>
> In our implementation, we leverage the LLM to rely on instructions and examples in the prompt to determine whether a demo is sufficiently summarized.
>
> ### Claims of generalization
> Please refer to the global response where we clarify:
> - The difference between the train and test tasks
> - The code complexity of the different test tasks
>
> We show that Demo2Code can generalize to tasks that have longer horizons, more states, and more control flows than examples in the prompt. Our long-horizon claims refer to the result that Demo2Code can solve longer tasks (up to 114 states) compared to existing LLM-based planners [1-7] (up to 20 states).
>
> #### **On the planning ability of LLMs**
>
> Please refer to the "Code vs other policy representation" section where we clarify that we focus on using LLMs to generate robot task codes from demonstrations rather than LLMs' planning abilities. We acknowledge the importance of symbolic planners and their complementary strengths to our summarization framework.
>
> ### Issues on evaluations of learning from demonstrations
> We evaluate solutions in a manner similar to how imitation learning evaluates policies:
> - When we generate demonstrations, we also generate held-out reward functions, captured as unit tests, that check for satisfying a set of sub-goals and constraints.
> - We use these held-out unit tests to evaluate the code.
>
> [1] Jacky Liang, et al. Code as policies: Language model programs for embodied control
>
> [2] Ishika Singh, et al. Progprompt: Generating situated robot task plans using large language models
>
> [3] Jimmy Wu, et al. Tidybot: Personalized robot assistance with large language models
>
> [4] Andy Zeng, et al. Socratic models: Composing zero-shot multimodal reasoning with language
>
> [5] Brian Ichter, et al. Do as i can, not as i say: Grounding language in robotic affordances
>
> [6] Wenlong Huang, et al. Inner monologue: Embodied reasoning through planning with language models
>
> [7] Kevin Lin, et al. Text2motion: From natural language instructions to feasible plans
>
> [8] Toki Migimatsu, et al. Grounding predicates through actions
>
> [9] Kei Kase, et al. Transferable task execution from pixels through deep planning domain learning
>
> [10] Claudio Menghi, et al. Specification patterns for robotic missions
>
> [11] Bo Liu, et al. LLM+P: Empowering large language models with optimal planning proficiency

---

> > ### Comment · Reviewer_sxrs · 2023-08-14
> > **Thank you for the response**
> >
> > **User-defined modules**: I appreciate this reflection, and it would be a valuable addition to the paper to better define the user's/systems designer's role
> >
> > **Role of specification patterns**: I believe the authors misunderstood the intent of comparing selected tasks to Menghi's [ref-1 mine] framework. Menghi's framework is a taxonomy of temporal properties present in robotics tasks, and the authors should select tasks that fall into various categories from the hierarchy, and the selected tasks only fall under a few of the temporal properties.
> >
> > I also appreciate the author's efforts to characterize the complexity of the tasks that the system was tested on.
> >
> > **Definition of summarization**: I appreciate the clarification, and would like the authors to clarify if the computation of whether the demonstration is summarized actually leverages the 'explained away' criterion as mentioned (using logits of the LLM), or if it is asking an LLM to evaluate whether the instruction is adequately summarized?
> >
> > Thus I feel the authors have reiterated their position more clearly in the response, the central issue with the lack of diversity of temporal properties in tasks learned from demonstration still remains. As it stands I plan to retain my current score

---

> > > ### Author Response · Authors · 2023-08-16
> > > **Thank you for your feedback (1/2)**
> > >
> > > ## Question
> > > **Does the computation of whether the demonstration is summarized actually leverage the 'explained away' criterion as mentioned (using logits of the LLM), or does it ask an LLM to evaluate whether the instruction is adequately summarized?**
> > >
> > > - At test time, the LLM is asked to evaluate if a trajectory is adequately summarized.
> > > - However, when we design the summarization prompt, we validate the prompt by using tasks with short demonstrations to check if concatenating those demonstrations to the beginning of the generated specification changes the code.
> > >     - If specification alone and specification with demonstrations cause the LLM to generate the same code, we can approximately show that the prompt satisfies the criteria that P(code | spec) = P(code | spec, demo).
> > > - Thus, our prompt validation approach essentially examines the arg max of the logits, which is the code, instead of the exact logits whose values may vary.
> > >
> > > ## Clarification on specification pattern
> > >
> > > We would like to emphasize that since we are predicting code as output, we must measure the complexity of tasks in terms of their code complexity. We categorize our tasks under a taxonomy of the number of loops, conditionals, functions, code length, and horizon lengths (see Table 4 in the attached pdf). Our tasks do coverage on these different axes. This is also consistent with how prior works [1-4] predicting task code have characterized their set of tasks.
> > >
> > > The challenge with using a different taxonomy, e.g. Menghi et al.[5], is a potential mismatch in categories in the taxonomy v.s. varying levels of complexity for the code generation model. Notably, two different categories can result in very similar code. For example:
> > > - **Avoidance category:** Don’t stack blocks above a certain height results in  "while stack_height < X"
> > > - **Trigger category:** Wait till patty is cooked results in "while not is_cooked(patty)"
> > > - Both are simply while() loops with different conditions.
> > >
> > > However, we are happy to characterize the current tasks we have in Menghi et al.[5]'s taxonomy as the reviewer requested. We also introduce 3 new tasks to increase the diversity in the new taxonomy. Please see the table below:
> > > | Planning Domain | Task Name | global avoidance (Avoidance) | upper restriction avoidance (Avoidance) | lower/exact restriction avoidance (Avoidance) | wait (Trigger) | instantaneous reaction (Trigger) | delayed reaction (Trigger) | patrolling (Surveillance) |
> > > |:---:|:---:|:---:|:---:|:---:|:---:|:---:|:---:|:---:|
> > > | Tabletop | Place A on top of B |  |  |  |  | X |  |  |
> > > |  | Stack all blocks/cylinders |  | X |  |  |  |  |  |
> > > |  | Stack all objects into two stacks | X |  |  |  |  |  |  |
> > > | Robotouille | Cook and cut |  |  | X | X | X |  |  |
> > > |  | Cook two patties |  |  |  | X | X |  |  |
> > > |  | Cut two lettuces  |  |  | X |  |  |  |  |
> > > |  | Assemble two burgers |  |  |  |  |  |  |  |
> > > |  | Make a burger | X |  | X | X | X | X |  |
> > > |  | Make two burgers | X |  | X | X | X | X |  |
> > > |  | **Keep making burgers (new)** | **X** |  | **X** | **X** | **X** | **X** | **X** |
> > > |  | **Keep assembling burgers (new)** | **X** |  |  |  |  | **X** | **X** |
> > > |  | **Cooking multiple patties simultaneously (new)** |  |  |  | **X** | **X** |  | **X** |
> > > | EPIC-KITCHENS | Washing dishes |  |  | X |  | X |  |  |
> > >
> > > For the new "keep making burgers" and "keep assembling burgers" clusters of tasks, we report Demo2Code's execution success rate and unit test pass rate below:
> > > | Task Cluster | Requirements | Execution Success Rate | Unit Test Success Rate |
> > > |:---:|:---:|:---:|:---:|
> > > | Keep making burgers | stacking lettuce atop patty immediately | 0 | 0 |
> > > |  | stacking patty atop lettuce immediately | 1 | 0 |
> > > |  | stacking lettuce atop patty after preparation  | 0 | 0 |
> > > |  | stacking patty atop lettuce after preparation  | 0 | 0 |
> > > |  | substitute lettuce with cheese | 1 | 1 |
> > > |  | substitute patty with chicken | 1 | 1 |
> > > |  | add tomato | 1 | 1 |
> > > | Keep assembling burgers | none | 1 | 1 |
> > > |  | do the tasks in parallel | 1 | 0 |
> > > | Overall |  | 0.67 | 0.44 |
> > >
> > > [1] Jacky Liang, et al. Code as policies: Language model programs for embodied control
> > >
> > > [2] Ishika Singh, et al. Progprompt: Generating situated robot task plans using large language models
> > >
> > > [3] Jimmy Wu, et al. Tidybot: Personalized robot assistance with large language models
> > >
> > > [4] Andy Zeng, et al. Socratic models: Composing zero-shot multimodal reasoning with language

---

> > > > ### Author Response · Authors · 2023-08-16
> > > > **Thank you for your feedback (2/2)**
> > > >
> > > > To clarify how we characterize our tasks, we provide one specific example for each mission specification pattern:
> > > > - Global avoidance
> > > >     - For the Robotouille "make a burger" tasks, although the language instruction just says "make a burger", the demonstrations show that the user wants to avoid using any burger patty and use chicken instead.
> > > >     - **"Not using burger patty" is an avoidance condition that globally holds throughout the mission.**
> > > > - Upper restriction avoidance
> > > >     - For the tabletop "stack all blocks/cylinders" tasks, a hidden world constraint (a maximum height for a stack) needs to be inferred from the demonstrations and is not explicitly stated in the language instruction.
> > > >     - **"Only placing a maximum of X number of objects into a stack" is a  restriction on the maximum number of occurrences (placing objects into a stack).**
> > > > - Lower/Exact restriction avoidance
> > > >     - In Robotouille, when the robot needs to cut lettuce, the demonstrations show that the robot needs to keep performing the "cut" action until the lettuce is cut. Currently, the unit test makes sure that the robot at least performs the minimal amount of "cut" actions in order to successfully cut the lettuce. Demo2Code generates a code that even satisfies the stricter requirements of performing the exact amount of "cut" actions because the code keeps calling cut() while the lettuce is not cut.
> > > >     - **"Cut the lettuce at least X amount of time" is a restriction on the minimum number of occurrences (performing cut actions).**
> > > >     - **"Cut the lettuce exactly X amount of time" is a restriction requiring an exact amount of occurrences.**
> > > > - Wait
> > > >     - In Robotouille, when the robot needs to cook a patty, the demonstrations show that after the robot starts the cooking process, it needs to wait until the patty is cooked.
> > > >     - **"Wait until the patty is cooked" is an example of when inaction (wait) is desired until a stimulus (patty becomes cooked) occurs.**
> > > > - Instantaneous reaction
> > > >     - For some variants of the "make a burger" Robotouille tasks, when the patty is cooked, the demonstrations show that the robot should immediately pick the patty up for subsequent subtasks (e.g. stacking the patty on top of the rest of the burger).
> > > >     - **"Once the patty is cooked, immediately pick it up" is an example of when the occurrence of a stimulus (the patty is cooked) instantaneously triggered a counteraction (pick up that patty).**
> > > > - Delayed reaction
> > > >     - For some variants of the "make a burger" Robotouille tasks, instead of immediately picking up the patty once it's cooked, the demonstrations show that the robot should complete some other subtasks (e.g. cutting the lettuce) before returning to pick up that patty. This behavior is applicable to when the user prefers the robot to prepare all the ingredients before assembling the burger.
> > > >     - **"Once the patty is cooked, pick the patty up but it is not necessary to do it immediately" is an example of when the occurrence of a stimulus (the patty is cooked) triggers a counteraction (pick up that patty) some time later.**
> > > > - Patrolling
> > > >     - Although the surveillance category where the robot needs to keep performing a task is out of the scope of our paper, we adapted some existing tasks to new ones with patrolling patterns to test the Demo2Code pipeline.
> > > >     - The "keep making burgers" and "keep assembling burgers" Roboutille tasks contain demonstrations where the robot would keep working until there are no more ingredients in the environment to use.
> > > >     - Another patrolling task for future work is when the robot is simultaneously cooking multiple patties.
> > > >         - Once the robot starts cooking, the robot needs to continuously patrol each stove to check on the patty. If a patty is cooked, the robot needs to remove that patty from the stove and place it on the table.
> > > >     - **"Keep visiting all the stoves" represents the requirement to keep visiting a set of locations (stoves), but not in a particular order.**

---

> > > > > ### Comment · Reviewer_sxrs · 2023-08-16
> > > > > **Thank you for adding this context**
> > > > >
> > > > > Addressing coverages under various taxonomies is a valuable excercise in identifying test cases to stretch system capabilities, using both taxonomies is helpful. This paper can set a great precedent for follow-on works even if past LLM planning papers have not considered the formal specification taxonomy. This alone significantly strengthens the paper.
> > > > >
> > > > > I am willing to raise my score to weak accept. My only holdout would be that evaluation standards in LLM planning papers and classical planning papers are quite divergent. A key principle in classical planning papers is to test the system to failure by progressively increasing task complexity, and while there are elements of it here, the two streams of research seem to be targetting different meanings of complexity, and there is little overlap between accepted evaluation methodologies in these areas despite having overlapping applications. However addressing this gap is best suited for future work, and with the inclusion of this and some additional discussion on how to evaluate LfD approaches in the final version, I will not argue against acceptance.
> > > > >
> > > > > Examples of pipelines going from demo to robot policies in the classical planning domain include works on inferring formal specifications from demonstrations in languages such as LTL and planning with respect to those formulas to generate code, and some examples of the works are provided below. While these are not direct baselines as Demo2Code has complementary strengths and adopts a different solution approach. There are lessons to be learned on evaluating inductive learning problems on complex tasks here.
> > > > >
> > > > > Specification inference from demonstrations (as LTL or automata) (an incomplete list):
> > > > >
> > > > > [1] - Vazquez-Chanlatte, M., Jha, S., Tiwari, A., Ho, M.K. and Seshia, S., 2018. Learning task specifications from demonstrations. Advances in neural information processing systems, 31.
> > > > >
> > > > > [2] -  Shah, A., Kamath, P., Shah, J.A. and Li, S., 2018. Bayesian inference of temporal task specifications from demonstrations. Advances in Neural Information Processing Systems, 31.
> > > > >
> > > > > [3] - Toro Icarte, R., Waldie, E., Klassen, T., Valenzano, R., Castro, M. and McIlraith, S., 2019. Learning reward machines for partially observable reinforcement learning. Advances in neural information processing systems, 32.
> > > > >
> > > > > Automated policy generation from specifications (again an incomplete list):
> > > > >
> > > > > [4] - De Giacomo, G., Iocchi, L., Favorito, M. and Patrizi, F., 2019. Foundations for restraining bolts: Reinforcement learning with LTLf/LDLf restraining specifications. In Proceedings of the international conference on automated planning and scheduling (Vol. 29, pp. 128-136).
> > > > >
> > > > > [5] - Icarte, R.T., Klassen, T.Q., Valenzano, R. and McIlraith, S.A., 2022. Reward machines: Exploiting reward function structure in reinforcement learning. Journal of Artificial Intelligence Research, 73, pp.173-208.
> > > > >
> > > > > [6] - Shah, A., Li, S. and Shah, J., 2020. Planning with uncertain specifications (puns). IEEE Robotics and Automation Letters, 5(2), pp.3414-3421.

---

> > > > > > ### Author Response · Authors · 2023-08-16
> > > > > > **Thank you for your suggestions**
> > > > > >
> > > > > > Thank you for your time and energy to help us improve and strengthen our paper!
> > > > > >
> > > > > > We appreciate the discussions about the gap in the evaluation standards for LLM-based planners and classical planners. We will incorporate a section discussing this gap and how to evaluate LfD approaches in the final version of the paper.

---

### Official Review · Reviewer_Gc7K · 2023-07-06

**Soundness:** 3 good
**Presentation:** 3 good
**Contribution:** 3 good
**Rating:** 7
**Confidence:** 3

**Summary:**

This paper presents a method that can take both demo and language in and teach LLM to perform new tasks. The idea makes sense and the algorithm is easy to understand and works very well. Evaluation results and ablations show improvement over existing works.

**Strengths:**

The paper is well written and the idea is clear and easy to understand.

**Weaknesses:**

It's probably better to define what "task" is. For example, the authors claim "Demo2Code can generalize across complex, long-horizon tasks." However, for real world robotics tasks, I would imagine there will be a lot of corner cases and the code needs to handle it and therefore it won't generalize to those tasks.

**Questions:**

It looks like the method does not involve any network training / fine tuning, right?
If so, do you expect to get better performance by training / fine tuning LLM?

**Limitations:**

As the author said, the ability of this framework is limited by the capability of LLM.

---

> ### Author Rebuttal · Authors · 2023-08-09
>
> We appreciate that the reviewer is excited about the capability of our approach and finds our paper to be clear and easy to understand. We also thank the reviewer for suggestions to make our paper clearer. We would like to answer the questions and then address the concerns.
>
> ### Questions
> #### **Q1: It looks like the method does not involve any network training / fine-tuning, right?**
> Correct, we are currently doing in-context learning, where we provide the LLM example query and responses [8].
>
> We do expect fine-tuning and updating the weights of the LLMs to increase the performance. However, fine-tuning large models is challenging because:
> - it requires access to the parameters
> - the process is computationally intensive
> - it requires a large amount of data due to the large number of parameters LLMs have
>
> Moreover, open-source models currently do not match the code-generating abilities of GPT3.5 [9].
>
> Hence, in-context learning with GPT-3.5 allows quick reusability across different environments without needing large amounts of data and computational power.
>
> ### Weakness
> #### **Clarification on tasks**
> We define a task as a goal that contains a series of sub-goals, which correspond to sub-tasks that the robot needs to complete. For example, the make-a-burger tasks contain sub-goals on the item's status (e.g. the lettuce needs to be cut, the lettuce must be on top of the patty) and on the order of sub-tasks to execute (e.g. preparing all the ingredients before assembling the burger).
>
> #### **Clarification on generalization ability to complex, long-horizon tasks**
> In the global response, we clarify how Demo2Code is able to generalize to tasks that are more complex and have a longer horizon than the tasks in the prompt, e.g. for Robotouille, an average of 11 states and 15 predicates at train time v.s. 32 states and 53 predicates at test time.
>
> In addition, Demo2Code is able to solve tasks with longer horizons (up to 71 states) compared to other existing LLM-based planners [1-7], which have an average of around 20 states. These planners mostly solve tasks that require chaining multiple low-level actions. In contrast, the tasks that Demo2Code can resolve require chaining multiple high-level subtasks, which each contains multiple low-level actions with loops and conditionals.
>
> #### **Clarification on how the code handles corner cases**
> We acknowledge the importance of ensuring that the generated policy can handle corner cases. That is why we developed Robotouille, which procedurally generates environments with different initial conditions to test our code in. This variability includes:
> - the robot starts in a new, unseen position,
> - items are in novel arrangements,
> - key items exist among other irrelevant items, etc.
>
> As future work, we are working on integrating Demo2Code into an end-to-end robotic system that would surface more realistic corner cases.
>
> [1] Jacky Liang, et al. Code as policies: Language model programs for embodied control. arXiv preprint arXiv:2209.07753, 2022.
>
> [2] Ishika Singh, et al. Progprompt: Generating situated robot task plans using large language models, 2022.
>
> [3] Jimmy Wu, et al. Tidybot: Personalized robot assistance with large language models, 2023.
>
> [4] Andy Zeng, et al. Socratic models: Composing zero-shot multimodal reasoning with language. arXiv:2204.00598, 2022.
>
> [5] Brian Ichter, et al. Do as i can, not as i say: Grounding language in robotic affordances. In 6th Annual Conference on Robot Learning, 2022.
>
> [6] Wenlong Huang, et al. Inner monologue: Embodied reasoning through planning with language models. In arXiv:2207.05608, 2022.
>
> [7] Kevin Lin, et al. Text2motion: From natural language instructions to feasible plans, 2023.
>
> [8] Mark Chen, et al. Evaluating large language models trained on code. arXiv:2107.03374, 2021.
>
> [9] Hugo Touvron, et al. Llama: Open and efficient foundation language models, 2023.

---

> > ### Author Response · Authors · 2023-08-21
> > **Thank you for your feedback**
> >
> > Hello! Thank you again for your review and questions! Since the discussion period is coming to a close, please let us know if any additional clarifications would be helpful in your evaluation.

---

### Author Rebuttal · Authors · 2023-08-10

We thank all reviewers for their time, energy, and helpful feedback! We are excited to see that reviewers view the problem as important and challenging *(Reviewer sxrs, ujZb, 325h, yYo2)*, find our LLM summarization framework to be novel *(325h)* and important *(sxrs, yYo2)*, and view Robotouille as a new benchmark for high-level task planning *(ujZb, yYo2)*. We are also pleased to see that reviewers find the paper well written *(Gc7K, ujZb)* and figures easy to understand *(ujZb, yYo2)*.

**Please find an attached pdf with new ablation studies, baseline comparison, and experiment details.**

### Clarification of scope
We build on prior works [1-4] that leverage LLMs to generate robot code from detailed language instructions. In contrast, Demo2Code generates robot task code from state-based demonstrations. People often cannot provide detailed language instructions and tend to underspecify. With Demo2Code, we show that even with little or no language input, we are competitive with prior works [1] that have access to detailed language instructions.

### Access to state-based demonstrations
Most robotic systems have perception modules [5-6] that can parse raw sensor data into predicate states. We think assuming access to state-based demonstrations is reasonable so that we can focus on generating code, which is nontrivial given how long demonstrations can get. As future work, we are working on integrating Demo2Code into an end-to-end robotic system.

We also conducted several ablations:
- Table 1 in Section G of the appendix shows ablation where we add noisy predicates to the EPIC-KITCHENS demonstrations. We find that the LLM's performance worsens and goes from correctly extracting 5 users' preferences to 3.
- We also conduct new studies on Robotouille where 10% of the predicates or 10% of the states are randomly removed. Missing 10% of the predicates only slightly worsens the performance (0.462 to 0.42), and missing 10% of the states (which could remove more than 10% of predicates) have a larger effect (0.462 to 0.327). More details are in Table 1 of the attached pdf.

### Why code?
We choose code as the output representation because:
- Much of high-level robot tasks are programmed as code.
- Code offers a flexible, concise way of expressing control flows, calling external libraries like perception and planning, etc.
- Code allows composability.
- Code is interpretable to engineers and verifiable through static analysis.
- LLMs have been trained on code and can generalize well in this output space [8-9].

### Code v.s. other policy representation
We don't claim our approach is an alternative to a symbolic planner. In fact, many problems such as re-arrangement tasks are more suited to be solved as a planning problem. For such cases, we can modify Demo2Code to summarize demonstrations to a task specification for a (PDDL) planner.

We experiment combining Demo2Code with LLM+P [7] and present a qualitative example in Figure 1 of the attached pdf. We find that while this combination produces valid plans:
- LLM+P fails to capture user preferences that aren’t observable in the final state.
- LLM+P needs to be run for every new initial condition, while the code generalizes to different environments.

Still, both code and PDDL are valid ways to represent a policy, and they can even co-exist, e.g. the code can call a planner. However, for the reasons stated in "Why code?", we chose to generate code.

### Clarification of generalization results
We clarify how Demo2Code can generalize and solve unseen complex tasks with longer horizons and more predicates compared to examples in the prompt at train time.

For Robotouille, the tasks at train time are significantly different from the ones at test time:
- Mean horizon length: 11 states (train) v.s. 32 states (test)
- Mean number of predicates: 15 predicates (train) v.s. 53 predicates (test)

Compared to baselines, Demo2Code performs the best for long burger-making tasks (an average of 26 states and 43 predicates) even though the prompt doesn't show this type of task. Our pipeline also shows its ability to identify control loops, e.g. solving a task to make two burgers with patties, lettuce, and tomatoes (which has 71 states and 115 predicates). Demo2Code used a for-loop to make two burgers, generalized to unseen subtasks (e.g. cut tomatoes), and composed 7 distinct subtasks together to make one burger.

We also test the code against various initial conditions procedurally generated by the Robotouille simulator, where the robot starts in a new position, items are in novel arrangements, etc. These environments check the code against corner cases, e.g. when a key item is in a stack, the robot should unstack the item instead of directly picking it up.

### Clarification of the definition of "latent task specification"
We wish to clarify the sentence: “Our key observation is that the input (demonstrations) and the output (code) share a latent task specification”. We use the term “latent” to evoke the analogy with an encoder-decoder architecture (encoder being the summarizer and decoder being the code expansion). "Task specification” refers to the detailed language description of how the task should be completed.

[1] Jacky Liang, et al. Code as policies: Language model programs for embodied control

[2] Ishika Singh, et al. Progprompt: Generating situated robot task plans using large language models

[3] Jimmy Wu, et al. Tidybot: Personalized robot assistance with large language models

[4] Andy Zeng, et al. Socratic models: Composing zero-shot multimodal reasoning with language

[5] Toki Migimatsu, et al. Grounding predicates through actions

[6] Kei Kase, et al. Transferable task execution from pixels through deep planning domain learning

[7] Bo Liu, et al. LLM+P: Empowering large language models with optimal planning proficiency

[8] Mark Chen, et al. Evaluating large language models trained on code

[9] OpenAI. Gpt-4 technical report

---

### Decision · Program_Chairs · 2023-09-21

**Decision:**

Accept (poster)

**Comment:**

The paper presents Demo2Code, a method to generate robot task code from natural language instructions and demonstrations. It recursively summarizes demonstrations into a "task specification" using a language model, then expands the specification into executable code.

Reviewers found the problem important and challenging, and recommended acceptance across the board. They praised the novelty of the hierarchical summarization approach and the new cooking environment benchmark Robotouille. However, some questioned the assumptions of having full state access and well-specified language instructions.

In their rebuttal, the authors clarified the scope - the focus of the paper is on code generation given state-based demos, leaving state estimation to other modules, which I believe is a reasonable assumption given a typical robotics stack contains focused state estimation algorithms. Furthermore, the expectation is that code is a flexible policy representation for robot tasks. They added ablation studies showing performance degrades with noisy state inputs. For underspecified language, they proposed integrating user feedback loops to refine the generated code, which is similar to the conversational ability of mainstream chat-LLMs.

Overall, the authors appear to have satisfactorily addressed major reviewer concerns, with some differences in opinion around novelty and assumptions. The experiments demonstrate solid results on multiple domains. I would recommend acceptance given the authors expand on the scope and evaluation details in the camera ready version.